# m⁶A modification of mutant huntingtin RNA promotes the biogenesis of pathogenic huntingtin transcripts

Anika Pupak [1,2,3], Irene Rodríguez-Navarro[1,2,3], Kirupa Sathasivam[4], Ankita Singh [5,6],
Amelie Essmann[1,2,3], Daniel del Toro[1,2,3], Silvia Ginés [1,2,3], Ricardo Mouro Pinto [7,8,9], Gillian P Bates[4],
Ulf Andersson Vang Ørom [5], Eulàlia Martí[1,10] & Verónica Brito [1,2,3 ✉]

## Abstract

In Huntington's disease (HD), aberrant processing of huntingtin (*HTT*) mRNA produces *HTT1a* transcripts that encode the pathogenic HTT exon 1 protein. The mechanisms behind *HTT1a* production are not fully understood. Considering the role of m⁶A in RNA processing and splicing, we investigated its involvement in *HTT1a* generation. Here, we show that m⁶A methylation is increased before the cryptic poly(A) sites (IpA1 and IpA2) within the huntingtin RNA in the striatum of *Hdh+/Q111* mice and human HD samples. We further assessed m⁶A's role in mutant *Htt* mRNA processing by pharmacological inhibition and knockdown of METTL3, as well as targeted demethylation of *Htt* intron 1 using a dCas13-ALKBH5 system in HD mouse cells. Our data reveal that *Htt1a* transcript levels are regulated by both METTL3 and the methylation status of *Htt* intron 1. They also show that m⁶A methylation in intron 1 depends on expanded CAG repeats. Our findings highlight a potential role for m⁶A in aberrant splicing of *Htt* mRNA.

**Keywords** Huntington's Disease; Splicing; *HTT1a*; m⁶A; RNA Editing
**Subject Categories** Molecular Biology of Disease; Neuroscience; RNA Biology

## Introduction

Huntington's disease (HD) is considered the most common monogenetic neurodegenerative disorder showing dominant inheritance. This disorder is characterized by a triad of motor, cognitive and psychiatric symptoms that largely affect patients´ quality of life (McColgan and Tabrizi, 2018), eventually leading to their death 15-20 years after diagnosis (Walker, 2007). HD is caused by an unstable CAG repeat expansion in exon 1 of the gene that encodes huntingtin (*HTT*), resulting in an abnormally long polyglutamine tract in the huntingtin protein (HTT), which causes protein misfolding and consequent neurotoxicity (Ross and Tabrizi, 2011). In the context of expanded CAGs, it has been described that in addition to full-length (FL) *HTT* mRNA isoforms, two small transcripts containing exon 1 and a 5' region of intron 1 sequences (*HTT1a*) can be generated by incomplete splicing due to aberrant polyadenylation at cryptic polyA sites within intron 1 followed by premature termination of transcription (Neueder et al, 2018; Sathasivam et al, 2013). *HTT1a* not only encodes the aggregation-prone HTTexon1 protein that is known to be highly pathogenic (Mangiarini et al, 1996) but can also form mRNA nuclear clusters that are resistant to treatment with *HTT* antisense oligonucleotides (ASOs) (Fienko et al, 2022; Ly et al, 2022). As the alternative processing of *HTT* mRNA, and consequently the levels of *HTT1a* and HTTexon1, increase with increasing CAG repeat length (Neueder et al, 2017), it has been suggested that it may be a mechanism through which somatic CAG expansion exerts its pathogenic effects (Smith et al, 2022). Several regulatory mechanisms influencing the production of *HTT1a* have been proposed. Expanded CAG repeats within the *HTT* gene region can form RNA:DNA hybrid structures that impede or slow down RNA polymerase II (Pol II) elongation, thereby inducing a kinetically controlled disruption of splicing and polyadenylation within HTT intron 1 (Neueder et al, 2018). Additionally, splicing factors may bind to the expanded CAG repeats or within intron 1 of *HTT* mRNA, further influencing these processes (Schilling et al, 2019; Gipson et al, 2013; Neueder et al, 2018). However, the impact of RNA modifications on the transcriptional and posttranscriptional regulation of *HTT* has not yet been explored.

[1]Departament de Biomedicina, Facultat de Medicina, Institut de Neurociències, Universitat de Barcelona, Barcelona, Spain. [2]Institut d'Investigacions Biomèdiques August Pi i Sunyer (IDIBAPS), Barcelona, Spain. [3]Centro de Investigación Biomédica en Red sobre Enfermedades Neurodegenerativas (CIBERNED), Madrid, Spain. [4]Department of Neurodegenerative Disease, Huntington's Disease Centre and UK Dementia Research Institute at UCL, Queen Square Institute of Neurology, UCL, London WC1N 3BG, UK. [5]Department for Molecular Biology and Genetics, Aarhus University, Aarhus C, Denmark. [6]Department of Biomedicine, Aarhus University, Aarhus C, Denmark. [7]Center for Genomic Medicine, Massachusetts General Hospital, Boston, MA, USA. [8]Department of Neurology, Harvard Medical School, Boston, MA, USA. [9]Program in Medical and Population Genetics, Broad Institute of MIT and Harvard, Cambridge, MA, USA. [10]Centro de Investigación Biomédica en Red de Epidemiología y Salud Pública (CIBERESP), Madrid, Spain. ✉E-mail: veronica.brito@ub.edu

N(6)-methyladenosine (m⁶A) is the most abundant internal modification in eukaryotic mRNA (Satterlee et al, 2014; Cantara et al, 2011) and is present in 0.2-0.6% of all adenosines in mammalian mRNA (Roundtree et al, 2017a). The discovery of the m⁶A methyltransferase complex (METTL3, METTL14 and WTAP) and the demethylase proteins (FTO and ALKBH5) (writer and erasers, respectively) showed that this modification exhibits a dynamic pattern, strengthening its regulatory role in gene expression control (Shafik et al, 2021). Indeed, m⁶A has been shown to influence several steps of RNA metabolism, such as transcription of nascent RNA, including alternative splicing and polyadenylation, nuclear export, translation and finally degradation (Zhao et al, 2021; Xiao et al, 2016; Wang et al, 2022; Roundtree et al, 2017b; Zhou et al, 2019), processes that are mediated by m⁶A-reader proteins. Recently, we have demonstrated in the $Hdh^{+/Q111}$ mouse hippocampus that alterations of m⁶A modifications occur in mRNA during disease progression and that these alterations are involved in the cognitive disturbances of these HD mice (Pupak et al, 2022). One crucial finding of this work was a pronounced differential methylation in the proximal region of intron 1 of huntingtin transcripts. Notably, m⁶A can control pausing of Pol II and impact transcription termination (Akhtar et al, 2021; Yang et al, 2019) as well as slow down the kinetics of mRNA processing when deposited in introns (Louloupi et al, 2018). Altogether, these findings emphasize the need for a deeper characterization of the effects of m⁶A on the processing of *HTT* RNA. Therefore, to gain more insight into the potential role of m⁶A in HD pathology, we studied *Htt* m⁶A RNA modification in the striatum of HD samples and explored its contribution to *Htt1a* generation. Our findings indicate that m⁶A methylation is present in intron 1 of mutant huntingtin (*mHtt*) in the striatum of $Hdh^{+/Q111}$ mice and is further maintained upon maturation of the RNA. Moreover, we identified a GGACA motif present in the human sequence of intron 1 to be differentially methylated in human HD fibroblasts and *postmortem* samples, highlighting its pathological relevance. Notably, pharmacological inhibition, knockdown of METTL3 or targeted demethylation of *Htt* intron 1 specifically decreases the transcript levels of *Htt1a* in HD cells. Finally, we demonstrate that m⁶A methylation in intron 1 is likely dependent on CAG repeats. Collectively, our findings support the involvement of m⁶A in the generation of aberrantly spliced *Htt1a*, which could have important implications for gene therapy strategies designed to specifically lower *mHTT* in HD patients.

# Results

## m⁶A methylation levels of *Htt1a* are significantly increased in the striatum of $Hdh^{+/Q111}$ mice from early disease stages

Our previous study revealed by methylated RNA immunoprecipitation sequencing (MeRIP-seq) analysis a significant enrichment of m⁶A in the 5′ proximal region (278 bp downstream of the 5′ splice site in the mouse sequence) of *Htt* intron 1 in hippocampal samples from $Hdh^{+/Q111}$ mice (Pupak et al, 2022). To extend our findings to the most affected brain region in HD, we performed MeRIP followed by qPCR to analyze m⁶A methylation levels of *Htt* transcripts in the striatum of $Hdh^{+/Q111}$ mice at two different disease

stages, 2 and 8 months of age. First, qPCR analysis with different primer-probe sets spanning *Htt* was performed (Fig. 1A). The employed assays included probes before the first cryptic poly(A) site (I1-pA1) able to identify *Htt1a* transcripts terminated at both cryptic poly(A) sites; probes before the second cryptic poly(A) site (I1-pA2) that only detects those transcripts terminated at the second poly(A) site; probes at the 3′ end of intron 1 (I1-3′) to identify the incompletely spliced intron 1 sequences that have not terminated at cryptic poly(A) signals; probes spanning intron 3 to account for unspliced pre-mRNA (I3); and probes spanning exons 36 and 37 to determine full-length spliced *Htt* (FL-*Htt*). As it has been described in other knock-in mouse models (Sathasivam et al, 2013; Papadopoulou et al, 2019), qPCR analysis of input samples detected higher levels of intronic sequences generated at I1-pA1 and I1-pA2 in $Hdh^{+/Q111}$ mice than in WT mice at both ages (Fig. 1B,D). The levels of intronic sequences I1-3′ were comparable to intron 3; therefore, WT intronic sequences levels detected were indicative of pre-mRNA background, or that any level of incomplete splicing in WT mice might be below the levels of detection as previously described (Papadopoulou et al, 2019). The increase in intronic sequences I1-pA1 and I1-pA2 was accompanied by a significant decrease in FL-*Htt* compared to WT levels at 8 months but not at 2 months (Fig. 1C,E). Next m⁶A enrichment of these transcripts was evaluated through qPCR analysis of the m⁶A-immunoprecipitated RNA. To demonstrate selective enrichment for endogenous methylated targets using MeRIP, we used *Grm1* (positive control) and *Rps14* (negative control) to evaluate enrichment in the m⁶A immunoprecipitated and unbound fractions. These genes were selected due to their high abundance and distinct m⁶A peak presence (Meyer et al, 2012). As expected, we observed substantial immunodepletion of *Grm1* in the unbound fraction (Appendix Fig. S1A). In contrast, transcripts that lack m⁶A peaks such as *Rps14* were detectable at high levels in the unbound fraction (Appendix Fig. S1A). Our MeRIP-qPCR analysis detected a significant m⁶A enrichment in I1-pA1 transcripts in the striatum of 2- (Fig. 1F) and 8-month-old $Hdh^{+/Q111}$ mice (Fig. 1G; Appendix Fig. S1A), with a significant increase of m⁶A levels in I1-pA2 detected only at 8 months compared with $Hdh^{Q7Q7}$ mice (Fig. 1G). No differential enrichment was observed in the levels of intronic sequences at the 3′ end of intron 1 (I1-3′) or intron 3 (I3) (Appendix Fig. S1B), indicating that I1-pA1 and I1pA2 transcripts are specifically enriched in the striatum of $Hdh^{+/Q111}$ mice. These results further validate the differential peaks observed by MeRIP-Seq in these mice (Pupak et al, 2022). Interestingly, higher fold m⁶A enrichment in I1-pA1 and I1-pA2 transcripts is observed in the striatum at 8 months of age when somatic CAG expansions are significantly more abundant and of greater magnitude than at 2 months ($P < 0.0001$, Fig. EV1A,B), which is consistent with previous findings of somatic instability in this HD mouse model (Lee et al, 2011).

We further analyzed *Htt* transcripts levels and their m⁶A enrichment in ST$Hdh^{Q7/Q7}$ and ST$Hdh^{+/Q111}$ cells and we observed similar significant changes to that obtained by HD mice with higher levels of intronic sequences generated at I1-pA1 and I1-pA2 and reduced levels of FL-*Htt* in ST$Hdh^{Q111/Q111}$ compared with ST$Hdh^{Q7/Q7}$ cells (Appendix Fig. S2A,B). A significant m⁶A enrichment of I1-pA1 and I1-pA2 transcripts was also observed in ST$Hdh^{Q111/Q111}$ cells (Appendix Fig. S2C). Similar to $Hdh^{+/Q111}$ (Fig. 1F,G), the observed m⁶A enrichment in ST$Hdh^{Q111/Q111}$ cells was specific to

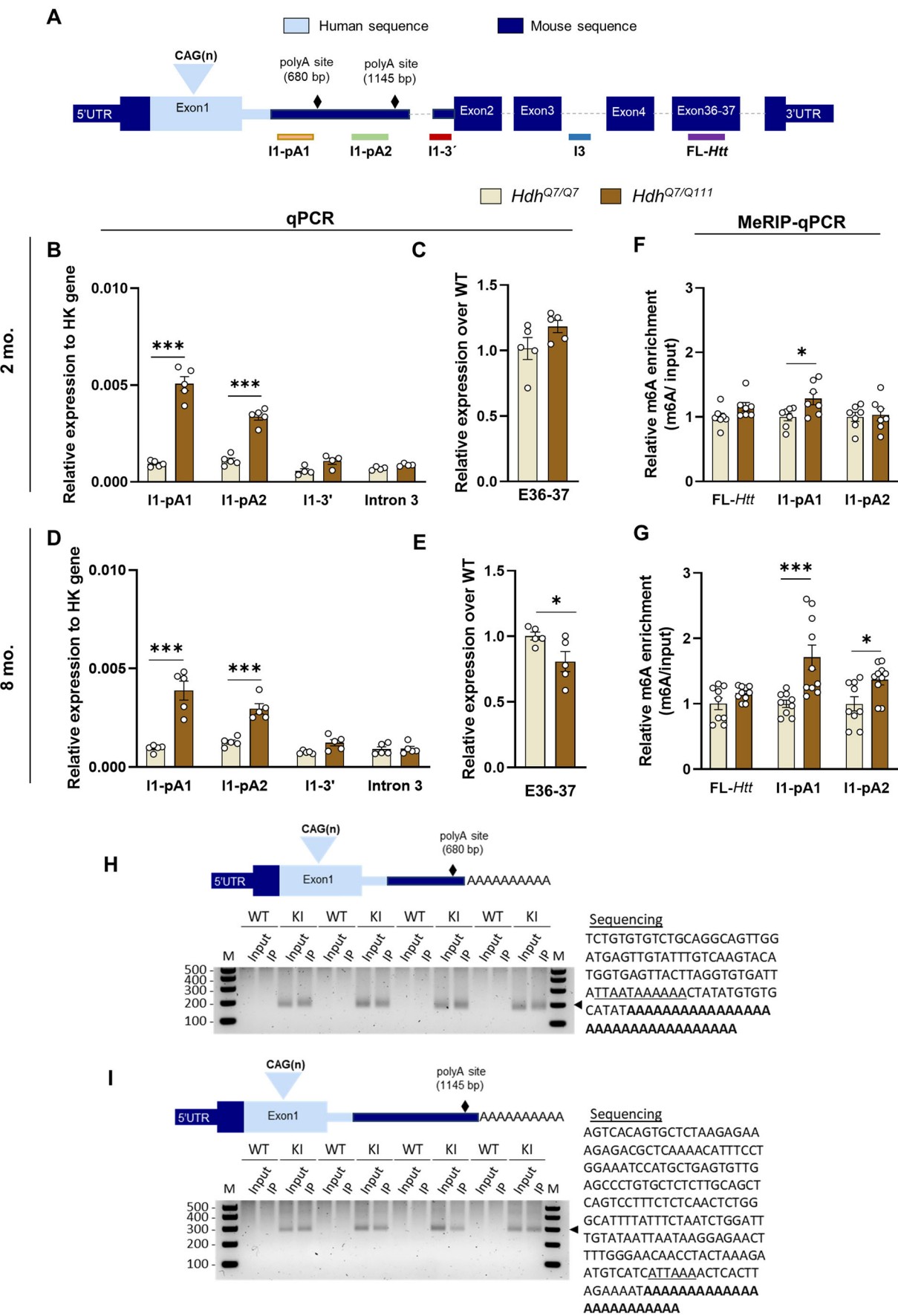

◄

**Figure 1. m⁶A methylation levels of *Htt1a* transcripts are increased in the *Hdh*^+/Q111 mouse model.**

(A) Schematic of the location of the primer-probe sets used for qPCR amplification of *Htt* transcripts. (B–E) qPCR analysis of *Htt* transcripts in the striatum of (B, C) 2-month-old ($n = 4$–5/genotype) and (D, E) 8-month-old *Hdh*^+/Q111 mice ($n = 5$/genotype). The expression of intronic sequences (I1-pA1, I1-pA2, I3 and I1-3') is presented relative to the housekeeping gene (B, D), and the relative levels of FL-*Htt* are shown relative to WT (C, E). Data represent the mean ± SEM. Data were analyzed using Student's two-tailed *t* test, (B) I1-pA1, ***$P < 0.0001$; I1-pA2, ***$P < 0.0001$; comparison between I3 and I1-3' is not significant, (C) FL-*Htt*, $P$ = n.s., (D) I1-pA1, ***$P = 0.0003$; I1-pA2, ***$P = 0.0003$; comparison between I3 and I1-3' is not significant (E), FL-*Htt*, *$P = 0.0456$. (F, G) Analysis of m⁶A enrichment was measured by MeRIP-qPCR in the striatum of (F) 2-month-old ($n = 7$/genotype) and (G) 8-month-old *Hdh*^+/Q111 mice ($n = 9$–10/genotype). Enrichment of m⁶A was normalized to input. Data represent the mean ± SEM. Data were analyzed using Student's two-tailed *t* test (F) *$P = 0.0256$ (G) I1-pA1, ***$P = 0.004$; I1-pA2, *$P = 0.0136$. FL: full-length; I: intron; pA: polyA site. (H, I) 3'RACE product in striatal samples generated from the cryptic poly(A) site at 680 bp (H) and 1145 bp (I) into intron 1 of *Htt*. ($n = 4$/genotype). M: DNA ladder. Right panel: SANGER sequencing of the generated product. The cryptic polyadenylation signal is underlined, and the poly(A) tail is shown in bold. The sequence was obtained from MeRIP and input samples ($n = 1$ mouse for position 680 bp; $n = 2$ mice for position 1145 bp). Source data are available online for this figure.

*Htt1*a since no significant differences were observed in FL-*Htt* (Appendix Fig. S2C). These results suggest that transcripts produced by aberrant splicing are enriched in m⁶A.

To confirm the m⁶A methylation in polyadenylated *Htt1a* transcripts generated by the cryptic poly(A) sites at 680 bp and 1145 bp, we performed 3'RACE on input and the MeRIP RNA derived from striatal samples of 8-month-old WT and *Hdh*^+/Q111 mice (Fig. 1H,I). Analysis of the input confirmed the presence of polyadenylated *Htt1a* generated by the two described cryptic poly(A) sites exclusively in *Hdh*^+/Q111 mice. Interestingly, when examining the MeRIP fraction, all *Hdh*^+/Q111 samples generated a 3'RACE product, indicating that polyadenylated mRNAs generated by aberrant splicing of *mHtt* were indeed m⁶A-modified.

## Aberrant m⁶A methylation at specific DRACH consensus motifs is conserved in human HD samples

Deposition of m⁶A mainly takes place at the DRACH consensus motif, where D=A, G or U; R= G or A; and H=A, C or U (Schwartz et al, 2014; Meyer et al, 2012; Dominissini et al, 2012). Accurate identification of m⁶A sites in specific mRNAs is invaluable for better understanding their biological functions. Therefore, we performed m⁶A mapping of *Htt1a* in polyA enriched RNA from the striatum of 8- months-old *Hdh*^+/Q111 mice using Nanopore direct RNA sequencing (DRS) which allows effective single-read detection of m⁶A RNA modifications (Liu et al, 2019). We analyzed two regions in the chimeric sequence of *Htt1a* (Fig. 2A), the human sequence in the first 267 bp of *Htt* intron 1 and the mouse sequence in the subsequent 563 bp where we have previously detected an m⁶A enriched peak via MeRIP-seq (Fig. EV2A,B) (Pupak et al, 2022). By aligning to the mutated sequence single-read m⁶A modification predictions identified 10 sites upstream the 1st cryptic Poly(A) site in the mouse sequence of *Htt1a* (Fig. 2A) validating our previous results obtained with MeRIP-seq. Interestingly, the analysis also predicted 4 sites in the human sequence of the chimeric allele. Mapping of both short mutant *Htt* polyadenylated transcripts is shown in Fig. EV2C. As expected, no coverage for these transcripts was observed in *Hdh*^Q7/Q7 mice.

Next, based on the prediction of m⁶A modifications by DRS we interrogated selected m⁶A sites in *Htt* intron 1 from different HD cell lines. Methylation levels were analyzed by MazF-qPCR, an approach that relies on the ability of the bacterial RNase MazF to cleave RNA at unmethylated sites occurring at ACA motifs but not at the methylated counterparts m⁶A-CA (Fig. EV2D) (Garcia-Campos et al, 2019). Three m⁶A sites containing ACA motifs were chosen to be located within this first 523 bp of *Htt* intron 1: the

human (hm) GGACA site and the two mouse (ms) m⁶A sites, AGACA and GGACA (Fig. 2B). We designed different sets of primers to amplify the regions containing these m⁶A consensus motifs as well as a set of primers designed to flank an adjacent region in the same gene that did not harbor an ACA site to serve as a control probe. We analyzed methylation levels in *Htt/HTT* intron 1 RNA in three different HD mouse cell lines (Fig. 2B): ST*Hdh*^Q111/Q111 cells, which express the chimeric human/mouse (hm/ms) mutant *Htt* gene that contains 267 bp of human intron 1; embryonic mouse fibroblasts (MEFs) from zQ175 mice, which express mutant *Htt* in which 84 bp of the 5′ end of mouse intron 1 have been replaced with 10 bp from the 5′ end of human intron 1 (Mason et al, 2020); and MEFs from YAC128 mice, which express human mutant *HTT* (Fienko et al, 2022). MazF-qPCR analysis in ST*Hdh*^Q111/Q111 cells revealed an increased methylation ratio in GGACAhm and AGACAms sites compared to the GGACAms site located further downstream in mouse intron 1 (Fig. 2C). Similar results were observed in the striatal samples from *Hdh*^+/Q111 mice where methylation ratio is increased in the AGACAms site compared with Wt mice (note that GGACAhm motif is only present in *Hdh*^+/Q111 mice, Fig. EV2E). On the other hand, we observed that in MEF zQ175 cells, the AGACA mouse motif appears more methylated than the downstream GGACA mouse site (Fig. 2D). In YAC128 MEFs, we also found significant differences between the different sites analyzed, with the highest methylation ratio observed in the GGACA site (Fig. 2E), as we also observed in the ST*Hdh*^Q111/Q111 cells (Fig. 2C). These data suggest that the single peak of m⁶A in intron 1 detected in our MeRIP-seq analysis (Pupak et al, 2022) reflects the overall levels of methylation of more than one m⁶A site. Moreover, they indicate that methylation occurs at different rates in different coexisting m⁶A motifs in accordance with the observations of the predicted m⁶A sites detected in this region by DRS.

Next, we further evaluated the pathological relevance of m⁶A methylation in the 5' region of *mHtt* intron 1 by analysis of the methylation ratio in HD post-mortem samples and HD skin-derived fibroblasts. We focused on the human GGACA site (Fig. 3A) since it showed high confidence prediction in DRS and was highly methylated in ST*Hdh*^Q111/Q111 and YAC128 MEF cells expressing the human intronic sequence (Fig. 2C,D). A trend for a gradual increase in m*HTT* intron 1 methylation at the GGACA site was detected along the progression of the neuropathology, showing significant differences in post-mortem samples with Vonsattel grades 2–3 and 3 when compared to controls (Fig. 3B). MazF-qPCR analysis in HD skin fibroblasts showed significant increased methylation levels in Pre-HD adult as well as in S-HD adult and

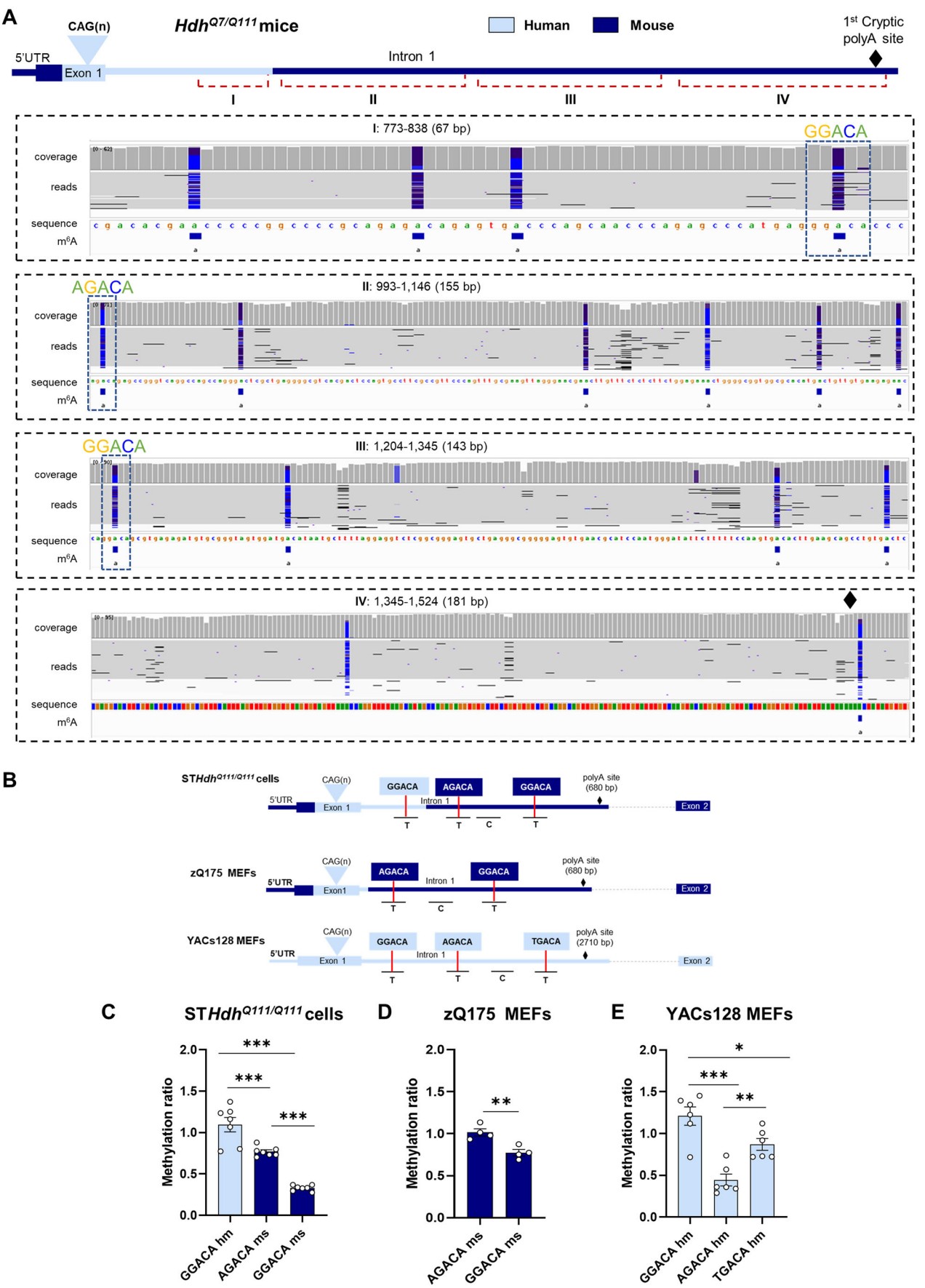

**Figure 2. Mapping of m⁶A sites in the proximal region of m*Htt* intron 1.**

(A) Analysis of m⁶A sites in *Htt1a* by Nanopore direct RNA sequencing in PolyA enriched RNA from striatum of $Hdh^{+/QIII}$ mice ($n = 2$ replicates, each replicate is a pool of 3–4 mice). A mouse sequence (GRCmm39 genome) *Htt* gene including the human insert was used as reference gene (chr5: 34,919,088–35,070,342). The intronic region analyzed starts at 686 bp from the 5′ UTR and ends at 1819 bp (*chr5*: 34,919,774–34,920,048). IGV snapshots show the regions were m⁶A sites were predicted. Purple and blue dots represent high and low confidence m⁶A sites, respectively. Mapping of m⁶A sites in the whole *Htt1a* transcripts is shown in EV2C. Methylation sites analyzed by MazF-qPCR are highlighted in the tracks of the IGV snapshot. (B) Schematic of m*Htt* intron 1 m⁶A motifs analyzed by a qPCR-based assay coupled with MazF digestion in HD cell models, ST*Hdh*$^{QIII/QIII}$, zQ175 MEFs and YAC128 MEFs. (C–E) Methylation ratio obtained by MazF-qPCR analysis in (C) ST*Hdh*$^{QIII/QIII}$ cells ($n = 7$ technical replicates/motif), (D) zQ175 MEFs ($n = 4$ technical replicates/motif) and (E) YAC128 MEFs ($n = 6$ technical replicates/motif). The levels of a targeted amplicon (labeled "T") are measured against a control (labeled "C") amplicon in a MazF-digested sample and normalized against a nondigested sample. Data represent the mean ± SEM. Data were analyzed using one-way ANOVA with Tukey´s multiple comparisons test (C) ***$P < 0.0001$, (D) **$P < 0.0057$, (E) *$P = 0.0342$, **$P = 0.0082$, ***$P < 0.0001$. hm human (light blue), ms mouse (dark blue). Source data are available online for this figure.

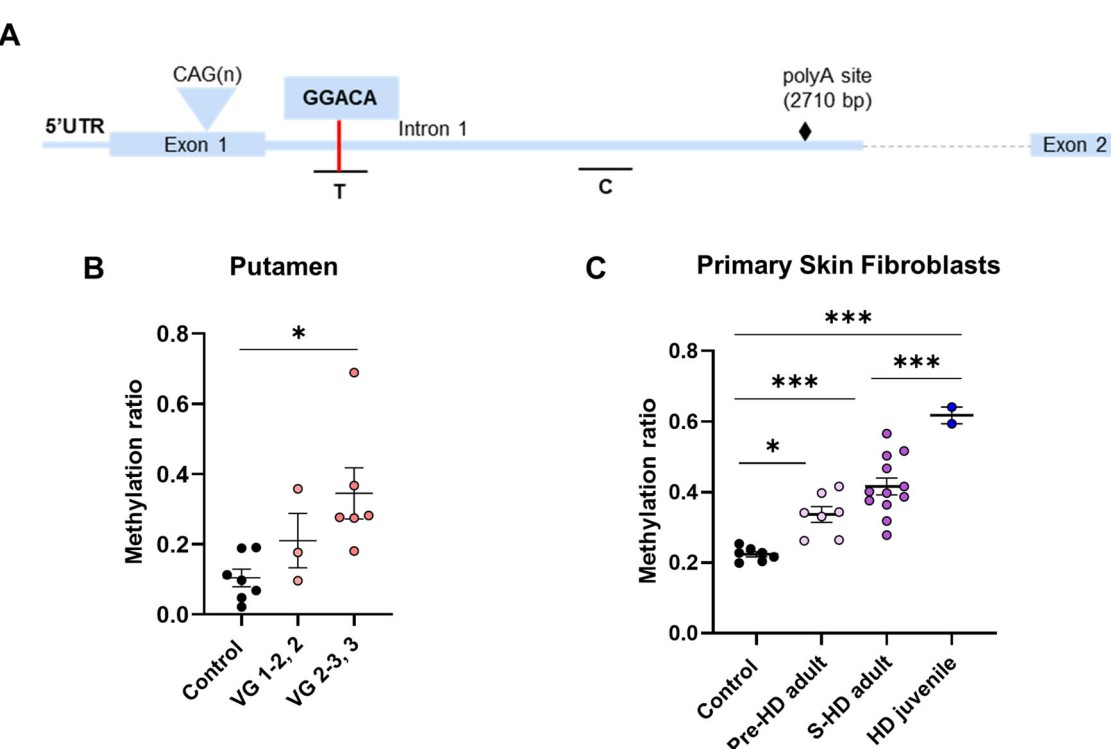

**Figure 3. Increased methylation levels are detected at the m⁶A GGACA motif of *Htt* intron 1 in human samples.**

(A) Schematic of the DRACH motif GGACA hm in *HTT* intron 1 analyzed by a qPCR-based assay coupled with MazF digestion in human samples. (B, C) Methylation ratio obtained by MazF-qPCR analysis in (B) human *postmortem* samples of the putamen ($n = 3$–7 individuals/group) and (C) in human skin fibroblasts ($n = 2$–12 patients/ group; HD adult: Q40-Q56; HD juv: Q80 and Q180). VG: Vonsattel grade. VG1-2 (samples were from individuals with Vonsattel grades ranging between 1 and 2) VG 2-3 (samples were from individuals with Vonsattel grades ranging between 2 and 3). Pre-HD adult: pre-symptomatic HD adult; S-HD adult: symptomatic HD adult; HD Juv: HD Juvenile. The levels of a targeted amplicon (labeled "T") are measured against a control (labeled "C") amplicon in a MazF-digested sample and normalized against a nondigested sample. Data represent the mean ± SEM. Data were analyzed using one-way ANOVA with Tukey´s multiple comparisons test. (B) *$P = 0.0141$; (C) *$P = 0.0169$, **$P = 0.0024$; ***$P < 0.0001$. Source data are available online for this figure.

HD Juvenile fibroblasts compared to controls (Fig. 3C). Although there is a noticeable trend for methylation increase with disease progression from pre-symptomatic to moderate/advance symptomatic stages ($P = 0.07$), no statistically significant correlation was observed (Appendix Fig. S3A). Interestingly, a trend for methylation to increase with CAG length was also observed in HD skin fibroblasts samples (Appendix Fig. S3B). However, this correlation was only significant ($p = 0.01$) when the two HD juvenile were included in the analysis. Notably, the increased methylation observed in HD human samples is not accompanied by significant increased levels of *HTT1a* (Appendix Fig. S4A,B), except in HD fibroblast samples with CAG repeat lengths in the juvenile-onset range. This confirmed previous findings that the generation of this short *HTT* transcript increases with longer CAG tracts, occurring with 60 or more CAGs (Neueder et al, 2017; Hoschek et al, 2024). These data suggest that in these samples m⁶A methylation is primarily detected by MazF-qPCR on the nascent RNA where it might play a role in the generation of *HTT1a* when longer CAG repeats are present in the mutant *HTT* gene, as observed in HD mice and cell lines. Overall, these findings indicate the potential pathological relevance of m⁶A methylation in m*HTT* RNA for HD patients.

## METTL3 modulates *Htt* intron 1 methylation and reduces the expression levels of *Htt1a* transcripts

The METTL3/14 writer complex regulates the deposition of m$^6$A in both intronic and exonic regions (Wei et al, 2021), modulating several aspects of the mRNA lifecycle, including splicing (Adhikari et al, 2016; Louloupi et al, 2018). To analyze the potential role of METTL3 in the methylation of *Htt* intron 1 and the expression of *Htt* transcripts, we inhibited METTL3 with STM2457, a novel METTL3-specific inhibitor that can bind to the S-adenosyl-L-methionine (SAM) binding site (Yankova et al, 2021). ST*Hdh* cells were treated with 10 μM and 20 μM of STM2547 for 48 h and a global reduction of m$^6$A levels in total RNA was confirmed through the colorimetric EpiQuik assay (Appendix Fig. S5A). We further analyzed by MazF-qPCR the methylation ratio of the different methylation sites in *Htt* intron 1 described in Figs. 2 and 3. This analysis revealed a significant decrease of m$^6$A levels at the GGACA site within the human region of chimeric m*Htt* intron 1 in ST*Hdh*$^{Q111/Q111}$ cells following METTL3 inhibition (Fig. 4A). When analyzing by qPCR the levels of *Htt1a* intronic sequences upstream of the first cryptic polyA site (I1-pA1) and FL-*Htt*, a significant reduction in I1-pA,1 but not in FL-*Htt*, was observed in ST*Hdh*$^{Q111/Q111}$ cells treated with both STM2457 concentrations compared to the vehicle (Fig. 4B). No changes were detected in the relative levels of intron 3 (Appendix Fig. S5B) or in the levels of FL-*Htt* in ST*Hdh*$^{Q7/Q7}$ cells treated with STM2457 (Appendix Fig. S5C). Inhibition of METTL3 in YAC128 MEFs with the highest concentration also decreased the ratio of methylation at the GGACA site without affecting the other motifs analyzed (Fig. 4C). No significant differences were observed with 10 μM, suggesting that this concentration may not be sufficient to fully inhibit METTL3 activity on *Htt1a* transcripts in these MEFs cells which have a different cellular context than the ST*Hdh*$^{Q111/Q111}$. Moreover, these cells express human *Htt1a* transcripts which might be more resistant to methylation changes due to their sequence or structure. Notably, a significant decrease in *HTT1*a but not FL-*HTT* transcripts was observed at both concentrations (Fig. 4D) indicating that METTL3 inhibition by STM2457 may exert indirect effects on the transcriptional machinery or splicing factors that influence the generation of *HTT1a* transcripts independently of direct methylation changes at GGACA or other methylation sites. We also evaluated the impact of METTL3 inhibition in zQ175 MEFs, which lack the human intronic region with the GGACA human (GGACAhm) motif. Consistent with our observations in ST*Hdh*$^{Q111/Q111}$ and YAC128 MEFs cells we did not find significant changes in the methylation levels when murine motifs (AGACAms and GGACAms) were analyzed (Fig. 4E). *Htt1a* levels were also downregulated by STM2457, albeit to comparatively lower levels than in ST*Hdh*$^{Q111/Q111}$ and YAC128 MEFs cells (Fig. 4F). These results ensured the reliability of the human GGACA motif identified by DRS and validated by MazF-qPCR method and indicate that the methylation of this motif is dependent on METTL3 activity. Importantly, the observed decrease in m$^6$A levels is not due to a reduction in mRNA levels but rather a true reduction in the methylation status, as the approach uses a formula where methylated I1-pA1 transcripts are normalized against total levels of these intronic sequences (non-ACA regions) (Fig. EV2D).

While the STM2457 is a specific small molecule inhibitor of METTL3 with no evidence of off-target effects (Yankova et al, 2021), we further validated the specific involvement of METTL3 by siRNA knockdown (Fig. EV3). METTL3 mRNA (Fig. EV3A) and protein expression levels (Fig. EV3B,C) were downregulated after transfection with different siRNAs against METTL3 and global m$^6$A levels in total RNA were reduced (Fig. EV3D), although to lesser extent compared to the STM2457 inhibitor (Appendix Fig. S5A). Importantly, the results are consistent with previously observed effects of STM2457 on m$^6$A methylation in the GGACA site (Fig. EV3E) and downregulation of I1-pA1 transcripts expression (Fig. EV3F) suggesting a critical role for METTL3 and m$^6$A modifications in *Htt/HTT* RNA metabolism.

## Demethylation of intron 1 using CRISPR/dCas13b fused to Alkbh5 downregulates the expression of *Htt1a* transcripts

To elucidate causal relationships between the specific presence of m$^6$A in intron 1 and the downregulation of *Htt1a*, we conducted targeted demethylation of *Htt* intron 1 in ST*Hdh*$^{Q111/Q111}$ cells using a CRISPR–Cas13-based approach. (Fig. 5A). We designed plasmid constructs that expresses the dCas13b alone and the dCas13b fused to ALKBH5 or the catalytically inactive mutant of ALKBH5 (H204A) through a link with the C-terminus of inactive Cas13b (catalytically dead type VI-B Cas 13 enzyme named dPspCas13b in Cox et al (Cox et al, 2017)) (Fig. 5A). To site-specific manipulate m$^6$A in *Htt* intron 1, ST*Hdh* cells were stably transfected with the different constructs expressing the fusion protein and non-targeting gRNA (NT-gRNA) or gRNAs. The three gRNAs were designed to target three distinct positions (gRNA1,2 and 3) located inside intron 1, around three m$^6$A sites identified by DRS and previously analyzed by MazF-qPCR and upstream of the first polyA site (Appendix Fig. S6). These gRNAs were designed to be 30 nt long and 100–300 nt away from the methylated sites to enhance demethylation efficiency of the system according to the characterization of a CRISPR–Cas13b-based tool for targeted demethylation of specific RNA described by Li et al (Li et al, 2020).

First, we confirmed the efficiency of transfection analyzing ALKBH5 expression by immunohistochemistry as shown in Fig. 5b. We also analyzed by western blot the weight shift of ALKBH5, indicating successful transfection of the cells (the fusion protein weighs approximately 150 kDa, which corresponds to the sum of dCas13b (124 kDa) and ALKBH5 (44 kDa) (Appendix Fig. S7A). We then verified the effect of our RNA editing system by evaluating the methylation levels of m$^6$A sites. MazF-qPCR showed that all three dCas13b-ALKBH5 systems expressing the different gRNAs significantly decreased the methylation ratio of the targeted GGACA hm site compared with the control dCas13b-ALKBH5 NT-gRNA, while no changes were observed in the other two motifs (Fig. 5C). The strongest and most significant demethylation was observed with gRNA2, which targets a 200 nt downstream region from the m$^6$A GGACA hm site, resulting in 30–50% demethylation (Fig. 5C). Similarly, to reported evidence using dCas13b-ALKBH5 (Li et al, 2020) RNA demethylation of *Htt* intron 1 seems not to be influenced by either the 5′ or 3′ sequence of the dCas13b-targeted site, but it may be dependent on space between dCas13b-targeted and m$^6$A-methylated sites.

The analysis of the expression levels of the different *Htt* transcript isoforms showed that transfection with gRNA-2 and -3 significantly decreased the expression of transcripts generated by the first and second cryptic poly(A) sites, while no differences were detected in FL-*Htt* levels (Fig. 5D). Next, we analyzed stably transfected cells with dCas13b-gRNA2 fused to H204A (H204-gRNA2). Consistent with the previous results, stable transfection of ST*Hdh*$^{Q111/Q111}$ cells with the dCas13b-ALKBH5 gRNA2

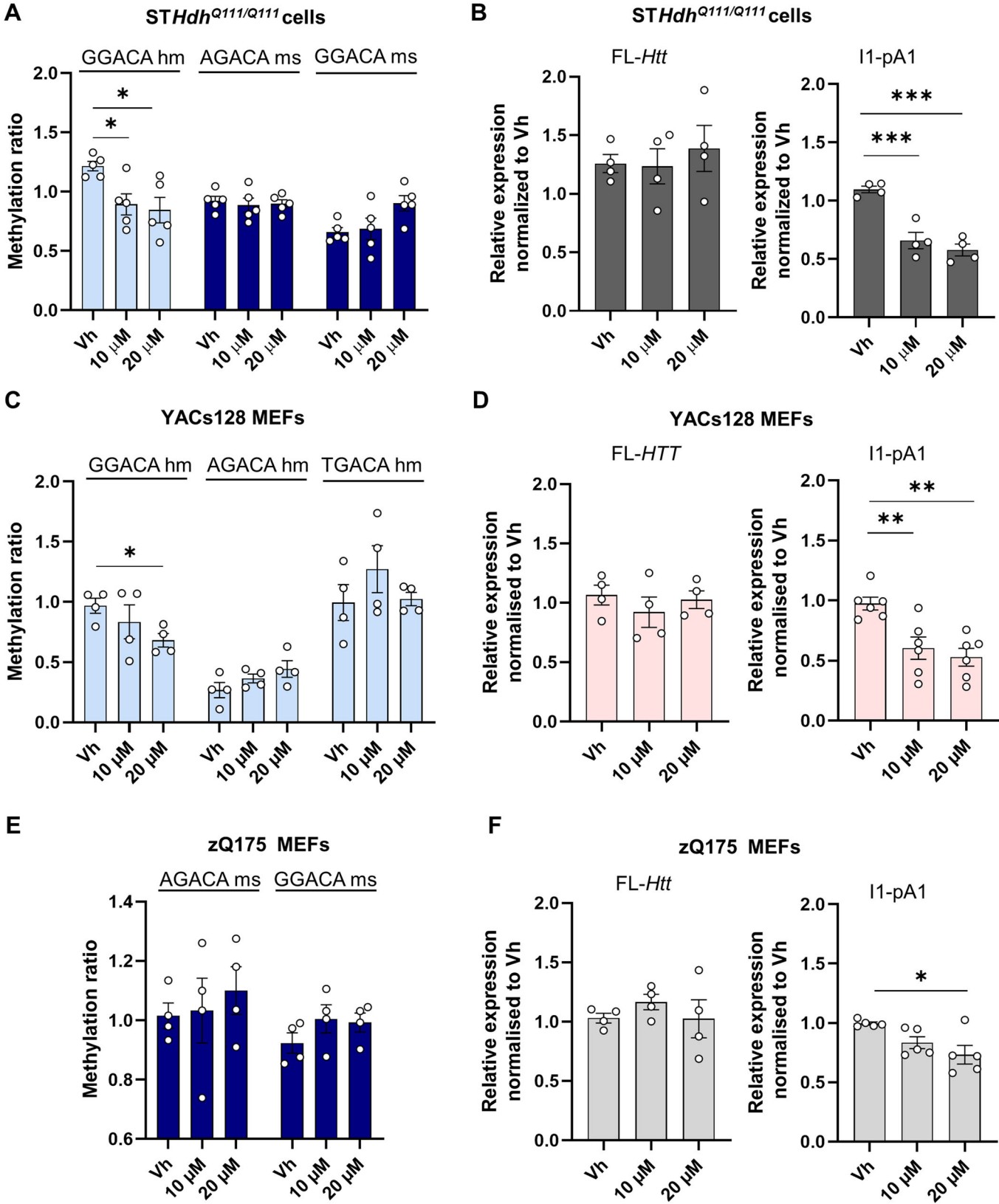

◄

**Figure 4.** Pharmacological inhibition of METTL3 by STM2457 regulates the expression of *Htt1a* in HD in vitro models.

ST*Hdh*$^{QIII/QIII}$ cells (**A, B**), YAC128 cells (**C, D**), and zQ175 MEFs (**E, F**) were treated with DMSO (vehicle (Vh)), 10 μM STM2457, or 20 μM STM2457 for 48 h. (**A, C, E**) Methylation ratio obtained by MazF-qPCR analysis of *Htt* intron 1 in (**A**) ST*Hdh*$^{QIII/QIII}$ cells ($n = 5$ independent experiments; 2–4 technical replicates/experiment), (**C**) YAC128 ($n = 4$ independent experiments); and (**E**) zQ175 MEFs ($n = 4$ independent experiments) Data in (**A, C, E**) represent the mean ± SEM. Data were analyzed using one-way ANOVA with Tukey´s multiple comparisons test. (**A**) *$P = 0.0470$, 10 μM STM2457 compared to Vh; *$P = 0.0230$, 20 μM STM2457 compared to Vh; (**C**) *$P < 0.0145$. (**B, D, F**) qPCR analysis of *Htt* transcripts (FL-*Htt* and I1-pA1) in (**B**) ST*Hdh*$^{QIII/QIII}$ cells, ($n = 4$ independent experiments; 2 technical replicates/experiment) (**D**) YAC128 cells ($n = 4$–6 independent experiments; 2–3 technical replicates/experiment) and (**F**) MEFs ($n = 4$–6 independent experiments; 2–3 technical replicates/experiment). Data in (**B, D, F**) represent the mean ± SEM. Data were analyzed using one-way ANOVA with Tukey´s multiple comparisons test (**B**) ***$P = 0.0006$, 10 μM STM2457 compared to Vh; ***$P = 0.0002$, 20 μM STM2457 compared to Vh; (**D**) **$P = 0.009$, 10 μM STM2457 compared to Vh; **$P = 0.002$, 20 μM STM2457 compared to Vh; (**F**) *$P = 0.013$, 20 μM STM2457 compared to Vh. Source data are available online for this figure.

(A5-gRNA2) showed a significant reduction of approximately 20% of the *Htt1a* transcripts generated by the first and second cryptic poly(A) sites while no differences were found in cells transfected with the catalytically inactive H204A enzyme (Fig. 5E). In addition, confirming previous observations, no significant changes were observed in FL-*Htt* transcripts (Fig. 5E) or when testing the probes against I1-3′ or the control I3 (Appendix Fig. S7B). These results indicate that demethylation of *Htt* intron 1 RNA was achieved specifically using the RNA editing ALKBH5 constructs further supporting a role of m⁶A in the expression of *Htt1a* transcripts.

To investigate whether dCas13b-ALKBH5-induced downregulation of *Htt1a* transcripts was caused by m⁶A-mediated mRNA decay, we performed RNA lifetime profiling by collecting and analyzing RNA from targeted demethylated and control samples obtained at different time points after transcription inhibition with actinomycin D (ActD) (Fig. 5F). RNA stability assays showed that targeted demethylation of m*Htt* intron 1 with dCas13b-A5 gRNA2 did not change the RNA half-life, displaying comparable RNA decay levels with control stably transfected cells (dCas13b-A5-NTgRNA and dCas13b-gRNA2) when analyzing both *Htt1a* and FL-*Htt* (Fig. 5F). These results indicate that demethylation in m*Htt* intron 1 does not influence the stability of *Htt1a* but rather affects other mechanisms, such as aberrant splicing, thus regulating the production of *Htt1a* transcripts.

Given the evidence that persistent damage in neuronal DNA contributes to early HD pathogenesis (Pradhan et al, 2022), we evaluated the impact of targeted *Htt1a* intron demethylation on basal DNA damage in stably transfected cells. To this aim, we analyzed the phosphorylation of histone H2AX at Serine 139 (gamma-H2AX), a marker of DNA double-strand breaks, by immunocytochemistry. We detected a significant decrease in the percentage of nuclei with γH2AX foci and in the number of γH2AX foci per cell in cells stably transfected with dCas13b-A5-gRNA2 compared with control transfected cells (dCas13b-A5-NTgRNA and dCas13b-gRNA2) (Fig. 5G). Since DNA damage is associated with low levels of ATP (Ayala-Peña, 2013) and the ST*Hdh*$^{Q111/Q111}$ are known to display an ATP deficit (Gines, 2003; Lim et al, 2008) we also measured ATP levels in the stably transfected cells. Our results show a significant increase in ATP levels in ST*Hdh*$^{Q111/Q111}$ expressing dCas13b-A5-gRNA2 compared with control transfected cells (Fig. 5H). Together, our results show that m⁶A methylation in *Htt* intron 1 is involved in the incomplete splicing that generates *Htt1a* and this regulation might influence the DNA damage response and DNA repair mechanisms. It remains to be clarified to what extent DNA damage is associated with altered levels of *HTT1a*.

## CAG repeat expansion regulates methylation in *Htt* intron 1 and affects the expression of *Htt1a*

Context-dependent features and RNA secondary structure play a key role in determining m⁶A deposition (Shachar et al, 2024). Therefore, we explored whether CAG-trinucleotide repeats, which are known to form RNA stable hairpin structures with protein binding properties (Krzyzosiak et al, 2012; Jasinska, 2003; Galka-Marciniak et al, 2012), could contribute to m⁶A deposition in the proximal region of *Htt* intron 1. We used locked nucleic acid–modified antisense oligonucleotides complementary to the CAG repeat (LNA-CTG) that preferentially bind to mutant *Htt* to block CAG expansions (Rué et al, 2016). The LNA-CTG binding to *Htt* RNA was confirmed by the absence of PCR amplification within the LNA-bound region due to the strong incompatibility of LNA-CTG:CAG duplexes with retrotranscription and subsequent PCR amplification (Fig. 6A,B). We transfected ST*Hdh*$^{Q111/Q111}$ cells with different concentrations of LNA-CTG or the analogous scrambled control LNA-ASO (LNA-SCB) and monitored LNA-CTG binding to the CAG repeat at exon 1 (Fig. 6B) as well as *Htt* transcript expression (FL-*Htt* and *Htt1a*) (Fig. 6C). As previously demonstrated (Rué et al, 2016), the lack of *HTT* exon 1 mRNA amplification with primers spanning the CAG repeat (HTT-e1*) supports LNA-CTG binding to the expanded transgene (Fig. 6B). PCR with primers amplifying Exon 1 outside the CAG repeat (HTT-e1) revealed no changes in *Htt* RNA levels in accordance with published data (Fig. 6B) (Rué et al, 2016). We performed qPCR to analyze *Htt* transcripts and we detected an LNA-CTG dose-dependent decrease in the levels of *Htt1a* by qPCR, while no changes in FL-*Htt* were observed (Fig. 6C). These data suggest that blocking expanded CAG repeats with LNA-CTG ASOs specifically affects the production of *Htt1a*, favoring the concept that changes in expanded CAG structure and activity are crucial to produce *Htt1a* pathogenic species.

Next, using the LNA-CTG ASOs, we determined the effect of blocking the activity of the CAG repeat expansion on methylation status of *Htt* intron 1 at the selected motifs previously described. The methylation ratio obtained with MazF-qPCR analysis revealed a significant decrease in m⁶A abundance in the GGACA hm motif of *Htt* intron 1 present in the ST*Hdh*$^{Q111/Q111}$ cells (Fig. 6D). Furthermore, using RNA immunoprecipitation assay RIP-qPCR (Fig. 6E) we found that a direct interaction between METTL3 and I1-pA1 or I1-pA2 sequences is significantly enhanced in ST*Hdh*$^{Q111/Q111}$ cells compared with that in ST*Hdh*$^{Q7/Q7}$ cells (Fig. 6F,G) while no significant binding was found in FL-*Htt* transcripts (Fig. 6H). Blocking CAGs with LNA-CTGs ASOs decrease significantly METTL3 interaction with I1-pA1 and I2-pA2 RNA transcripts (Fig. 6F,G). These results suggest that METTL3 recruitment and methylation of this specific site in intron 1

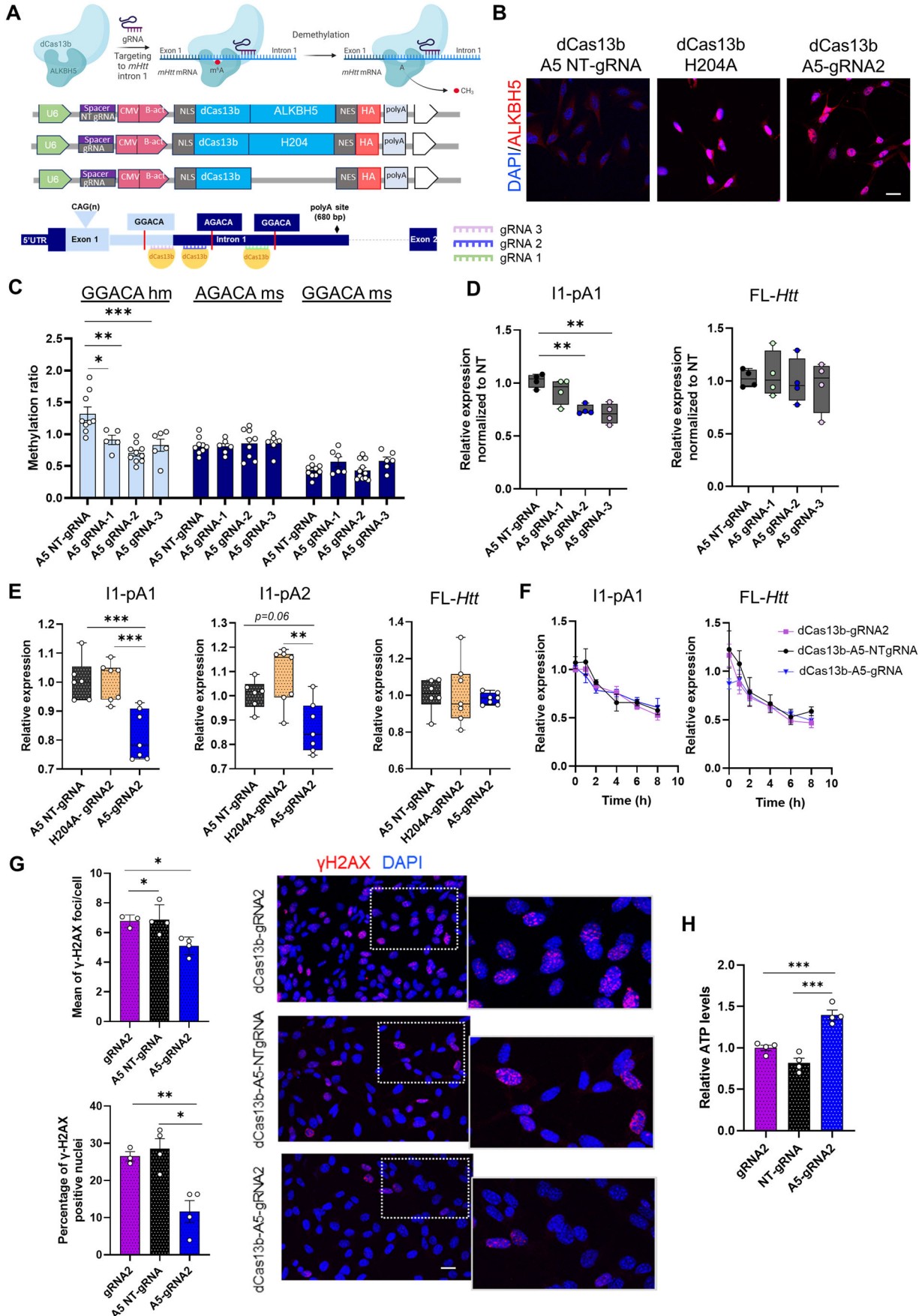

**Figure 5. Target demethylation of *Htt* intron 1 regulates the expression of *Htt1a* in ST*Hdh*^Q111/Q111 cells.**

(A) Schematic representation of CRISPR dCas13b plasmid constructs and positions of the m⁶A site within *Htt* intron 1 mRNA and regions targeted by three different gRNAs. (B) Representative images from immunofluorescence staining of ALKBH5 in transfected ST*Hdh*^Q111/Q111 cells with dCas13b (control, inactive H204A and ALKBH5). Nuclei are stained with DAPI (blue). Scale bar, 20 μm. (C) MazF-qPCR analysis of the DRACH motifs in *Htt* intron 1 in stably transfected cells ($n = 6$–9 technical replicates from 3 to 4 independent experiments). Data represent the mean ± SEM. Data were analyzed using one-way ANOVA with Tukey´s multiple comparisons test. *$P = 0.0185$, **$P = 0.0022$, ***$P < 0.0001$ compared to NT-gRNA. (D) qPCR analysis of the expression levels of I1-pA1 and FL-*Htt* transcripts in ST*Hdh*^Q111/Q111 cells transfected with dCas13b-ALKBH5 (A5) combined with NT-gRNA (control) or gRNA 1, 2 and 3 ($n = 4$ independent experiments; 3–4 technical replicates/experiment). Data represent the mean ± SEM. Data were analyzed using one-way ANOVA with Tukey´s multiple comparisons test. I1-pA1: **$P = 0.0036$ (A5-gRNA3 vs NT-gRNA) and **$P = 0.0014$ (A5-gRNA2 vs NT-gRNA). (E) Expression levels of *Htt* transcripts (I1-pA1, I1-pA2, FL-*Htt*) in ST*Hdh*^Q111/Q111 cells transfected with dCas13b-NT-gRNA, dCas13b-H204A-gRNA2 and dCas13b-A5-gRNA2 ($n = 6$–7 independent experiments; 2 technical replicates/ experiment). Box plot representing the distribution of relative expression normalized to NT-gRNA for the transfection with different constructs. The box extends from the first quartile to the third quartile, with a horizontal line indicating the median. The whiskers extend to the minimum and maximum values. Data were analyzed using one-way ANOVA with Tukey´s multiple comparisons test. I1-pA1: ***$P = 0.0007$ (A5-gRNA2 vs H204A-gRNA2), I1-pA2: **$P = 0.0008$ (A5-gRNA2 vs. NT-gRNA2); I1-pA2, **$P = 0.0025$ (A5-gRNA2 vs H204A-gRNA2). (F) RNA decay profile of *Htt* transcripts in transfected ST*Hdh*^Q111/Q111 cells treated with actinomycin-D (Act-D) for the indicated times. Data represent the mean ± SEM ($n = 4$ independent experiments). Data were analyzed using one-way ANOVA with Tukey´s multiple comparisons test. (G) Histograms showing the percentage of nuclei with γ-H2AX foci and the average of γ-H2AX foci per cell in transfected ST*Hdh*^Q111/Q111 cells with dCas13b-ALKBH5, dCas13b-NT gRNA or dCas13b-A5 gRNA2. Data represent the mean ± SEM. Data were analyzed using Student's *t* test (10–15 images from 3 to 4 independent experiments). Percentage of nuclei with γ-H2AX foci: *$P = 0.04$ (gRNA2 vs A5-gRNA2), *$P = 0.02$ (NT-gRNA vs. A5-gRNA2); average of γ-H2AX foci per cell: *$P = 0.018$ (gRNA2 vs A5-gRNA2), **$P = 0.0036$ (NT-gRNA vs. A5-gRNA2). Representative images of γ-H2AX foci (red) in ST*Hdh*^Q111/Q111 cells. Nuclei are stained with DAPI (blue). Scale bar, 20 μm. (H) Histogram showing relative measurement of ATP in ST*Hdh*^Q111/Q111 stably transfected cells. ATP was assessed with Cell Titer Glo (Promega). ATP measurements were normalized to DNA concentration. Data represent the mean ± SEM ($n = 4$ independent experiments; three technical replicates /experiment). Data were analyzed using one-way ANOVA with Tukey´s multiple comparisons test. ***$P < 0.001$. Source data are available online for this figure.

of m*Htt* RNA is directly affected by the CAG expansion, supporting the idea that expanded CAG plays a mechanistic role in the aberrant splicing of *Htt* RNA via regulation of m⁶A levels in intron 1.

## Discussion

Our study reveals that m⁶A methylation in intron 1 of m*Htt* RNA contributes to the generation of the aberrantly spliced mRNA variant *Htt1a*. Building upon our previous evidence showing that *Htt* transcripts in the hippocampus of *Hdh*^+/Q111 mice are significantly methylated (Pupak et al, 2022), our present study revealed m⁶A hypermethylation in *Htt* intron 1 in the striatum of the *Hdh*^+/Q111 mice and identified the m⁶A methylation sites in *Htt1a*. Importantly, we show that human samples are methylated in the same site that is methylated in the *Htt* intron 1 of *Hdh*^+/Q111 mice. Pharmacological inhibition, siRNA knockdown of METTL3 and targeted demethylation of this m*Htt* intronic region in HD mouse cells regulate the expression of *Htt1a*, suggesting that m⁶A contributes to the incomplete splicing of m*Htt*. This methylation is further influenced by CAG repeats. Collectively, these data reveal a new CAG-dependent mechanism involved in the aberrant processing of m*Htt* that relies on m⁶A RNA modification.

The modification m⁶A is typically installed cotranscriptionally by the writer complex, with enrichment near the translation stop codon and 3′ untranslated region (3′ UTR) (Dominissini et al, 2012; Meyer et al, 2012). However, recent studies have revealed that m⁶A can also be found at the 5′UTR, exons and introns of nascent transcripts (Louloupi et al, 2018; Ke et al, 2017; Hu et al, 2022; Wei et al, 2021), supporting earlier findings showing m⁶A on chromatin-associated nascent pre-mRNAs, including introns (Carroll et al, 1990; Salditt-Georgieff et al, 1976). Here, we demonstrate that m⁶A is enriched in intronic sequences upstream of cryptic polyA sites expressed in *Htt1a* in both the striatum of *Hdh*^+/Q111 mice and ST*Hdh*^Q111/Q111 cells. This aligns with our MeRIP-seq data from *Hdh*^+/Q111 mice hippocampus, showing significant m⁶A enrichment in the 5′ proximal region of *Htt* intron 1. Similar enrichment has been reported toward the 5′-end of introns,

particularly in regions involved in alternative splicing (Wei et al, 2021; Hu et al, 2022) suggesting a potential functional role of m⁶A in the alternative processing of *Htt* RNA.

Furthermore, we detected m⁶A enrichment in polyadenylated *Htt1a* mRNAs by 3′RACE PCR and identified the m⁶A modifications in the intronic region of *Htt1a* by direct RNA sequencing, indicating its persistence during RNA maturation and potential roles in *Htt1a* mRNA fate. Notably, *Htt1a* can be found in the nucleus in the form of RNA foci in YAC128 mice (Fienko et al, 2022) and in HD *postmortem* brains, likely due to somatic expansion of the CAG repeats (Ly et al, 2022). Although the direct function of m⁶A modifications in mRNAs stress granules partitioning in vivo is unclear, it has been proposed that m⁶A, particularly in longer mRNAs containing multiple heavily modified m⁶A sites, might contribute in stress granule recruitment (Khong et al, 2022). This raises the possibility that m⁶A methylation, via interactions with scaffold reader proteins (Ries et al, 2019; Gao et al, 2019b; Fu and Zhuang, 2020), could contribute to *Htt1a* accumulation in nuclear clusters. Supporting this, a recent study found that m⁶A is enriched in intronic polyadenylated transcripts and regulates the retention of mRNAs containing intact 5′ splice site motifs in nuclear foci (Eliza S. Lee et al, 2024).

m⁶A DRACH motifs are widespread throughout cellular transcriptomes, but only a small fraction has been reported to be significantly methylated in vivo (Meyer et al, 2012; Dominissini et al, 2012). Therefore, to cross-validate the known m⁶A sites within *Htt* intron 1, we used Nanopore direct RNA sequencing for m⁶A mapping in *Hdh*^+/Q111 mice alongside an ortholog method to quantify methylation status at specific sites in various HD mouse cell lines. We identified methylation at one motif (GGACA) located 147 nt from the exon 1-intron 1 splice junction in intronic human sequence of *Htt* in *Hdh*^+/Q111, ST*Hdh*^Q111/Q111 and MEF YAC128 cells. Methylation at this site was consistently regulated by METTL3 pharmacological inactivation and siRNA knockdown or targeted demethylation, showing a decrease in the methylation ratio by approximately 30–40%, further confirming the methylation status of the GGACA site. While it is intriguing that no effect is observed at other sites analyzed by

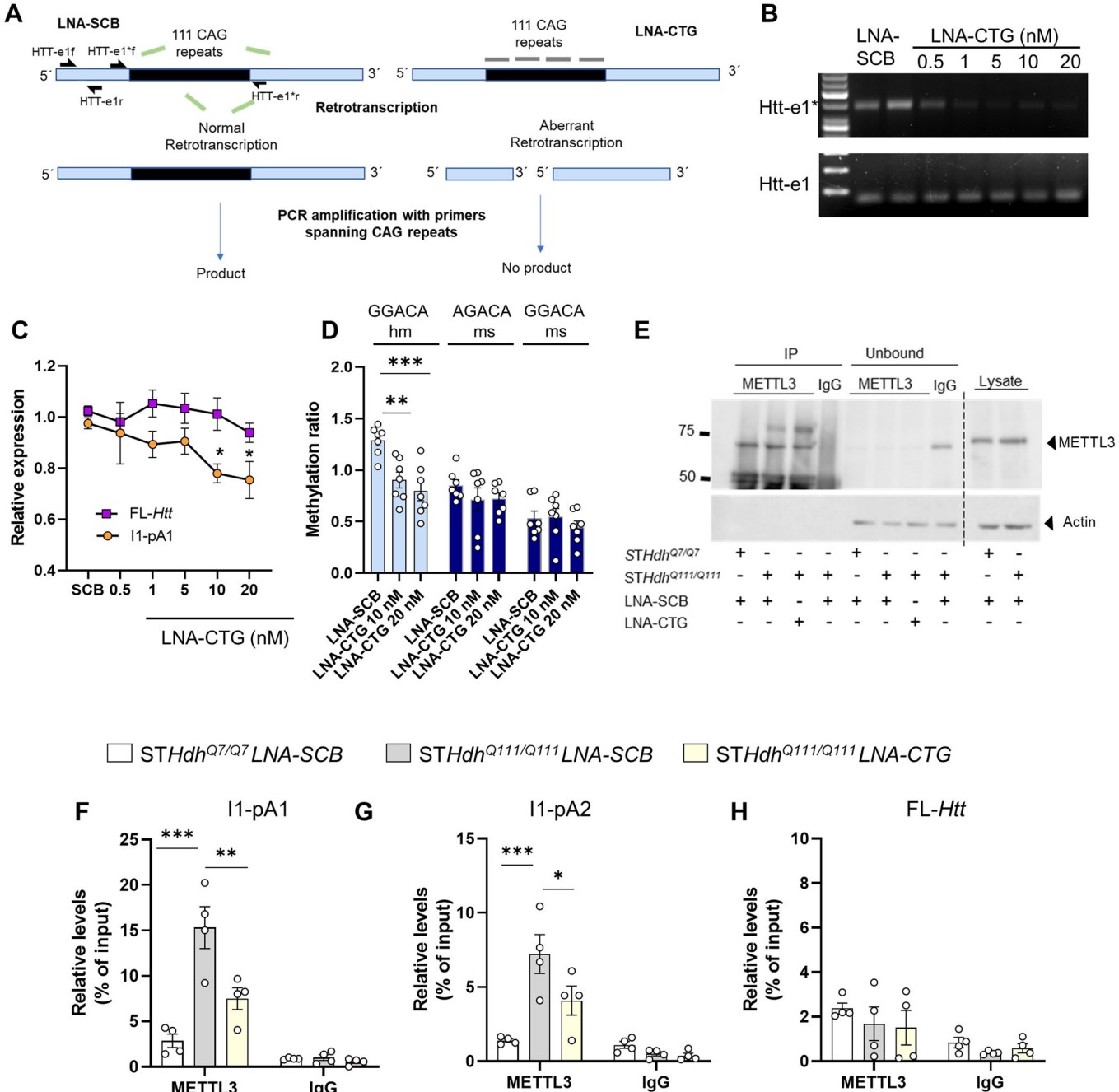

MazF-qPCR, it is possible that those sites are not sufficiently methylated, and the assay might not be sensitive enough to detect small reductions in m⁶A methylation. This aligns with previous findings showing that less than 20% of transcript copies in the cell will have m⁶A at a specific site, with only a small subset of DRACH sites having methylation as high as 20% or possibly even higher (Murakami and Jaffrey, 2022). Indeed, it has been suggested that every DRACH site may be methylated to some degree, existing along a spectrum of methylation. This is reflected in our analysis of predicted m⁶A sites in *Htt1a* by DRS showing different confidence of m⁶A modification at the different positions. It is important to note that the MazF-qPCR method used in this study is limited to quantifying m⁶A methylation at

ACA motifs (Garcia-Campos et al, 2019; Zhang et al, 2019). Therefore, we cannot exclude methylation of the other m⁶A sites predicted by DRS that could not be assessed. For instance, MEF zQ175 cells, lacking the human GGACA motif, might have another m⁶A-modified site not analyzed in this work. Consistent with this, *Htt1a* levels decreased in zQ175 MEFs when METTL3 was inhibited.

The HD models used in our study are characterized by long CAG tracts, often associated with juvenile HD. These models may not precisely recapitulate the distribution of germline human HD alleles commonly associated with adult onset HD, which typically range around 40-50 repeats. However, somatic repeat expansion invariably results in significantly longer alleles in HD brains, often

Figure 6.   Blockage of expanded CAG repeats using LNA-CTG ASOs downregulates methylation in m*Htt* intron 1 RNA and decreases the levels of *Htt1a*.

(A) Scheme showing LNA-CTGs binding determination by the lack of PCR amplification within the LNA-bound region due to the strong incompatibility of LNA-CTG:CAG duplexes with retrotranscription and subsequent PCR amplification. *HTT*-e1binding sites of the primers used for PCR amplification in *HTT* exon 1 (*HTT_e1** and *HTT_e1* sets of primers) are shown. (B) Gel electrophoresis showing *HTT* RT-PCR products from ST*Hdh*$^{Q111/Q111}$ cells transfected with different concentrations of LNA-CTG or LNA-SCB using primer sets HTT-e1* (for amplification of CAG expansions) and HTT-e1 (for amplification at 5′ of CAGs expansions). (C) qPCR analysis of the expression levels of FL-*Htt* and I1-pA1 transcripts in ST*Hdh*$^{Q111/Q111}$ cells transfected with LNA-SCB and LNA-CTG at 0.5, 1, 5, 10 and 20 nM (n = 3–8 independent experiments; 2–3 technical replicates/experiment). Data represent the mean ± SEM. Data were analyzed using one-way ANOVA with Tukey´s multiple comparisons test. *P < 0.05 compared to LNA-SCB-transfected cells. (D) MazF-qPCR analysis of the methylation ratio of *Htt* intron 1 in ST*Hdh*$^{Q111/Q111}$ cells transfected with 10 and 20 μM LNA-CTG or LNA-SCB. Data represent the mean ± SEM. Data were analyzed using one-way ANOVA with Tukey´s multiple comparisons test. **P < 0.01, ***P < 0.001 compared to LNA-SCB-transfected cells (n = 7 independent experiments). (E–H) RIP-qPCR analysis to detect interaction between METTL3 and m*Htt* transcripts in ST*Hdh*$^{Q7/Q7}$ treated with LNA-SCB and ST*Hdh*$^{Q111/Q111}$ treated with LNA-SCB and LNT-CTGs. (E) Western blot analysis showing the presence of METTL3 in the immunoprecipitated (IP) and unbound fractions (supernatant) in ST*Hdh*$^{Q7/Q7}$ and ST*Hdh*$^{Q111/Q111}$ treated with LNA-SCB and LNT-CTGs. (F–H) RIP-qPCR analysis showing enrichment of (F) I1-pA1, (G) I1-pA2 and (H) FL-*Htt* transcripts precipitated by anti-METTL3 and anti-IgG in LNA-SCB and LNA-CTG treated cells (n = 4 independent experiments). The RNA enrichment is presented as IP/input. Data represent the mean ± SEM. Data were analyzed using one-way ANOVA with Tukey´s multiple comparisons test. *P < 0.05, **P < 0.01, ***P < 0.001 compared to LNA-SCB-transfected ST*Hdh*$^{Q7/Q7}$ cells. Source data are available online for this figure.

expanding to 100-500+ CAGs in HD-vulnerable Medium Spiny Neurons (Robert E. Handsaker et al, 2024; Mätlik et al, 2024; Telenius et al, 1994; Kennedy, 2003) which is a CAG length comparable to the model systems used in this study. Nevertheless, to better understand the relevance of our findings in HD patients we analyzed human samples with different CAG lengths. The observed increase in m⁶A levels at the GGACA site in human samples suggest that this methylation in *Htt*/*HTT* intron 1 is conserved between mouse and human *Htt* transcripts within a unique context affected by *Htt* mutation. It is possible that in putamen samples with VG2-3, 3 the increase in methylation might be driven by CAG somatic expansions that has been recently shown to be a critical first step in HD pathogenesis leading to several dysfunctional cellular process in MSNs from HD brains (Mätlik et al, 2024). However, whether and how these perturbations impact the m⁶A epitranscriptomic machinery in the HD brain remains unknown but warrants exploration in future studies.

The central question arising from the identification of m⁶A in this region of intron 1 is whether the modification is involved in the aberrant processing of mHTT RNA. Our results in HD mouse cell lines show that inhibition of METTL3 activity resulted in a specific reduction of approximately 50% in *Htt1a* levels without changing the expression levels of FL-*Htt*. Although the observed decrease in m⁶A levels in *Htt* intron 1 following STM2457 treatment could contribute to the observed effect in *Htt1a* levels, we cannot rule out the impact of reduced m⁶A modifications on other factors that could be critically involved in *Htt* RNA processing. For instance, it has been shown that METTL3 regulates RNA splicing through m⁶A-mediated translational control of splicing factors (Wu et al, 2023). A broad distribution of m⁶A modifications across the CDS and 3'UTR has been demonstrated to regulate the expression of several targets of TDP43 (McMillan et al, 2023), a nuclear RNA-binding protein integrally involved in RNA processing and previously associated with HD pathology (Sanchez et al, 2021; Tada et al, 2012). Interestingly, a recent study proposed a coregulatory role of TDP-43 with m⁶A modification in posttranscriptional RNA processing in HD (Thai B. Nguyen et al, 2023). On the other hand, m⁶A can mediate mRNA degradation through the combined effects of YTHDF readers on m⁶A target transcripts (Zaccara and Jaffrey, 2020). Thus, we propose that METTL3 could influence *Htt1a* expression by regulating the deposition of m⁶A in m*Htt* intron 1 as well as in other m⁶A-dependent transcripts with potential roles in splicing or stability. Moreover, we cannot exclude the

possibility that the interaction of m⁶A with other RNA binding proteins, such as TDP43, is involved in alternative splicing.

To elucidate whether m⁶A deposition in *Htt* intron 1 is directly involved in *Htt1a* generation, we interrogated the effect of m⁶A modifications using a CRISPR/dCas13b system fused to ALKBH5 to demethylate *Htt* intron 1 in a target-specific manner. Recent studies have reported the potential of CRISPR technology in the targeting of m⁶A modifications. A fusion protein linking inactive dCas13b to truncated METTL3 (Wilson et al, 2020) or ALKBH5 (Li et al, 2020; Chen et al, 2021) allowed site-specific m⁶A incorporation or removal, respectively, with low off-target effects. N6-methyladenosine editing with CRISPR/dCas13 has already been successfully applied to several cancer models by targeting aberrant methylation of oncogenes (Gao et al, 2020; Li et al, 2020), hence constituting a promising approach to modulate m⁶A at specific transcripts. Notably, using this system in our immortalized mutant ST*Hdh*$^{Q111/Q111}$ cells, we detected a greater reduction in m⁶A methylation in comparison with the pharmacological inhibition of METTL3, showcasing the unique advantages of this CRISPR-based approach. In contrast to the broad inhibition of METTL3, a CRISPR/Cas system provides optimal targeting ability for the removal of m⁶A on specific sites, avoiding global m⁶A regulation interference (Zhang et al, 2021). This precision enhances the reliability of results when investigating the biological functions of m⁶A. Indeed, the precise regulation of m⁶A in *Htt* intron 1 was associated with a moderate but consistent reduction of ~20% in *Htt1a* without affecting the expression of FL-*Htt*, indicating that intronic m⁶A modifications are specifically involved in the regulation of *Htt1a* expression (Sathasivam et al, 2013). It's important to note that the decrease in *Htt1a* observed with different approaches in this study did not translate to an increase in FL-*Htt* levels. This might be due to additional regulatory mechanisms influencing FL-*Htt* mRNA levels. *Htt* gene transcripts have been shown to exhibit lengthened poly(A) tails (Picó et al, 2021), which could increase mRNA stability, potentially masking changes in expression levels resulting from the shift from incomplete splicing to constitutive splicing. The possibility that decreased levels of *Htt1a* observed by demethylation are a consequence of RNA degradation was ruled out by performing an RNA decay assay suggesting that m⁶A modifications may not affect the stability of *Htt1a* as previously described for other m⁶A-containing mRNAs (Wang et al, 2014). Thus, we conclude that our RNA editing system mediates efficient m⁶A demethylation in *Htt* intron 1 and allows us to establish a causal relationship between m⁶A deposited in *Htt* RNA and *Htt1a* generation.

To understand how m⁶A levels in *Htt* intron 1 impact HD pathology, we performed site-specific manipulation of m⁶A levels and evaluated DNA damage and ATP production in ST*Hdh*^(Q111/Q111) cells. Our results showed that site-specific target demethylation of m*Htt* leads to a reduction of the DNA damage which is mainly affected in these cells by the augmentation of stress pathways, activated DNA damage response and apoptotic signals (Trettel, 2000; Cattaneo et al, 2022). This effect was accompanied by an increase in ATP production, which is typically reduced in these cells due to mitochondrial dysfunction (Gines, 2003) and compromised during DNA damage repair (Formentini et al, 2009). The N-terminus of the HTT protein has been shown to disrupt DNA damage repair mechanism(s), leading to the excessive accumulation of DNA damage/strand breaks and to localize in sites of DNA damage (Gao et al, 2019a). Thus, the observed effect in our experiment could be driven by a reduction in the production of the toxic 90 aa N-terminal HTT-exon1 protein caused by the downregulation of *Htt1a* transcripts. However, in line with the potential role of *Htt*a in disease progression by the formation of RNA foci at transcriptional sites (Ly et al, 2022), it is also possible that reduced methylation in *Htt1a* would reduce the interaction with other *Htt* transcripts in RNA clusters, avoiding, for instance, sequestering RNA binding proteins involved in DNA damage as well as in mitochondrial function (Fijen and Rothenberg, 2021).

Several studies have shown that m⁶A can modulate RNA splicing, potentially creating crosstalk between transcription and pre-mRNA splicing (Mendel et al, 2021; Kasowitz et al, 2018; Louloupi et al, 2018; Zhou et al, 2019; Akhtar et al, 2021; Yang et al, 2019). Here, we propose that one possible mechanism for m⁶A-dependent aberrant splicing regulation in m*Htt* involves the pausing of RNA polymerase II (Pol II) (Akhtar et al, 2021; Zhou et al, 2019) and the formation of R-loops to facilitate transcription termination (Yang et al, 2019). This aligns with evidence showing that CAG repeats and elements in intron 1 can reduce Pol II elongation, potentially leading to *HTT1a* generation (Neueder et al, 2018). Our findings highlight intronic m⁶A modifications as a potential mechanism contributing to aberrant splicing of m*Htt*. Further investigations are needed to elucidate whether m⁶A influences mis-splicing by directly controlling Pol II pausing or by promoting stalling through R-loop formation.

Finally, we investigated whether expanded CAGs influence the deposition of m⁶A in this proximal site of intron 1. We used LNA-CTG ASOs that strongly bind to CAG RNAs, potentially disrupting their secondary structure and/or blocking their activity in exon 1 (Rué et al, 2016). Our data show that ASOs downregulated methylation levels in intron 1 and *Htt1a* expression levels, suggesting a potential role for CAG repeats in regulating this methylation. In the context of m*Htt* exon 1, CAG repeats form RNA stable hairpin structures (Jasinska, 2003) that aberrantly interact with several proteins, the majority of which belong to the spliceosome pathway (Schilling et al, 2019) and can reduce the elongation rate of PolII (Neueder et al, 2018). This finding aligns with previous reports demonstrating that m⁶A deposition can be determined by RNA secondary structure, sequence motifs and exon–intron architecture (Meiser et al, 2020; Schwartz et al, 2013; Gao et al, 2020; Uzonyi et al, 2023). Indeed, it has been recently shown that m⁶A hypermethylation may result from loss of endogenous RNA exon architecture and exon junction complex (EJC) protection (He et al, 2023), as well as the presence of a pause PolII during transcription (Wang et al, 2024). In this context, the CAG repeats in *Htt* RNA might either directly deprotect a proximal region by EJC protein recruitment or, through its secondary structure, influence the dynamics of the

methyltransferase complex, leading to higher m⁶A deposition during *Htt* pre-mRNA transcription. Here, we demonstrate that METTL3 interacts significantly with *Htt1a* transcripts in ST*Hdh*^(Q111/Q111) cells but not in ST*Hdh*^(Q7/Q7) cells, suggesting that METTL3 recruitment to intron 1 is promoted by long CAG repeats. Additionally, we observed a significant reduction in this interaction when ST*Hdh*^(Q111/Q111) cells were treated with LNA-CTGs. Based on these findings, we hypothesize that ASOs bind to the CAG repeats of the transcript during RNA elongation, potentially altering RNA exon 1 architecture and influencing both PolII speed and METTL3 recruitment. Alternatively, ASOs might reduce the recruitment of EJC proteins and other factors that contribute to m⁶A methylation through different mechanisms. Hence, our results point to the existence of context-dependent pathological features that guide m⁶A modification of *Htt* RNA, contributing to incomplete *Htt* splicing. Interestingly, in *Hdh*^(+/Q111) mice a substantial increase in m⁶A enrichment in *Htt1a* is detected at 8 months coinciding with the time when somatic expansions are more abundant. This further support the potential role of CAG expansions in promoting m⁶A deposition. However, the causality between instability of CAG expansions and methylation levels in *Htt* transcripts remains to be determined. Future studies aimed at better understanding this relationship are warranted.

While our results support the role of m⁶A in the generation of *Htt1a* in HD mouse models, we cannot establish a correlation between the m⁶A methylation in m*HTT* intron 1 and levels of *HTT1a* in human HD brain cells. This discrepancy could be due to several factors. First, previous studies have reported challenges in detecting increased *HTT1a* expression in adult post-mortem HD brain samples(Neueder et al, 2017; Hoschek et al, 2024). *HTT1a* can form RNA clusters in HD human brains (Ly et al, 2022) and might not have been fully solubilized during RNA extraction. This suggests that the amount of *HTT1a* generated might be underestimated, since only the cytoplasmic, soluble *HTT1a* RNA fraction can be analyzed. Second, bulk RNA analysis from post-mortem putamen samples might not be a sensitive enough approach to detect significant *HTT1a* changes. Studies suggest extensive somatic CAG expansions (>100 CAG repeats), which can promote *HTT1a* generation, appears to be present in a minority of striatal medium spiny neurons (Robert E. Handsaker et al, 2024). Furthermore, it is generally understood that neurons with extreme CAG repeat expansions (150–180+ CAGs) are short-lived and undergo rapid cell death. This means that even when *HTT1a* is produced from such expansions, it encodes a highly toxic protein and may be difficult to detect due to concurrent neuronal loss. Our current findings in human HD samples suggest that m⁶A methylation is primarily detected in pre-mRNA. We hypothesize that this modification might play a role in the generation of *HTT1a* when longer CAG repeats are present in the mutant HTT gene, as observed in our HD mouse models and cell lines.

Overall, our study provides new insights by demonstrating the presence of m⁶A modifications in mutant huntingtin intron 1 and its potential contribution to the pathogenic mechanism affecting m*Htt* RNA metabolism. Importantly, our evidence may support the development of therapeutic strategies already proposed (Tabrizi et al, 2019) such as targeting the pathological processing of *HTT* mRNA or the *HTT* exon 1-intron 1 junction to lower *HTT1a* in HD mutation carriers. For instance, these modifications might be relevant to consider when designing ASOs targeting *HTT1a*, as the m⁶A-modified sites could hinder ASO binding or destabilize ASO due to improper base pairing, thereby reducing its effectiveness. Moreover, we demonstrate that the CRISPR/dCas13b-ALKBH5 approach could achieve a moderate reduction in the toxic mutant *Htt1a* fragment without affecting the WT allele.

Intriguingly, using nonallele-specific ASOs, reduction of 43% in *mHtt* mRNA has been reported to be enough to prevent further brain loss in symptomatic R6/2 mice and significantly increase lifespan (Kordasie-wicz et al, 2012). In contrast, our strategy selectively targets the biogenesis of a highly pathogenic transcript derived from m*Htt*, obtaining a reduction of approximately 20% by merely modifying the methylation status of m*Htt* RNA. Given that slight reductions in *mHtt* mRNA levels are enough to ameliorate HD symptomatology in mouse models, future experiments validating the effects of *Htt1a* demethylation in vivo could provide encouraging results.

Our study highlights the need for a deeper understanding of the pathogenic mechanisms influencing m*HTT* RNA metabolism, with a particular emphasis on RNA modifications. This under-standing could open new avenues for novel gene therapy strategies aimed at targeting the mutant RNA allele or modifying the splicing process that generates *HTT1a*, both of which warrant further exploration.

## Methods

### Reagents and tools table

| Reagent/resource | Reference or source | Identifier or catalog number |
|---|---|---|
| **Experimental models** | | |
| Human *postmortem* putamen samples (*H. sapiens*) | Neurological Tissue Bank (Biobanc-Hospital Clínic-Institut d'Investigacions Biomèdiques August Pi I Sunyer (IDIBAPS)) | Table EV1 |
| Human skin fibroblasts (*H. sapiens*) | Garcia-Forn et al, 2023; Coriell Institute of Medical Research | Table EV2 |
| *Hdh*$^{+/Q111}$ knock-in mice (*M. musculus*) | Wheeler, 1999 | N/A |
| ST*Hdh*$^{Q7/Q7}$ / ST*Hdh*$^{Q111/Q111}$ striatal cell line (*M. musculus*) | Trettel, 2000 | N/A |
| YAC128 mouse embryonic fibroblasts (*M. musculus*) | Fienko et al, 2022 | N/A |
| zQ175 mouse embryonic fibroblasts (*M. musculus*) | Mason et al, 2020 | N/A |
| **Recombinant DNA** | | |
| dPsPCas13b-ALKBH5 SP-gRNA 1: 5′-TAG TTA AAC CAG GTT TTA AGC ATA GCC AGA -3′ | This study | N/A |
| dPsPCas13b-ALKBH5 SP-gRNA 2: 5′-ACT CCA GTG CCT TCG CCG TTC CCA GTT TGC-3′ | This study | N/A |
| dPsPCas13b-ALKBH5 SP-gRNA 3: 5′-AGC CTT GTT GGG GCC TGT CCT GAA TTC GAT-3′ | This study | N/A |
| dPsPCas13b-ALKBH5 NT-gRNA: 5′-AGT GCT CAC TCT GGT GTC ACA GTG CTG CA-3′ | This study | N/A |
| **Antibodies** | | |
| Mouse monoclonal anti-m6A antibody (5 µg) | Synaptic Systems | cat# 202 111 |
| Rabbit monoclonal anti-METTL3 antibody (5 µg) | Abcam | cat# 195352 |
| Mouse monoclonal anti-phospho-histone H2AX (Ser139) (1:1000) | Sigma-Aldrich | cat# 05-636-I |
| Rabbit polyclonal anti-ALKBH5 (1:2000) | Sigma-Aldrich | cat# HPA007196 |
| Cy3 AffiniPure Goat Anti-Rabbit IgG (1:500) | Jackson ImmunoResearch | Cat# 111-165-003 |
| **Oligonucleotides and other sequence-based reagents** | | |
| PCR primers | This study | Tables EV4–7 |
| siRNAs against METTL3 | Ambion | cat# AM16708; Table EV3 |
| siRNA pool against METTL3 | Santa Cruz | cat# sc-149387; Table EV3 |
| siRNA NTC | Ambion | cat# AM4611; Table EV3 |
| LNA-CTG: CTGCTGCTGCTGCTGCTGCTGCT | Qiagen; Rué et al, 2016 | N/A |
| LNA-SCB: GTGTAACACGTCTATACGCCCA | Qiagen; Rué et al, 2016 | N/A |
| Oligo(dT)18 Primer | Invitrogen | cat# SO131 |
| **Chemicals, enzymes and other reagents** | | |
| DMEM—high glucose | Sigma–Aldrich | cat# D5671 |
| Fetal bovine serum (FBS) | Diagnovum | cat# D061-500ML |
| Penicillin–streptomycin | Diagnovum | cat# D910-100ML |
| Geneticin (G418 Sulfate) | Thermo Scientific | cat# 11811-023 |
| Blasticidin | Gibco | cat# R21001 |
| Opti-MEM Reduced Serum Medium | Gibco | cat# 31985070 |
| STM2457 | TargetMol | cat# T9060 |
| Lipofectamine 3000 reagent | Invitrogen | cat# L3000-008 |
| RNeasy Lipid Tissue Mini Kit | Qiagen | cat# 74804 |
| DNase I | Sigma–Aldrich | cat# AMPD1 |
| Protein A Dynabeads | Invitrogen | cat# 10002D |
| N6-Methyladenosine 5′-monophosphate sodium salt (m$^6$A salt) | Sigma–Aldrich | cat# M2780 |
| RNase Inhibitor | Promega | cat# N2615 |
| RNeasy MinElute Cleanup kit | Qiagen | cat# 74204 |
| mRNA Interferase - MazF | Takara Biotechnology | cat# 2415A |
| High-Capacity cDNA Reverse Transcription Kit | Applied Biosystems | cat# 4368814 |

| Reagent/resource | Reference or source | Identifier or catalog number |
|---|---|---|
| M-MLV reverse transcriptase | Invitrogen | cat# 28025013 |
| Premix Ex Taq master mix for probe-based real-time PCR | Takara Biotechnology | cat# RR390A |
| HotStar Taq Plus DNA Polymerase | Qiagen | cat# 203603 |
| GeneScan 500 LIZ | Thermo Fisher Scientific | cat# 4322682 |
| Dynabeads mRNA DIRECT Micro Purification Kit | Invitrogen | cat# 61021 |
| Qubit RNA High Sensitivity (HS) | Invitrogen | cat# Q33224 |
| Direct RNA Sequencing Kit | Oxford Nanopore Technologies | cat# SQK-RNA004 |
| Actinomycin D | Sigma-Aldrich | cat# A9415 |
| 3'RACE System for Rapid Amplification of cDNA Ends | Invitrogen | cat# 18373-019 |
| Platinum II Hot-Start PCR Master Mix | Invitrogen | cat# 14000-013 |
| Gel loading dye | NewEngland Biolabs | cat# B7024S |
| SYBRS Safe DNA gel stain | Invitrogen | cat# S33102 |
| 1 Kb Plus DNA ladder | Thermo Fisher Scientific | cat# 10787018 |
| GeneJET Gel extraction Kit | Thermo Fisher Scientific | cat# K0691 |
| ExoSAP-IT Express PCR Product Cleanup kit | Applied Biosystems | cat# 75001 |
| EpiQuik m6A RNA Methylation Quantification Kit | Epigentek | cat# P-9005 |
| CellTiter-Glo 2.0 Cell Viability Assay Kit | Promega | cat# G9241 |
| CyQUANT Cell proliferation Assay Kit | Invitrogen | cat# C7026 |
| 4',6-diamidino-2-phenylindole, dihydrochloride (DAPI) | Sigma-Aldrich | cat# D9542 |
| **Software** | | |
| OligoAnalyzer IDT | https://www.idtdna.com/pages/tools/oligoanalyzer | |
| Nucleotide BLAST NCBI | https://blast.ncbi.nlm.nih.gov/Blast.cgi | |
| GeneMapper v5 | https://www.thermofisher.com/order/catalog/product/es/es/A38892 | |
| MinKNOW version 24.02.19 | https://nanoporetech.com/es/news/news-introducing-new-minknow-app | |
| dorado version 0.7.2 | https://github.com/nanoporetech/dorado | |
| samtools version 1.20 | https://github.com/samtools/samtools/releases/ | |

| Reagent/resource | Reference or source | Identifier or catalog number |
|---|---|---|
| minimap2 (v 2.17-r941) | https://github.com/lh3/minimap2/releases | |
| Modkit version 0.3.1 | https://github.com/nanoporetech/modkit | |
| Integratives Genomic Viewer (IGV) | https://igv.org/ | |
| CellProfiler | https://cellprofiler.org/ | |
| GraphPad Prism version 8.0.2 | https://www.graphpad.com/features | |
| SnapGene viewer | https://www.snapgene.com/ | |
| **Other** | | |
| Nanodrop 1000 spectrophotometer | Thermo Fisher Scientific | |
| StepOnePlus Real-Time PCR System | Applied Biosystems | |
| ABI 3730xl DNA analyzer | Thermo Fisher Scientific | |
| Agilent 2200 TapeStation System | Agilent | |
| Qubit Fluorometer | Invitrogen | |
| Oxford Nanopore PromethION 24 Series device | Oxford Nanopore Technologies | |
| ABI3730XL DNA Analyzer | Thermo Fisher Scientific | |
| Leica Confocal SP5 microscope | Leica | |
| Infinite 200 PRO reader | Tecan | |
| Chemidoc Imaging system | Bio-rad | |

## Post-mortem brain tissue

Human post-mortem samples derived from the putamen (7 controls and 9 HD patients) were obtained from the Neurological Tissue Bank (Biobanc-Hospital Clínic-Institut d'Investigacions Biomèdiques August Pi I Sunyer (IDIBAPS) according to the guidelines and approval of Barcelona's Clinical Research Ethical Committee (Hospital Clínic). All ethical guidelines contained within the latest Declaration of Helsinki were taken into consideration and approved by the Institutional Review Board of the University of Barcelona (IRB00003099, 06/28/2021). Clinical details of controls and HD patients are summarized in Table EV1.

Informed consent was obtained from all subjects involved in the study.

## Human skin fibroblasts

Human skin fibroblasts were obtained from controls and HD patients at different clinical stages (7 controls, 7 pre-symptomatic HD patients and 12 symptomatic HD patients) (Garcia-Forn et al, 2023). The two HD juvenile skin fibroblasts (GM9197 with 180 CAG repeats and GM4281 with 80 CAGs) were obtained from Coriell Institute for Medical Research. All procedures were approved by the Ethics Committees of the Hospital de la Santa Creu I San Pau de Barcelona and the Universitat de Barcelona (IRB00003099, 07/20/2023), and informed written consent was obtained from all subjects. The clinical

data of the subjects are summarized in Table EV2. Cells derived from sterile, nonnecrotic skin biopsies were grown at 37 °C and 5% $CO_2$ in Dulbecco's modified Eagle's medium (DMEM) with 25 mM glucose (Gibco, ref. 41966-029) supplemented with 10% fetal bovine serum (FBS), 1% penicillin–streptomycin and 1% amphotericin B.

## Animals

Heterozygous $Hdh^{+/Q111}$ knock-in mice (Wheeler, 1999) were used as an HD mouse model. These mice were maintained on a C57BL/6J (Charles River) genetic background and present a targeted insertion of 109 CAGs in the murine huntingtin gene that extends the resulting polyglutamine segment to 111 residues. The CAG repeat size for the mice used in this study was 112–119. Male $Hdh^{Q7/Q7}$ WT mice were crossed with female heterozygous $Hdh^{+/Q111}$ mice to obtain age-matched WT and $Hdh^{+/Q111}$ littermates. Only males from each genotype were analyzed. Mice were housed with access to food and water ad libitum in a colony room kept at 19–22 °C and 40–60% humidity under a 12:12 h light/dark cycle. Animals were sacrificed at 2 and 8 months of age through cervical dislocation, and brains were rapidly frozen in dry ice and stored at −80 °C until further analysis. All mouse procedures were performed in compliance with the National Institutes of Health Guide for the Care and Use of Laboratory Animals and approved by the local animal care committee of the Universitat de Barcelona (448/17) and the Generalitat de Catalunya (9878 P2), in accordance with the European (2010/63/EU) and Spanish (RD53/2013) guidelines for the care and use of laboratory animals.

## Immortalized cell cultures

Conditionally immortalized murine homozygous wild-type ST$Hdh^{Q7/Q7}$ and mutant ST$Hdh^{Q111/Q111}$ striatal cell lines (Trettel, 2000) presenting endogenous levels of normal or mutant huntingtin (with 7 and 111 glutamines, respectively) were used. Cells were maintained at 33 °C and 5% $CO_2$ in DMEM (Sigma-Aldrich, ref. D5671) supplemented with 10% FBS, 1 mM sodium pyruvate, 2 mM L-glutamine, 1% penicillin–streptomycin and 400 μg/mL Geneticin (G418 Sulfate) (Thermo Scientific, ref. 11811-023). Transformed mouse embryonic fibroblast (MEF) lines had been derived from YAC128 mice (Fienko et al, 2022) and zQ175 knock-in mice. YAC128 MEFs carry wild-type mouse (Mm) Htt mRNA (7 CAG) and a full-length human (Hs) HTT transgene modified in exon 1 to undergo 125 glutamines repeat expansion (composed primarily of CAG codons but also containing 9 interspersed CAA codons) (Pouladi et al, 2012; Fienko et al, 2022). Transformed zQ175 MEFs were established as previously described (Fienko et al, 2022). They carry exon 1 from human HTT with a highly expanded CAG repeat (~190 CAG repeats) (Mason et al, 2020). MEFs were maintained in DMEM supplemented with 10% FBS, 1 mM sodium pyruvate, 2 mM L-glutamine, and 1% penicillin–streptomycin in a humidified incubator at 37 °C with 5% $CO_2$. HD cells were seeded in 6- or 12-well plates (for qPCR experiments) or in 24-well plates (for immunocytochemistry) at a density of $1.5 \times 10^6$ cells/cm².

## METTL3 pharmacological inhibition with STM2457 and siRNA expression knockdown

Pharmacological inhibition of METTL3 was performed with STM2457 (TargetMol, ref. T9060) at 10 μM and 20 μM for 48 h

(Yankova et al, 2021). To reduce the expression of METTL3, we transfected ST$Hdh^{Q111/Q111}$ striatal cells with two different siRNAs for METTL3 (Ambion, ref. AM16708; sequence provided in Table EV3) and a pool of 3 METTL3-specific 19–25 nt siRNAs designed to knockdown (Santa Cruz, ref.sc-149387). Cells ($2 \times 10^5$ cells/well) were grown in antibiotic-free growth medium with low fetal bovine serum and incubated in a $CO_2$ incubator at 37 °C for one day prior to transfection. For cell transfection, a mixture of Lipofectamine 3000 reagent (Invitrogen, ref. L3000-008), siRNAs (30 nM) and Optimem was used.

## Generation of dCas13b-ALKBH5 plasmid constructs

Site-specific manipulation of m⁶A levels at Htt intron 1 mRNA was achieved with the programmable RNA editing system dCas13b-ALKBH5. The RNA editor construct was designed as previously described (Cox et al, 2017; Li et al, 2020) with some modifications. Briefly, RNA-targeting catalytically inactive Type VI-B Cas13 enzyme from Prevotella sp P5-125 (dPspCas13b) (Cox et al, 2017) was fused to the m⁶A-demethylase ALKBH5 at the C-terminus of dCas13b with a six amino acid (GSGGGG) linker. The spacer (SP) sequences bound to the guide RNA (gRNA) were designed based on the intron 1 sequence of Htt, upstream of the first cryptic poly(A) site at 680 bp. Further evaluation of the SP sequences was performed using the OligoAnalyzer Tool (IDT) and Nucleotide BLAST (NCBI) to avoid matches at off-target locations. The sequences of the SP were as follows: 5′-TAG TTA AAC CAG GTT TTA AGC ATA GCC AGA-3′ (SP-gRNA 1); 5′-ACT CCA GTG CCT TCG CCG TTC CCA GTT TGC-3′ (SP-gRNA2); 5′-AGC CTT GTT GGG GCC TGT CCT GAA TTC GAT-3′ (SP-gRNA 3). A non-targeting gRNA (NT-gRNA) was used as a negative control: 5′-AGT GCT CAC TCT GGT GTC ACA GTG CTG CA-3′. The resulting fusion protein with its corresponding SP-gRNA was cloned and inserted into the PX458 vector by GenScript. The HA tag was included for detection of the fusion protein. The blasticidin S deaminase gene was also inserted into the construct to allow for the generation of stable cell lines. To control for the effects of transfection itself and steric hindrance, two control plasmids were used: one plasmid containing the catalytically dead ALKBH5 (H204A) (Feng et al, 2014) fused to dCas13b and the dCas13b plasmid without the demethylase, both with the spacer sequence (GeneScript).

## Cell transfections

Plasmid transfection was performed using Lipofectamine 3000 reagent (Invitrogen, ref. L3000-008) following the manufacturer's instructions. For six-well assays, cells were transfected with 2.5 μg/well of the corresponding plasmid (dCas13b-ALKBH5, dCas13b-ALKBH5 H204A and dCas13b-control). Forty-eight hours after transfection, cells were treated with the appropriate concentration of selection antibiotic (6 μg/mL Blasticidin for ST$Hdh^{Q7/Q7}$ cells and 4 μg/mL for ST$Hdh^{Q111/Q111}$ cells), and the medium with Blasticidin was changed every 2–3 days. Polyclonal cells that had integrated the plasmid of interest were expanded and seeded for immunohisto-chemistry, western blot analysis and RNA extraction.

Locked nucleic acid-antisense oligonucleotides (LNA-ASOs) were transfected with Lipofectamine 3000 at the dosages indicated in the figure legends. The LNA-ASO complementary to the CAG

repeat (LNA-CTG) consisted of a 20-nt oligonucleotide, CTGCTGCTGCTGCTGCTGCTGCT, with an LNA located every third T and a phosphorothioate-modified backbone. LNA-CTG and the control scrambled LNA-modified sequence (LNA-SCB) 5′-GTGTAACACGTCTATACGCCCA-3′ were obtained from Qiagen as previously described in Rue et al (Rué et al, 2016).

## RNA isolation

RNA from the corresponding biological samples was extracted using the RNeasy Lipid Tissue Mini Kit (Qiagen, ref. 74804), following the instructions of the manufacturer. Briefly, the frozen tissue was placed in QIAzol (Qiagen) and homogenized using a 25G syringe. In the case of cell cultures, the growth medium was discarded, and the cells were washed once with PBS and then collected from the plates by directly adding QIAzol reagent and homogenizing with a scraper. The purified RNA was eluted in nuclease-free $H_2O$, and the quantity and quality were measured using a Nanodrop 1000 spectrophotometer (Thermo Fisher Scientific). Total RNA was subjected to DNase treatment (Sigma-Aldrich, ref. AMPD1) according to the manufacturer's instructions. Samples were stored at $-80\,°C$ until use.

## MeRIP-qPCR

Relative quantification of $m^6A$ levels of the genes of interest was performed through m6A-RNA immunoprecipitation (MeRIP)-qPCR as described elsewhere (Pupak et al, 2022). Briefly, 3–4.5 µg of total non-fragmented RNA was incubated with anti-$m^6A$ antibody (Synaptic Systems, ref. 202 111) conjugated to Protein A Dynabeads (Invitrogen, ref. 10002D) in IP buffer (150 mM NaCl, 10 mM Tris-HCl, pH = 7.5, 0.1% IGEPAL CA-630 in nuclease-free $H_2O$) at 4 °C. The immunoprecipitated RNA was subjected to two rounds of competitive elution with $m^6A$-containing buffer (45 µL of 5× IP buffer, 75 µL of 20 mM $m^6A$ (Sigma–Aldrich, ref. M2780), 7 µL of RNase inhibitor and 98 µL of nuclease-free water), and the eluted RNA was then concentrated using the RNeasy MinElute Cleanup kit (Qiagen, ref. 74204). The immunoprecipitated RNA was then reverse-transcribed and used for qPCR. The fold enrichment was determined by calculating the Ct values of the MeRIP sample relative to the input sample.

## RIP-qPCR

Cells were lysed with cold RIP buffer (25 mM Tris-HCl pH 7.4, 5 mM EDTA, 150 mM KCl, 0.5 mM DTT, 0.5% NP-40, RNase inhibitor and protease inhibitor cocktail). After 10 min of incubation on ice cell lysates were centrifuged for 15 min at 12,000×$g$, the supernatant was collected, and 10% of this supernatant was removed per RIP reaction to act as 10% Input. For preparation of magnetic beads and immobilization of the antibody, 5 µg of anti-METTL3 antibody (abcam, ref. 195352) or rabbit IgG control were mixed with 50 µl of Dynabeads Protein A (Life technology. ref.10002D) for 1 h at room temperature. After antibody binding, 500 µg of these lysates were incubated with the anti-METTL3 antibody or rabbit IgG control bound to Dynabeads Protein A. After overnight incubation at 4 °C, the immunoprecipitation complex was washed twice with high-salt buffer (50 mM Tris-HCl pH 7.4, 300 mM NaCl), followed by two additional washes with low-salt buffer (50 mM Tris-HCl pH 7.4, 150 mM NaCl).

RNA was eluted from the beads with elution buffer (containing 20 mg/mL proteinase K and 1% SDS) and extracted with QIAzol reagent for further analysis. IP enrichment ratio of a transcript was calculated as ratio of its amount in IP to that in the input as follow: $\Delta CT$ (normalized RIP) = (average Ct [RIP] – (average Ct [Input]-log2 (Input dilution factor))), where Input dilution factor=(fraction of the input saved). To calculate the % input for each RIP fraction: % input $= 2^{(-\Delta CT[normalized\ RIP])}$.

## MazF-qPCR

For validation and stoichiometric quantification of $m^6A$ sites, we followed the protocol published by Garcia-Campos et al (Garcia-Campos et al, 2019), with minor modifications. Total RNA was heat denatured and digested with 10–20 U of MazF enzyme (TakaRa, ref. 2415A) for 15 min at 37 °C. RNA was then subjected to a cleanup protocol using the RNeasy MinElute Cleanup kit (Qiagen, ref. 74204), followed by RNA elution in water. Primer-probe sets for qPCR analysis were designed flanking a potential $m^6A$ motif that contains the "ACA" sequence (test motif), and a control primer-probe set (no ACA control) with no "ACA" site was designed in a nearby region of the motif of interest (Table EV4). Methylation levels were calculated based on the Ct values obtained from MazF-digested and nondigested samples for the test motif and the no ACA control (methylation ratio = ((MazF-digested test motif/MazF-digested no ACA motif)/ (nondigested test motif/ nondigested no ACA motif))).

## cDNA synthesis and real-time quantitative PCR (qPCR) assays

45–500 ng of total, MeRIP, RIP or MazF-digested RNA was reverse transcribed using the High-Capacity cDNA Reverse Transcription Kit (Applied Biosystems, ref. 4368814) according to the manufacturer's instructions. On the other hand to analyze *Htt/HTT* transcripts, reverse transcription was performed on 0.5–1 µg of RNA using the M-MLV reverse transcriptase (RT) (Invitrogen) according to the company's protocol and using oligo-dT (18) primers (Invitrogen) as previously described (Papadopoulou et al, 2019). PrimeTime qPCR Assays were purchased from Integrated DNA Technologies (IDT) to measure genes of interest and housekeeping genes (Tables EV4 and EV5). The qPCR was performed on 96-well plates in a final volume of 12 µL using the Premix Ex Taq Probe-based qPCR assay (Takara Biotechnology, ref. RR390A). All reactions were run in duplicate on a StepOnePlus Real-Time PCR System (Applied Biosystems) set to the following cycling program: 1 cycle 95 °C for 30 s; 40 cycles 95 °C for 5 s, 60 °C for 20 s. Relative enrichment was calculated using the ΔΔCt method, with actin-ß (mouse) expression serving as a housekeeping gene.

To evaluate the performance of reverse transcription in the presence of bound LNA-ASOs, we performed PCRs using HotStartTaq Plus DNA Polymerase (Qiagen ref:203603). PCR amplification was performed using exon 1 sequence-specific primers (Table EV6). PCR products were loaded and run on a 2% agarose gel.

### *HTT* CAG repeat expansion analysis

Somatic CAG instability was evaluated in the striatum of $Hdh^{+/Q111}$ at 2 and 8 months of age ($n = 4$ 5), using RNA converted to cDNA as described above. Tail genomic DNA from the same mice was

used to determine the respective inherited CAG length. Somatic instability was determined as previously described (Pinto et al, 2013), using a human-specific PCR assay that amplifies the *HTT* CAG repeat from the knock-in allele, but does not amplify the mouse sequence. The forward primer was fluorescently labeled with 6-FAM and products were resolved using the ABI 3730xl DNA analyzer (Thermo Fisher Scientific) with GeneScan 500 LIZ as internal size standard (Thermo Fisher Scientific). GeneMapper v5 (Thermo Fisher Scientific) was used to generate CAG repeat size distribution traces. Somatic CAG expansion indices were calculated as previously described (Lee et al, 2010), using a 5% relative peak height threshold cut-off and normalization to the peak with the greatest intensity within each trace.

## Nanopore sequencing

Total RNA was extracted from $Hdh^{Q7/Q7}$ and $Hdh^{+/Q111}$ mouse striatal tissue using the RNeasy Lipid Tissue Mini Kit (Qiagen, ref. 74804), following the instructions of the manufacturer. Enrichment of polyadenylated RNA from total RNA was performed using Dynabeads™ mRNA DIRECT™ Micro Purification Kit (Invitrogen, ref.61021) following the manufacturer's protocol. Polyadenylated RNA was isolated from pooled RNA samples of 3–4 mice per genotype, yielding 600–750 ng per biological replicate ($n = 2$ for $Hdh^{+/Q111}$ mice, $n = 1$ for WT). The integrity was quantified using the Agilent 2200 TapeStation System. RNA concentration was measured with Qubit using RNA HS Assay (Invitrogen™ Q33224).

The obtained mRNA (600–750 ng) was used for library preparation using the Direct RNA Sequencing Kit following the manufacturer's instructions (Oxford Nanopore Technologies, SQK-RNA004). Four libraries with 2 biological replicates per genotype were generated, each consisting of pooled mRNA from 4 to 5 mice. The prepared libraries were sequenced on an Oxford Nanopore PromethION 24 Series device using a FLOPRO004R flow cell and the sequencing data was collected for 48 h. The quality parameters of the sequencing runs were monitored in real-time using the MinKNOW platform version 24.02.19. Libraries preparation and sequencing was performed in Centro Nacional de Análisis Genómico (CNAG).

## Analysis of direct RNA sequencing datasets

Raw pod5 files from WT and KI PromethION runs were basecalled using dorado (https://github.com/nanoporetech/dorado) version 0.7.2, with the modified base model rna004_130bps_sup@v3.0.1_-m6A_DRACH@v1. To examine different mapping settings and parameters, BAM output files from dorado were converted to FASTQ files using samtools version 1.20, with a F3840 flags, also keeping the methylation information (auxiliary tags) in the FASTQ files. Alignment was then performed using minimap2 (v 2.17-r941) to the mouse genome (mm39) supplemented with the knock-in mutated *HTT* gene construct, with the following parameters: -ax map-ont -k 14. The modkit tool (https://github.com/nanoporetech/modkit) version 0.3.1 was used on the BAM files to produce bedMethyl files. m⁶A sites were predicted for those positions with sequencing depth>=25 and non-0 modification frequencies, generating BED files that were used for visualization of m⁶A sites in the Integratives Genomic Viewer (IGV).

## Assessment of mRNA stability

ST$Hdh^{Q111/Q111}$ cells were stably transfected with dCas13b with gRNA2 without ALKBH5, dCas13b-A5 combined with NT-gRNA (control) and dCas13b-A5 combined with gRNA2 constructs. After 24 h, the cells were treated with actinomycin D (Act-D, Catalog #A9415, Sigma, USA) at 10 µg/ml for 2, 4, 6 and 8 h. The cells were collected, and RNA was isolated for real-time PCR.

## 3'RACE

Amplification of poly(A) mRNA was performed using the 3'RACE System for Rapid Amplification of cDNA Ends (Invitrogen, ref. 18373-019) according to the manufacturer's instructions. First strand cDNA synthesis of mouse striatal MeRIP samples (IP and input control) was performed following the instructions for transcripts with high GC content. One hundred nanograms of the RNA of interest was reverse transcribed using the provided adapter primer (AP), followed by RNase H digestion to remove the template. The resulting cDNA was then amplified with a gene-specific primer (GSP) for $Htt1\alpha$ generated by the first cryptic poly(A) site (GSP-pA1) or the second cryptic poly(A) site (GSP-pA2). All PCRs were performed using Platinum II Hot-Start PCR Master Mix (Invitrogen, ref. 14000-013), containing 8 µL of the cDNA template, 0.8 µL of 10 µM primers, 8 µL of the Platinum GC Enhancer and 20 µL of the master mix. The thermocycler was set to the following cycling program: 1 cycle 94 °C for 3 min; 35 cycles 94 °C for 15 s, 60 °C for 15 s, 68 °C for 1 min 30 s; followed by cooling to 4 °C. Sequences of the primers used for 3'RACE are provided in Table EV7. Samples were run on a 2% agarose gel, bands of interest were extracted with the GeneJET Gel extraction Kit (Thermo Scientific, ref. K0691), and DNA was reamplified using the Platinum II Hot-Start PCR Master Mix and the corresponding GSP primers to ensure an optimal amplification product for SANGER sequencing. The amplification product was subjected to a PCR cleanup protocol using the ExoSAP-IT Express PCR Product Cleanup kit (Applied Biosystems, ref. 75001), followed by quantification on a Nanodrop 1000 spectrophotometer (Thermo Fisher Scientific). SANGER sequencing was performed on an ABI3730XL DNA Analyzer (Genomics Core Facility, Universitat Pompeu Fabra).

## Immunocytochemistry

CRISPR-dCas13b stably transfected ST$Hdh^{Q111/Q111}$ immortalized striatal cells were grown on coverslips in 24-well plates for 48 h and fixed for 10 min at room temperature, permeabilized for 10 min with 0.5% saponin in PBS and blocked with 15% horse serum in PBS. The cells were incubated with anti-phospho-histone H2AX (Ser139) (1:1000; Merck) or anti-ALKBH5 (1:2000; Sigma-Aldrich) and secondary antibody (Cy3, 1:500; Jackson ImmunoResearch). Nuclei were stained with DAPI. Single images were acquired digitally using a Leica Confocal Microscope SP5 with a 40× oil-immersion objective. The percentage of nuclei with ɣ-H2AX foci and the average number of ɣ-H2AX foci per cell were analyzed using the cell image analysis software CellProfiler. At least 20 images for each condition in three independent experiments were analyzed.

## Global m⁶A measurements

Total m$^6$A levels were assessed using the EpiQuik® m$^6$A RNA Methylation Quantification Kit (Epigentek®, cat no. P-9005). EpiQuik was performed according to the manufacturer's recommendations. Total RNA extracted from cells (300 ng) was bound to the provided strip wells, alongside with the negative and positive controls. The m$^6$A tagged RNA was then labelled using capture and detection antibodies, and the absorbance was read at 450 nm on a microplate spectrophotometer (Tecan Infinite 200 PRO reader) (m$^6$A is proportional to the Optical Density (OD) intensity). All samples were run in duplicate. Relative RNA methylation status was determined applying the formula provided by the kit.

## Measurement of ATP content

Total ATP content was assessed by CellTiter-Glo 2.0 Cell Viability Assay Kit (Promega) according to the manufacturer's instructions. Briefly, 96-well black multiwell plates were prepared with 2000 cells in culture medium. After 24h cells were equilibrated to room temperature for 30 min before addition of 100 μl premade CellTiter-Glo 2.0 reagent and incubated for 10 min followed by luminescence recording on a microplate spectrophotometer (Tecan Infinite 200 PRO reader). To normalize ATP measurements, DNA concentration was analyzed using CyQUANT Cell proliferation Assay Kit (Molecular Probes) following manufacturer's instructions.

## Statistical analysis

Raw data were processed using Microsoft Excel Office and transferred to GraphPad Prism version 8.0.2 for further analysis. The results are expressed as the mean ± SEM. Normal distribution was assessed with the Shapiro–Wilk test. For statistical analysis, unpaired Student's $t$ test (two-tailed) or one-way ANOVA was performed, and the appropriate post hoc tests were applied as indicated in the figure legends. A 95% confidence interval was used, considering differences statistically significant when $P < 0.05$. Pearson's correlation analysis was performed to analyze the correlation between the CAG repeats length and methylation ratio. Statistical analysis methods, sample sizes and $P$ values for each experiment are indicated in figure legends.

# Data availability

This study includes no data deposited in external repositories.

The source data of this paper are collected in the following database record: biostudies:S-SCDT-10_1038-S44319-024-00283-7.

# Peer review information

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

## Acknowledgements

We are very grateful to Ana Lopez and Maria Teresa Muñoz for technical assistance, Dr Teresa Rodrigo and the staff of the animal care facility (Facultat de Psicologia Universitat de Barcelona). We acknowledge Dr. Eva Novoa and Ana Milanovic from Center of Genomic Regulation for basecalling and analysis of Direct RNA sequencing datasets for the detection of m6A modifications in the *Htt* gene. This work was supported by the Ministerio de Ciencia e Innovación (PID2020-116474RB-100 to VB, PID2020-113953RB-I00 to EM and RTI2018-094374-B100 to SG); Hereditary Disease Foundation grant to VB; National Institutes of Health grant to RMP (R01 NS126420); PhD grant program by La Generalitat de Catalunya (2018FI_B_00487) to AP; PhD grant from PID2020-116474RB-100 (RE2021-097199) to IR.

## Author contributions

**Anika Pupak**: Conceptualization; Data curation; Formal analysis; Investigation; Methodology; Writing—original draft. **Irene Rodriguez Navarro**: Data curation; Formal analysis; Validation; Investigation; Methodology. **Kirupa Sathasivam**: Resources; Investigation; Methodology. **Ankita Singh**: Data curation; Software; Formal analysis; Visualization; Methodology. **Amelie Essmann**: Data curation; Formal analysis; Investigation. **Daniel del Toro**: Supervision; Investigation; Methodology. **Silvia Ginés**: Resources; Investigation. **Ricardo Mouro Pinto**: Data curation; Formal analysis; Visualization; Methodology; Writing—review and editing. **Gillian P Bates**: Conceptualization; Resources; Supervision; Investigation; Writing—review and editing. **Ulf Andersson Vang Ørom**: Data curation; Formal analysis; Investigation; Writing—original draft. **Eulalia Marti**: Conceptualization; Resources; Formal analysis; Investigation; Writing—original draft; Writing—review and editing. **Verónica Brito**: Conceptualization; Resources; Data curation; Formal analysis; Supervision; Funding acquisition; Validation; Investigation; Visualization; Methodology; Writing—original draft; Writing—review and editing.

Source data underlying figure panels in this paper may have individual authorship assigned. Where available, figure panel/source data authorship is listed in the following database record: biostudies:S-SCDT-10_1038-S44319-024-00283-7.

## Disclosure and competing interests statement

The authors declare no competing interests.

# Expanded View Figures

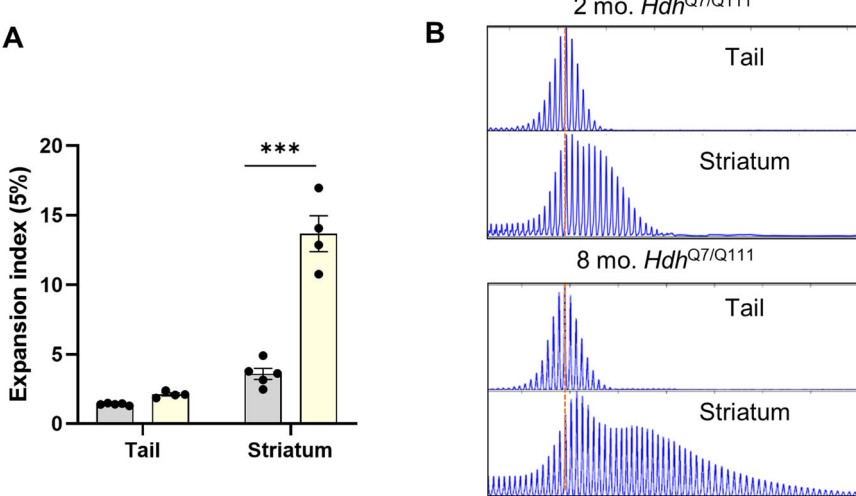

**Figure EV1. Comparison of somatic CAG repeat instability in 2 and 8 months old *Hdh*^+/*Qm* mice.**

(A) Quantification of somatic expansion indices, of *Htt* CAG PCR products from tails and striatum of *Hdh*^+/*Qm* at 2 and 8 months of age using a 5% peak height threshold. Error bars represent mean ± SEM; *n* = 4–5/age. Data were analyzed for each tissue using Student-T test, ***$P$ < 0.0001. (B) Representative GeneMapper traces showing somatic CAG repeat expansions in the striatum and tails at the different ages analyzed. Source data are available online for this figure.

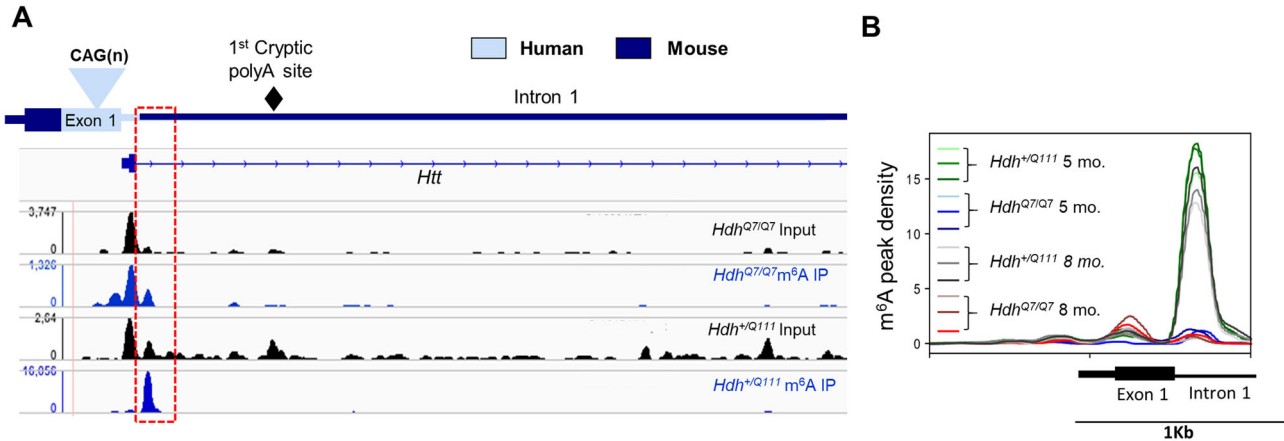

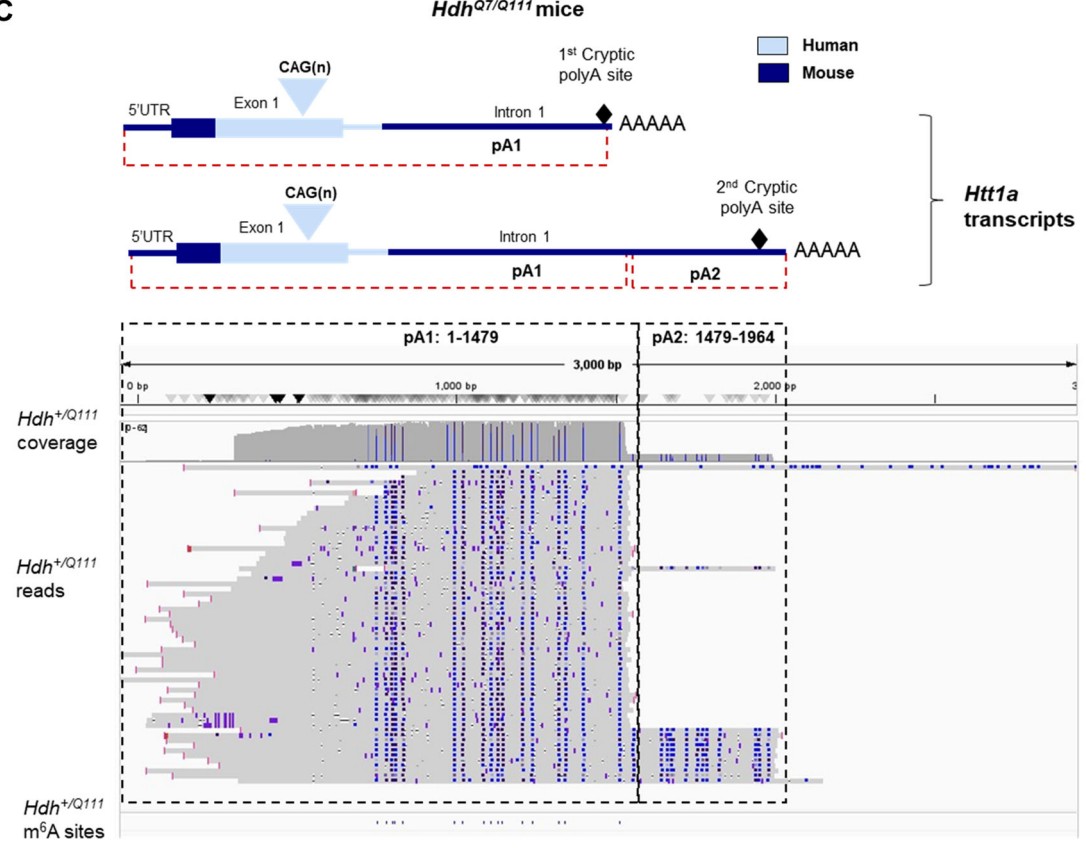

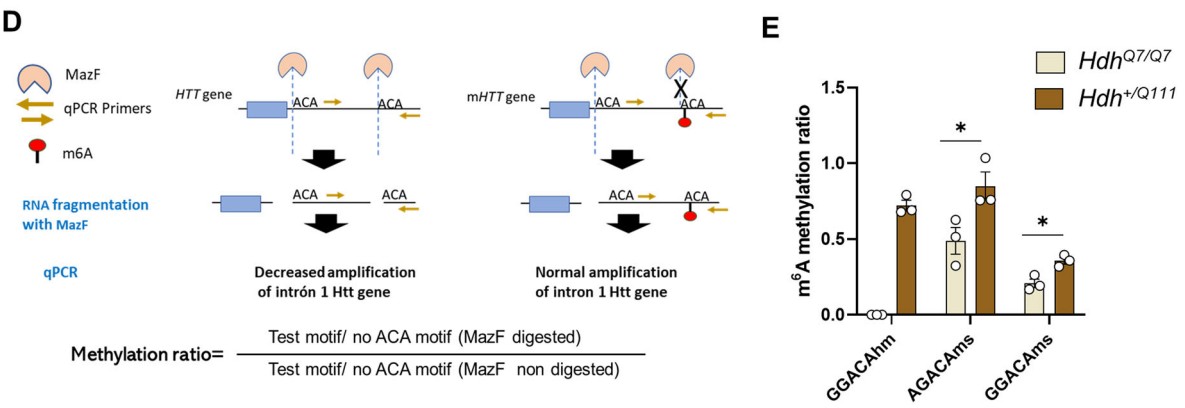

**Figure EV2.  m⁶A enrichment in *Htt* intron 1 of *Hdh*<sup>+/Q111</sup> mice.**

(A) Genome browser snapshots harboring m⁶A enrichment in the proximal region of *Htt* intron 1 to the 5′ exon1-intron 1 splice site. The sequence data, narrow Peak, and alignment data supporting the data is available NCBI GEO repository under the accession code GSE175618 (Pupak et al, 2022). (B) Comparison of fold enrichment distribution of methylation sites in the *Htt* intron 1 between 8- and 5-month old WT and *Hdh*<sup>+/Q111</sup> mice obtained by MeRIP-seq ($n = 3$/genotype/age, Pupak et al 2022). (C) Mapping of m⁶A sites in mutant *Htt1a* transcripts by direct RNA sequencing. IGV snapshot show m⁶A sites in *Htt1a* transcripts in the striatum of *Hdh*<sup>+/Q111</sup> mice. Mouse sequence (GRCmm39 genome) of the *Htt* gene including the human insert was used as reference gene (chr5: 34,919,088–35,070,342). Purple and blue dots represent high and low confidence m⁶A sites, respectively. (D) Schematic representation of the MazF-qPCR approach used for quantification of residue specific m⁶A methylation. MazF interfase enzyme only cuts at the ACA sequence when not methylated allowing for interrogation of specific m⁶A motifs and measurement of m⁶A ratio following formula shown in the figure. (E) Methylation ratio of three different m⁶A motifs obtained by MazF-qPCR analysis in the striatum of *Hdh*<sup>Q7/Q7</sup> and *Hdh*<sup>+/Q111</sup> ($n = 3$ mice/genotype). Data represent the mean ± SEM. Data were analyzed using Student-T test. *$P = 0.0416$ (AGACAms), *$P = 0.0144$ (GGACAms). Source data are available online for this figure.

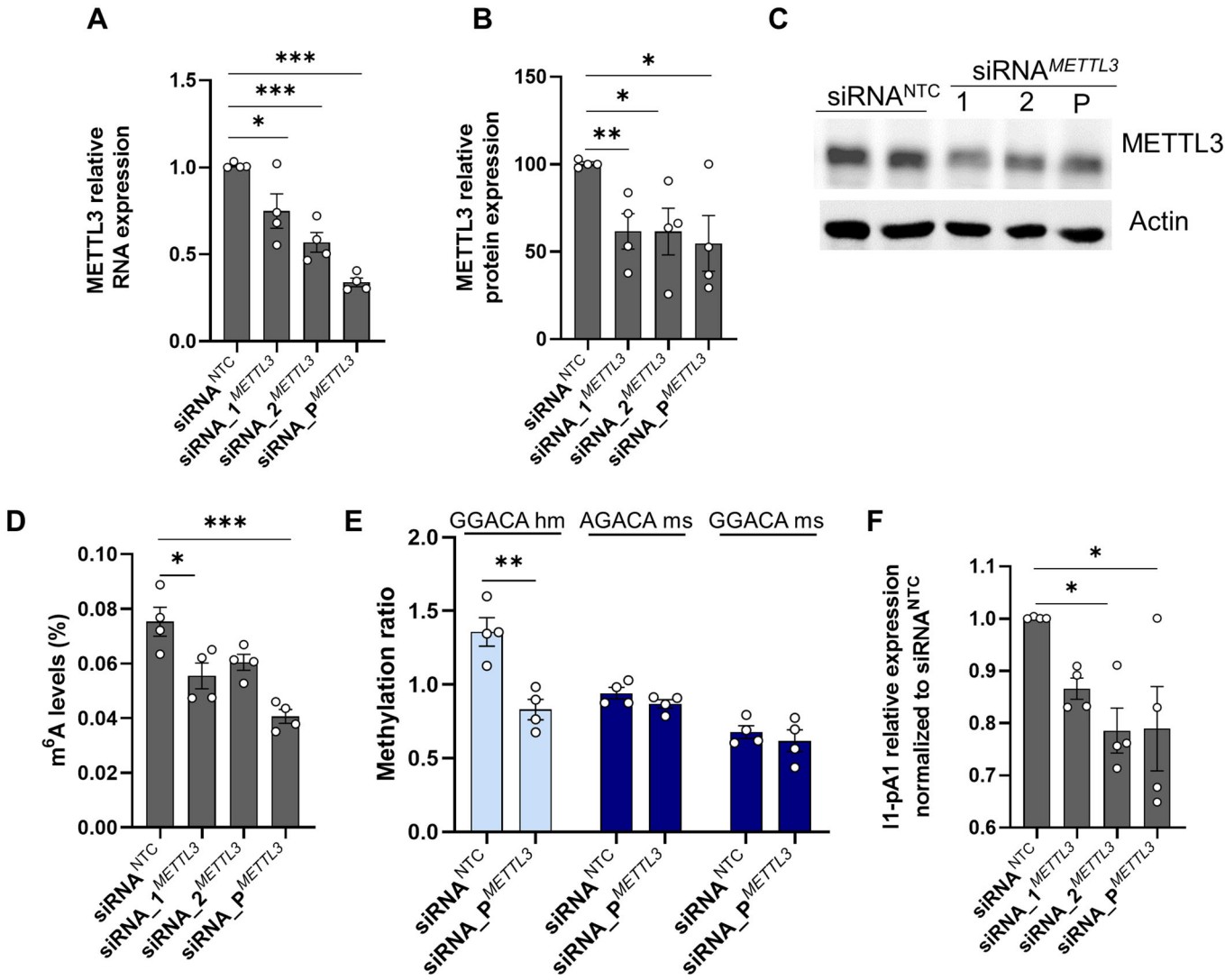

**Figure EV3. METTL3 knockdown with siRNA decreases I1-pA1 levels in STHdh$^{Q111/Q111}$ cells.**

METTL3 mRNA levels and protein expression levels analyzed by qPCR (A) and western blot (B) in STHdh$^{Q111/Q111}$ transfected for 24h with non-targeting control (siRNA$^{NTC}$), two different targeting sequences (siRNA_1$^{METTL3}$ and siRNA_2$^{METTL3}$) and a pool of 3 target-specific siRNA (siRNA_P$^{METTL3}$) against *METTL3*. (A) qPCR analysis of the expression levels of *METTL3* ($n = 3$–4 independent experiments; 2 technical replicates/experiment). Data represent the mean ± SEM. Data were analyzed using one-way ANOVA with Tukey´s multiple comparisons test. *$P = 0.0358$ (siRNA$^{NTC}$ vs, siRNA_1$^{METTL3}$), ***$P = 0.0009$ (siRNA$^{NTC}$ vs, siRNA_2$^{METTL3}$) and ***$P < 0.0001$ (siRNA$^{NTC}$ vs, siRNA_P$^{METTL3}$). (B) Western Blot analysis of the protein expression levels of METTL3. Data represent the mean ± SEM. Data were analyzed using Student-T test. **$P = 0.0091$ (siRNA$^{NTC}$ vs, siRNA_1$^{METTL3}$), *$P = 0.027$ (siRNA$^{NTC}$ vs, siRNA_2$^{METTL3}$) and *$P = 0.028$ (siRNA$^{NTC}$ vs, siRNA_P$^{METTL3}$). (C) Representative western blots showing expression of METTL3 and actin used as loading control. (D) Overall m$^6$A levels were measured using EpiQuik m$^6$A RNA Methylation Quantification Kit in STHdh$^{Q111/Q111}$ cells. Histograms show percentage of m$^6$A levels in total RNA ($n = 3$–4 independent experiments; 2 technical replicates/experiment). Data were analyzed using one-way ANOVA with Tukey´s multiple comparisons test. *$P = 0.0207$ (siRNA$^{NTC}$ vs, siRNA_1$^{METTL3}$), ***$P = 0.0003$ (siRNA$^{NTC}$ vs, siRNA_P$^{METTL3}$). (E) MazF-qPCR analysis of the DRACH motifs in *Htt* intron 1 ($n = 4$ independent experiments) in STHdh$^{Q111/Q111}$ transfected for 24 h with non-targeting control (NTC) and a pool of 3 target-specific siRNA (siRNA_P$^{METTL3}$) against METTL3. Data represent the mean ± SEM. Data were analyzed using Student-T test. **$P = 0.0043$ (siRNA$^{NTC}$ vs, siRNA_P$^{METTL3}$). (F) qPCR analysis of the I1-pA1 *Htt* transcript in STHdh$^{Q111/Q111}$ transfected for 24h with non-targeting control (NTC), two different targeting sequences (siRNA_1$^{METTL3}$ and siRNA_2$^{METTL3}$) and a pool of 3 target-specific siRNA (siRNA_P$^{METTL3}$) against METTL3 ($n = 4$ independent experiments). Data represent the mean ± SEM. Data were analyzed using one-way ANOVA with Tukey´s multiple comparisons test. *$P = 0.0309$, **$P = 0.0339$ compared with cells treated with siRNA$^{NTC}$. Source data are available online for this figure.

