## [Peer Review File · EMBO Reports]

m6A modification of mutant huntingtin RNA promotes the biogenesis of pathogenic huntingtin transcripts

Anika Pupak, Irene Navarro, Kirupa Sathasivam, Ankita Singh, Amelie Essmann, Daniel del Toro, Silvia Gines, Ricardo Pinto, Gillian Bates, Ulf Vang Orom, Eulalia Marti, and Veronica Brito

Corresponding author(s): Veronica Brito (veronica.brito@ub.edu)

Review Timeline:

Submission Date:	11th Dec 23
Editorial Decision:	23rd Jan 24
Revision Received:	31st Jul 24
Editorial Decision:	5th Sep 24
Revision Received:	20th Sep 24
Accepted:	27th Sep 24

Editor: Esther Schnapp

Transaction Report:

Dear Dr. Brito,

Thank you for the submission of your manuscript to EMBO reports. We have now received the full set of referee reports that is pasted below.

As you will see, the referees acknowledge that the findings are potentially interesting. However, they also point out that the data should be strengthened, and they have several suggestions for how this can be done. I think that all suggestions are good and should be addressed. Please let me know in case you disagree and we can discuss the exact revision requirements further, also in a video chat, if you like.

I would thus like to invite you to revise your manuscript with the understanding that the referee concerns must be fully addressed and their suggestions taken on board. Please address all referee concerns in a complete point-by-point response. Acceptance of the manuscript will depend on a positive outcome of a second round of review. It is EMBO reports policy to allow a single round of major revision only and acceptance or rejection of the manuscript will therefore depend on the completeness of your responses included in the next, final version of the manuscript.

We realize that it is difficult to revise to a specific deadline. In the interest of protecting the conceptual advance provided by the work, we recommend a revision within 3 months (24th Apr 2024). Please discuss the revision progress ahead of this time with the editor if you require more time to complete the revisions.

- 1) A data availability section providing access to data deposited in public databases is missing. If you have not deposited any data, please add a sentence to the data availability section that explains that.
- 2) Your manuscript contains statistics and error bars based on $n=2$. Please use scatter blots in these cases. No statistics should be calculated if $n=2$.

3) We replaced Supplementary Information with Expanded View (EV) Figures and Tables that are collapsible/expandable online. A maximum of 5 EV Figures can be typeset. EV Figures should be cited as 'Figure EV1, Figure EV2' etc... in the text and their respective legends should be included in the main text after the legends of regular figures.

5) a complete author checklist, which you can download from our author guidelines <https://www.embopress.org/page/journal/14693178/authorguide>. Please insert information in the checklist that is also reflected in the manuscript. The completed author checklist will also be part of the RPF.

6) Please note that all corresponding authors are required to supply an ORCID ID for their name upon submission of a revised manuscript (<https://orcid.org/>). Please find instructions on how to link your ORCID ID to your account in our manuscript tracking system in our Author guidelines <https://www.embopress.org/page/journal/14693178/authorguide#authorshipguidelines>

I look forward to seeing a revised form of your manuscript when it is ready.

Esther Schnapp, PhD

Referee #1:

Anika Pupak and colleagues presents an interesting extension of their initial characterization of m6A modification of mutant huntingtin RNA and the biogenesis of the highly-toxic Htt1a fragment in the context of Huntington's Disease (HD) pathology. The paper is testing a relevant biological process, which definitely warrants further characterization. One other point raised by the paper relates to the possibility to exploit these findings in search for possible therapeutic strategies for Huntington's Disease pathology.

There are few important issues which limit the enthusiasm for this work and challenge the paper's main findings.

Main concerns:

1. Relevance of the findings for human HD patients: alleles with lower CAGs. Most of the work is conducted in mouse animal models with very extended CAG tracts. While these model systems are useful to study the mechanism of pathology and, in general, replicate the human HD genetic mutation (no over-expression of short fragments and main use of heterozygous animals), nevertheless all the models are characterized by very extended CAG-tract (Striatal cells and Q111 mice have 111 CAGs, zQ175 around 180/200 CAGs and YAC128, roughly 125 CAGs). However, these situation does not faithfully recapitulate the distribution of the human HD alleles, the majority of which has between 40 and 50 CAG expansions.

Do the authors see any correlation between CAG expansion in the human samples and the methylation ratio? How do the authors explain the correlation with HD stage of pathology? Is that mediated by CAG length? Or CAG somatic expansion?

What are the levels of HTT1a in the brain (putamen) and primary fibroblasts samples presented in figure 3?

Independent knock-in lines presenting CAG-expansion in a more physiologic range (knock-in Q50) should be included to establish the relevance of this phenotype for i. the HD patients and ii. 'lower expanded' alleles.

2. Correlation with somatic instability: the authors propose that the increase in m6A modification and, in turn, Htt1a production might correlate with CAG-size and somatic instability *in vivo*.

This is a fascinating hypothesis which is not tested in this study. Adding information about CAG size in the striatum and/or cortex of the different animal models and at different time-points would add important information and reinforce the correlation.

Also, altering the expression levels of crucial players of somatic instability such as Fan1 or Pms2 could add information about the cross correlation between m6A modification, somatic instability and Htt1a biogenesis.

3. Target demethylation of Htt intron 1 and HD phenotypes modulation: while the reduction of phosphorylation of histone H2AX following target demethylation of m6A site at Htt intron 1 is interesting, several other -cellular and molecular - phenotypes have been described in striatal Q111 cells. How those phenotypes respond to the demethylation treatment

Moderate concern:

Reduction of FLHTT transcript expression usually correlates with somatic CAG expansion, however, in cultured cells, STHdhQ111/Q111 *in vitro* model, this somatic CAG expansion is thought to be almost negligible:

> How do the authors explain their results presented in suppl. Fig 1?

> How does the speed of PolIII correlate with CAG expansion and Htt1a biogenesis?

M6A methylation site in human HTT. This was found following the prediction from the mouse data: a dedicated approach to dissect M6A methylation in human would be recommended. Is this changing in different samples with different CAGs? And in different brain regions?

A better control of the MeRIP method would be suggested. Additional quality controls of the experiments, adding negative and positive controls with regions certainly methylated and not methylated, would strengthen the solidity of the data.

Referee #2:

The proposed manuscript by Pupak et al. addresses a fundamental question about the potential function of m6A RNA modifications in regulating huntingtin (htt) mRNA processing and its involvement in Huntington's disease. In this manuscript, the authors provide evidence using mouse models, human and mouse cell culture models and post-mortem brain tissue that m6A modifications accumulate in Intron 1 of the Htt gene and could participate in the generation of aberrant splicing isoforms that are deleterious. For that they have used notably an elegant RNA editing method to avoid indirect effects. They establish a correlation between m6A methylation of the htt1a gene intron and increased DNA damage, a factor contributing to pathogenesis. Finally, the use of locked nucleic acid-modified antisense oligos, targeting CAG repeats in exon 1, showed that this expansion of CAG repeats is involved in the regulation of m6A methylation in intron 1. Overall, the results provided are very interesting and open up new strategies for the treatment of the disease.

However, I have a few points that would need clarifications:

1) In figure 1: I find a little difficult to draw conclusions about the correlation between m6A enrichment and the expression of the spliced isoforms particularly I1-pA2 at 2 months. The sharp increase of I1-pA2 transcripts does not appear to be due to m6A accumulation. Moreover wouldn't we expect a decrease in the FL transcript expression here?

2) in figure 4: I'm also surprised that in YACs128 MEFs at 10uM of inhibitor there is no significant decrease of GGACA

methylation while you observe a strong decrease of I1-pA1 transcripts again going against a direct link between methylation and increased htt1a. It may also suggest that other sites are more important than those analyzed. This leads us to criticize the fact that there is no initial mapping of methylated sites along htt transcripts using alternative methods, perhaps calling into question the choice of sites tested.

3) It is also difficult to interpret the relatively low effect on GGACA and the lack of effect of the METTL3/METTL14 inhibitor on the methylation status of other sites. It would be necessary to compare these experiments for example after treatment with siRNA against METTL3/METTL14. A more global validation of the inhibitory effect of STM2457 on m6A levels in selected cell lines would also be required.

4) Relative to Figure5: Regarding DNA damage experiments with Y-H2AX, I'd also like to see controls other than NT-gRNA to avoid the possibility that this NT-gRNA context might somehow increase DNA damage, affecting the overall conclusion.

5) Figure6: An obvious missing experiment that could add mechanistic value would be to test whether blocking CAG repeats affects METTL3 recruitment or function. In other words, can the length of CAG repeats promote METTL3 recruitment?

Minor comments:

- Problem with links to Fig5 in the text.
- It would be interesting to describe the rationale of the choice of the position of gRNAs.
- Only males have been analyzed. Are there any sex differences in pathogenicity?
- sentence 388-389 repetition of m6A not needed.
- Paragraph 396-401 should be rephrased.
- Ref24 not cited at the right place in introduction

Referee #3:

In this manuscript, Pupak et al show that the RNA modification N6-methyladenosine (m6A) is enriched in HTT RNA 5' to a cryptic poly (A) site that affects the production of HTT1a in HD mouse model striatal cells and postmortem human brain tissue. The authors identified these m6A sites from previous studies, and here the group provides strong and convincing evidence that modulation of m6A levels alters the methylation status of 1 of 3 m6a sites and as a result can regulate HTT1a levels. The authors first used a global approach (STM2457 inhibition of Mettl3) to lower m6A levels, followed by a targeted approach by fusing the m6A eraser ALKBH5 to dCas13b with guideRNAs targeting regions near the m6A sites. The authors then showed that reduction of m6A at one of the sites reduces DNA damage. The results are convincing and have identified a novel approach to modulate the toxic HTT1a levels which provides therapeutic potential. Overall the findings presented in this study will provide the field with insights into the generation of the toxic HTT1a RNA fragment through RNA processing in a m6A-dependent manner.

Concerns:

Line 320-321 / Figure 1A: I3 and FL probes were described but not labeled properly in text as in the figure, we recommend that I3/FL be added for consistency.

Line 325 / Figure 1B: "The levels of intronic sequences I1-3' were comparable to intron 3;" the graph in figure 1B is convincing, however were there any statistical tests carried out for this?

Line 356: We suggest the following text changes: Supplementary Figure 2 Supplementary Figure 2a & 2b.

Line 356-359: Can the authors clarify how the two additional m6A motifs were selected, Supplementary figure 2 only shows enrichment for the first m6A site.

Line 399-401 / Supplementary Figure 3: Text should be modified to contain 3A and 3B where appropriate.

Line 438 / Figure 5D and 5E?: Missing reference to Figure 5D in text. Description of Figure 5E is missing from text.

Figure 5G: the lighter and darker grey color is hard to see, may not be that important since there were no difference in RNA decay between conditions.

Figure 5H, 5I and 5J: 5H & 5I Graphs are missing data points. 5J can benefit from insets showing differences of gamma H2AX foci between the two conditions.

Supplementary Figure 6: A & B matches Figure 6 A & B and is not referenced in text. 6A can benefit from larger texts. 6B can benefit from mentioning of top and bottom.

Line 483: From figure 6A schematic, HTT-e1f and HTT-e1r are primers that are outside the CAG repeats and HTT-e1*f and HTT-e1*r are primers spanning the CAG, the text mentions "PCR with primers amplifying Exon 1 outside the CAG repeat (HTT-e1*)..." can this be clarified.

Line 485: "However,..." can be removed.

Line 363-366 / Figure 2B: It would be useful to the field to understand if the methylation preference pattern between the 3 m6A sites are altered between HD and Controls, since the authors earlier showed (Figure 1) that m6A is increased in HD at I1-pA1.

Line 396-399 / 403-405: Can the authors hypothesize why decreasing I1-pA1 levels does not rescue FL-Htt levels? Or relate to previous findings around levels of the exon 1 protein versus FL.

Figure 6: from Figure 6C, transfected cells with LNA-CTG show decreased transcript levels of HTT1a which would be expected when blocking the CAG expansion. Can the authors clarify or rule out the possibility that the significant decrease in the m6A methylation ratio seen in Figure 6D is not simply due to lower HTT1a levels caused by LNA-CTG through an m6A-independent manner. From the citation [34] of LNA-CTG used, it seems like LNA-CTG will decrease mHTT, and if m6A is present more so on the mHTT transcript, then decreasing it will likely decrease m6A levels measured by the assays presented in this study. There could be multiple interpretations and as the section stands, it is overinterpreted.

Dr. Esther Schnapp
Editor-in-Chief
EMBO reports

Barcelona, July 31st 2024

Dear Dr. Esther Schnapp,

We would like to submit a revised version of our manuscript "**m⁶A modification of mutant huntingtin RNA regulates the biogenesis of pathogenic huntingtin transcripts**" by Pupak et al. for consideration at *EMBO reports*. We thank the Reviewers for their thoughtful comments. We have addressed each of the points and performed new experiments, as detailed below, strengthening the paper. We hope that these changes permit acceptance of the manuscript.

REVIEWER 1

Main Concerns

1. Relevance of the findings for human HD patients: alleles with lower CAGs. Most of the work is conducted in mouse animal models with very extended CAG tracts. While these model systems are useful to study the mechanism of pathology and, in general, replicate the human HD genetic mutation (no over-expression of short fragments and main use of heterozygous animals), nevertheless all the models are characterized by very extended CAG-tract (Striatal cells and Q111 mice have 111 CAGs, zQ175 around 180/200 CAGs and YAC128, roughly 125 CAGs). However, these situation does not faithfully recapitulate the distribution of the human HD alleles, the majority of which has between 40 and 50 CAG expansions.

We acknowledge the reviewer's concern. However, we would like to point out that while most adult onset HD patients inherit HD alleles with around 40-50 CAG repeat, that does not reflect the actual length of the CAG tract that is found in the human brain where CAGs expand to 100 to 500+ units, which is a CAG length comparable to the model systems used in this study. This is supported by several evidences showing that, CAG repeat is unstable in both somatic and germ cells (Duyao et al 1993; doi: 10.1038/ng0893-387, Telenius et al; 1994 <https://doi.org/10.1038/ng0494-409>). For instance, expansions of the inherited mHTT allele with very long CAG tracts (up to approximately 1000 CAGs) have been observed sporadically in some patient's brain structures, including the caudate nucleus and putamen (Kennedy et al 2003, doi: 10.1093/hmg/ddg352). Importantly, a causal role for these somatic expansions of the CAG repeat in HD pathogenesis is further supported by findings from a genome-wide association study looking for genetic modifiers of HD motor symptom onset other than CAG tract length itself (Genetic Modifiers of Huntington's Disease (GeM-HD) Consortium, 2019, 2024 <https://doi.org/10.1016/j.cell.2019.06.036>; <https://doi.org/10.1101/2024.06.10.597797>). Moreover, recent analysis of genomic DNA of different cell types from donors carrying most prevalent disease-causing CAG tract lengths (from 42 to 45 uninterrupted CAGs) revealed that only a small fraction of medium spiny neurons had mHTT copies with the original inherited CAG tract length, and approximately half of these neurons had CAG tracts that were expanded by more than 20 repeat units (Mätlik. et al 2023, <https://doi.org/10.1038/s41588-024-01653-6>, Handsacker et al 2024, <https://doi.org/10.1101/2024.05.17.592722>). According to the last

evidence, significant neuronal dysfunction in MSNs seems to begin when the CAG repeat number surpasses 150-180. Therefore, using HD models within this CAG range is a reasonable approach to address our hypothesis

In our study we demonstrate the relevance of our findings in the human context by evaluating *HTT*-intron 1 methylation in HD fibroblasts as well as in postmortem putamen samples (Fig. 3). Most of the human samples used in our study are from patients with germline human HD alleles around 40-56 CAG repeats. We observed an increase in m⁶A methylation in *HTT* intron 1 suggesting that the methylation in *Htt/HTT* intron 1 is conserved between the HD models and human *HTT* transcripts.

According to this concern we have added further discussion on the rational of using HD models with long CAG repeat length in the Discussion section (**Page 29, lines 745-752**)

-Do the authors see any correlation between CAG expansion in the human samples and the methylation ratio?

To address the reviewer concern we have analyzed the correlation between methylation ratio and CAG repeat length in HD-adult fibroblasts with 40-56 CAG repeats. No significant correlation was observed initially between methylation ratio and CAG repeat length. However, evaluating methylation ratio in two HD-juvenile included in the study for this revision (GM9197 with 180 CAG repeats and GM4281 with 80 CAGs), the analysis revealed a significant correlation (**Appendix Figure S4B**). These results are now described in the Results section (**Page 20, lines 518-525**)

For HD putamen postmortem samples, CAG repeat information wasn't available for all cases. Therefore, to perform correlation analysis we estimated repeat sizes using fragment analysis. We found no correlation between methylation ratio and CAG repeats length. Given the limited number of samples, we believe a categorical analysis of methylation ratio across different CAG repeat length ranges might be more appropriate. Therefore, we're excluding this analysis from the manuscript to avoid suggesting a lack of correlation where the study might be underpowered. Nevertheless, we include the data here in the rebuttal for the reviewer's consideration. Similarly, a preliminary analysis of correlation with metrics of somatic instability did not reveal any overt relationship. While this association with somatic CAG expansions is an interesting hypothesis, this would need to be better addressed in future studies with a significantly larger sample size than what is currently available to us at this moment.

Correlation analysis between CAG repeats length and methylation ratio in GGACA site in HD putamen postmortem samples using CAG sizes determined by fragment analysis.

-How do the authors explain the correlation with HD stage of pathology? Is that mediated by CAG length? Or CAG somatic expansion?

We did detect an increase in methylation ratio at the GGACA site in the putamen samples from Vonsattel grades (2-3, and 3) but the pathological grades are not sufficiently well defined and the numbers of samples are too small to draw conclusions about m⁶A accumulation and disease progression from these data. To avoid an overinterpretation of these results we have removed the sentence “*This result leads us to suggest that aberrant accumulation of m⁶A with disease progression could play a role in worsening HD symptomatology*” in the last paragraph of Results section and replaced by “*Overall, these findings indicate the potential relevance and possible pathological significance of m⁶A methylation in mHtt RNA for HD patients*”. (Page 20, lines 528-530).

As previously explained our data with human samples does not allow to drive conclusions on a correlation between methylation, HD stage of pathology or somatic CAG expansions. However, in our study we demonstrated that methylation in mHtt is related to CAG expansion (**new Fig. 6**) suggesting that in the brain, CAG expansions drive methylation which could contribute to *HTT1a* production. This hypothesis is supported by recent evidences showing that CAG somatic expansions are a critical first step in HD pathogenesis which lead to transcriptional dysregulation (Mätlik. et al 2023, <https://doi.org/10.1038/s41588-024-01653-6>). Further discussion, addressing the reviewer question have been added and can be found in Discussion section (Page 29, lines 755-760)

It is worth mentioning that postmortem putamen samples reflect a snapshot of multiple pathogenic processes occurring simultaneously in MSNs; and contain cells with diverse CAG expansions because of somatic instability. In HD brain samples it is also possible that two processes are overlapping, making it difficult to establish good correlations. That is, the degree of methylation might change both in progression and in a manner dependent on the CAG repeats. This, combined with somatic instability, would complicates the assessment of whether the degree of methylation correlates with progression and/or the number of expansions. However, as mentioned before, it is worth to explore in the future with a larger sample size.

-What are the levels of *HTT1a* in the brain (putamen) and primary fibroblasts samples presented in figure 3?

To answer the reviewer's question, we measured *HTT1a* levels in HD putamen samples alongside HD human fibroblasts (**Appendix Fig. S5A and S6B**). *HTT1a* increased in juvenile-onset HD fibroblasts (≥80 CAG repeats) compared to controls while a high level of variability was observed in the rest of the samples. Putamen samples from adult-onset HD revealed no significant differences in the expression of *HTT1a* compared with controls. This confirms previous findings of *HTT1a* elevation with longer CAG tracts (≥50 repeats) (Neueder et al. 2017, doi: 10.1038/s41598-017-01510-z; Hoschek et al. 2024, doi: 10.1186/s10020-024-00801-2). Our samples ranged from 40-49 CAG repeats, except for one fibroblast line with 56 repeats and two juvenile lines (GM4281, GM9197) with 80 and 180 CAG repeats. In postmortem putamen sampled one cannot expect highly elevated levels of *HTT1a* based on the analysis of bulk RNA levels. Extensive somatic CAG expansion in HD brains seems to be present in only select neuronal subpopulations like striatal medium spiny neurons or the Purkinje neurons of the cerebellum (Matlik et al. 2023, <https://doi.org/10.1038/s41588-024-01653-6>; Handsacker et al 2024, <https://doi.org/10.1101/2024.05.17.592722>). Additionally, it is worth noting that the current

understanding is that cells with hyper-expanded CAGs are fairly short lived (i.e neuronal loss within months upon expanding beyond 150-180 CAGs) and therefore never become preponderant.

However, our methylation assay has been sensitive enough to detected in total RNA increased methylation of *mHtt* intron 1. Given that the abundance of *HTT1a* detected in fibroblasts or putamen HD was not different compared to controls, it is possible that m⁶A methylation is mainly detected in a pre-mRNA that has not yet been completely spliced further supporting the hypothesis that this methylation is deposited in the pre-mRNA and plays a role in the generation of *HTT1a* when longer CAG repeats are present in the huntingtin gene, which is what we observe in the HD juvenile fibroblasts, *Hdh*^{+Q111} mice and HD cell lines.

We included the levels of *HTT1a* in **new Appendix Fig. S5** and describe results and conclusions in Results section (**Page 20, lines 520-530**)

- Independent knock-in lines presenting CAG-expansion in a more physiologic range (knock-in Q50) should be included to establish the relevance of this phenotype for i. the HD patients and ii. 'lower expanded' alleles.

We thank the reviewer suggestion; however, we believe that knock-in lines presenting CAG-expansion in a more physiological range are not able to establish the relevance of this phenotype. The levels of *Htt1a* detected in mice such as Q50 are extremely low (*Landles et al 2024*; <https://doi.org/10.1101/2024.06.11.598410>) and somatic expansion does not occur in these mice during their lifetime. Therefore, these mice could not be used to address the main question of our study regarding the role of m⁶A in *Htt1a* RNA biogenesis. As discussed above, it is now widely accepted, from the GWAS studies, that it is the somatic expansion of the CAG repeat that drives the age of onset and rate of disease progression for HD, and *Htt1a* is generated in a CAG expansion-dependent manner.

As mentioned before, to establish the relevance of the observed methylation in *mHtt* RNA for HD patients our study shows an analysis of m⁶A methylation in samples from HD patients (postmortem putamen tissue and fibroblasts) suggesting that the methylation pattern may be linked to disease progression, highlighting its potential pathological relevance

2. Correlation with somatic instability: the authors propose that the increase in m6A modification and, in turn, Htt1a production might correlate with CAG-size and somatic instability in vivo. This is a fascinating hypothesis which is not tested in this study. Adding information about CAG size in the striatum and/or cortex of the different animal models and at different time-points would add important information and reinforce the correlation.

Also, altering the expression levels of crucial players of somatic instability such as Fan1 or Pms2 could add information about the cross correlation between m6A modification, somatic instability and Htt1a biogenesis.

We agree with the reviewer that a correlation between CAG size, somatic instability, increased m⁶A modification, and subsequent *Htt1a* production is an intriguing hypothesis for further investigation. Validating this hypothesis, as the reviewer correctly points out, would require *in vivo* manipulation of key players in somatic instability, which falls outside the scope of this study.

Our primary aim here is to demonstrate, for the first time, that *Htt* intron 1 is methylated and contributes to *Htt1a* biogenesis.

Nevertheless, as suggested by the reviewer we added in a **Figure EV1**, information about CAG size in the striatum from *Hdh*^{+/*Q111*} at 2 different ages (2 and 8 months old); the HD mouse model in which we demonstrated by MeRIP-qPCR and RACE-PCR the m⁶A enrichment in *I1-PA1* and *I1-PA2* transcripts (Figure 1f, 1g, 1h and 1i). Consistent with previous findings, (Lee and Pinto et al., 2011 doi: 10.1371/journal.pone.0023647) our somatic expansions index quantification revealed a marked increase in repeat instability at 8 months. Interestingly, m⁶A enrichment in *Htt1a* transcripts also peaked at 8 months when higher somatic instability can be observed. However, we clarify that these observations do not establish a causality and further experiments should be conducted to investigate the role of somatic instability on m⁶A methylation in *Htt* intron 1. These new results are described in Results section (**Page 17, lines 441-447**).

3. Target demethylation of *Htt* intron 1 and HD phenotypes modulation: while the reduction of phosphorylation of histone H2AX following target demethylation of m⁶A site at *Htt* intron 1 is interesting, several other -cellular and molecular - phenotypes have been described in striatal Q111 cells. How those phenotypes respond to the demethylation treatment.

As suggested by the reviewer, we explored deeper in the effect of the demethylation treatment in other aspects of the phenotype such as the impaired energy metabolism described in the *STHdh*^{Q111/Q111} (Gines et al 2003, doi: 10.1093/hmg/ddg046. Lim et al 2018, doi: 10.1074/jbc.M704704200). We analyzed ATP generation using CellTiter-Glo[®] 2.0 Assay, a luciferase-based assay in which luminescence values are in direct proportion to ATP levels. Since this cell line proliferate, we normalized ATP measurements using a CyQUANT Cell proliferation Assay that measures DNA concentration. Our results show that target demethylation with dCas13-gRNA2 resulted in a moderate but significant increase in ATP levels compared with both controls (NT-gRNA and gRNA2).

We added this results in **new Fig. 5H** and the corresponding description has been included in Result section (**Page 25, lines 640-645**)

Moderate concern:

-Reduction of FLHTT transcript expression usually correlates with somatic CAG expansion, however, in cultured cells, STHdhQ111/Q111 in vitro model, this somatic CAG expansion is thought to be almost negligible:

> How do the authors explain their results presented in suppl. Fig 1?

Please, note that **Supplementary Fig. 1 is now Appendix Fig. S2**.

We agree with the reviewer that the reduction of FL-*Htt* transcript expression usually correlates with CAG length particularly in CAG140 and zQ175 mice according to transcriptome analysis of an HD allelic series conducted in knock-in mice (Langfelder et al 2016, doi: 10.1038/nn.4256) and a recent study by Landles et al (2024; <https://doi.org/10.1101/2024.06.11.598410>).

Nevertheless, all these findings are in adult mice and cannot be compared with the *STHdh*^{Q111/Q111} cells which is an immortalized embryonic striatal cell model of HD.

We agree with the reviewer that CAG somatic expansion in these cells is negligible. However, the detection of *Htt1a* levels suggests that the decrease in FL-*Htt* transcripts from both mutant alleles (carrying 111 CAG repeats) in these cells is a consequence of incomplete splicing of mHtt pre-mRNA between exon 1 and exon. For reviewer consideration we include in this rebuttal, data of three different experiments showing that FL-*Htt* transcripts are decreased in *STHdh*^{Q111/Q111} compared with the two splice forms I1-pA1 and I1-pA2. Nevertheless, this dramatic reduction of FL-*Htt* should be taken with caution when compared with WT cells since we cannot discard the possibility of immortalization artifacts impacting the levels of FL-*Htt* expression. As previously reported, *STHdh* cell lines exhibit divergent characteristics, which can hamper the direct comparison of WT and Q111 cell lines (Singer et al 2017., <https://doi.org/10.1038/s41598-017-17275-4>). However, our study mainly analyzed transcripts expression in mutant cells and is focused on the expression of I1-pA1 and I1-pA2 sequences which are barely detected in WT cells.

> How does the speed of PolII correlate with CAG expansion and Htt1a biogenesis?

The model described by Neueder et. al (2018, <https://doi.org/10.1038/s41467-018-06281-3>) suggests that expanded CAG repeats within the *HTT* gene can form RNA:DNA hybrids that impede or slow down Pol II elongation. This kinetic disruption can lead to alternative splicing and polyadenylation within *HTT* intron 1, promoting *Htt1a* biogenesis. We have better described this mechanism in the Background section (Page 3; lines 75-80), as well as in Discussion section (Page 31-32, lines 822-830).

Interestingly, this model also proposes that several elements in intron 1 can pause or stall Pol II at these sites, affecting the transcriptional process and favoring the recognition of cryptic poly(A) sites in intron 1, leading to the generation of *HTT1a*. This finding is in line with our hypothesis that Pol II can be influenced by other elements such as the m⁶A modification in *Htt* intron 1. Previous studies have shown that m⁶A can slow down the kinetics of mRNA processing when deposited in introns (Louloupi et al., doi: 10.1016/j.celrep.2018.05.077), control pausing of Pol II (Zhou et al., 2019, doi: 10.1016/j.molcel.2019.07.005; Akhtar et al., 2021, <https://doi.org/10.1016/j.molcel.2021.06.023>), and impact transcription termination (Yang et al., 2019, <https://doi.org/10.1038/s41422-019-0235-7>). Therefore, we propose that during transcription elongation, pausing or slowing down of Pol II can also be affected by m⁶A modifications within *Htt* intron 1. This has been further discussed on Page 32.

-M6A methylation site in human HTT. This was found following the prediction from the mouse data: a dedicated approach to dissect M6A methylation in human would be recommended.

We agree with the reviewer that a “*dedicated approach to dissect M6A methylation in human would be recommended*”. Ideally, a full investigation of human *HTT* RNA m6A methylation would be performed using both unbiased discovery high-throughput methods and targeted validation/quantification methodologies. In this study, we have leveraged our best knowledge of the potentially most relevant site and performed a sensitive and accurate quantification at that site, which we believe provides compelling evidence that m6A methylation is altered in the intronic region of *HTT* RNA in HD. Here to confirm that the proximal region of *HTT* intron 1 was also methylated in human, we have used the MazF-qPCR assay targeting a site -GGACA- present in the human intron region that is also present in *Hdh*^{Q7/Q111} mice. Using this MazF-qPCR approach we have demonstrated that *HTT* intron 1 RNA methylation levels were increased in HD patient cells and post-mortem samples. The effectiveness of the MazF-qPCR approach in identifying m6A modifications within specific RNA sequences has been well demonstrated by two independent and highly cited studies (*Garcia Campos et al 2019* <https://doi.org/10.1016/j.cell.2019.06.013>, *Yang et al 2019* doi: 10.1126/sciadv.aax0250). They demonstrated that this method is highly sensitive and specific for detecting m6A sites, providing a robust tool for studying RNA methylation dynamics. By employing this approach, it is possible to confidently attribute changes in m6A levels to actual methylation differences rather than variations in RNA abundance. This method provides a clear, quantitative measure of methylation at specific sites, thus allowing for a precise examination of the role of m6A in *HTT1a* transcript regulation. Of course, as many other approaches it has its limitations. It allows quantification of only a subset of m6A sites that both occur at ACA sites and are within suitable distances of adjacent ACA sites. But importantly, one advantage of this method is that can be performed with low input RNA.

In an attempt to map other possible m6A sites in *HTT* intron 1 we have set up an alternative assay GLORI (Liu et al 2023; doi: 10.1038/s41587-022-01487-9) coupled to PCR. This method deaminates unmodified A resulting in a very GC rich sequence for subsequent PCR. Given the already high GC content of *HTT* intron 1 this deamination resulted in a sequence difficult to amplify. Efforts to fully dissect RNA methylation in *HTT* in human samples are ongoing and will take long to be completed, and therefore will constitute future studies.

To accurately map m⁶A sites in *Htt1a* we performed Nanopore RNA direct sequencing in the polyA RNA obtained from striatum of *Hdh*^{+/Q111} mice. Using this approach, we could map m⁶A sites in the chimeric sequence of *Htt1a*. Among the sites identified, the GGACA site located in the human sequence of this chimera show the highest confidence. This is exactly the site that was interrogated in human samples by MazF-qPCR which strengthen our results. **(new Fig. 3A).**

Is this changing in different samples with different CAGs? And in different brain regions?

To analyze m6A methylation in samples with different CAGs, we used fibroblasts with varying CAG repeats and as previously described in page 2 of this rebuttal we observed a significant albeit moderate correlation when including HD fibroblasts with 80 and 180 CAG repeats. In postmortem brain regions, we analyzed one region only, the putamen due to the number of samples available for the study providing confidence in our results. In addition, we have performed MazF-qPCR analysis in cortex from postmortem samples and observed an increase ratio of methylation (see below), but the number of samples is insufficient to draw conclusive results. Therefore, we prefer not to include these data in the manuscript and instead to show the data from the Striatum/Putamen.

-A better control of the MeRIP method would be suggested. Additional quality controls of the experiments, adding negative and positive controls with regions certainly methylated and not methylated, would strengthen the solidity of the data.

As suggested by the reviewer, we have added the MeRIP-qPCR analysis of *I1-pA1* transcripts along with *Grm1* as a positive control and *Rps14* as a negative control. These genes have been selected because they are abundant transcripts characterized by the presence and absence of m6A peaks, respectively. (Meyer *et al* 2012 <https://doi.org/10.1016/j.cell.2012.05.003>). The results are shown in the **new Appendix Fig. S1A** as fold enrichment (m6A/Input or unbound/Input). We have also included MeRIP-qPCR relative m6A enrichment for I1-3' and I3 analyzed in the striatum of 8-month-old WT and KI mice (**new Appendix Fig. S1B**). These control experiments are described in result section in **Page 17, lines 427-439**

Reviewer 2:

Points that need clarifications

1-Figure 1

I find a little difficult to draw conclusions about the correlation between m6A enrichment and the expression of the spliced isoforms particularly I1-pA2 at 2 months. The sharp increase of I1-pA2 transcripts does not appear to be due to m6A accumulation.

We have revised the data and there was a mistake plotting values in **Fig. 1b**; I1-pA2 correspond to I1-pA1 and values for I1-pA 1 correspond to I1-pA2. We have corrected data in the graph and apologize for the mistake. Anyways, we agree with the reviewer that the increase of I1-pA2 transcripts might not be due to m6A accumulation at this early stage of the disease. We further confirmed this result adding three more WT and KI samples to address potential sample size issues. As discussed in Background and Discussion sections a mechanism involved in the generation of *Htt1a* have been already described by *Neueder et al 2018*. *Htt1a* generation is associated with expanded CAG repeats influencing the splicing factors and the speed of RNA polymerase within the *HTT* gene. This could essentially result in a non-1:1 correlation between the increase in m6A and the increase in pA1/pA2. It is possible that deposition of m6A contributes to generation of these splices form at later stages of the disease.

The correlation between m6A and *Htt1a* production in this study is better addressed by a targeted m6A demethylation of m*Htt* RNA *in vitro*. Using this approach, we reduced *Htt1a* transcripts in 20-30% suggesting that m6A deposition contribute to a more complex mechanism involving not only the m6A machinery but splicing factors, Pol II and CAG repeats expansions.

It's important also to note that MeRIP-qPCR is semi-quantitative, providing relative measures of m6A enrichment rather than absolute quantification. The method typically shows fold enrichment compared to input RNA, but not the exact number of m6A-modified transcripts. Normalization to input RNA, as we performed in our analysis, can reduce some variability, but differences in immunoprecipitation efficiency remain (Meyer, K. D., & Jaffrey, S. R., 2017, doi: 10.1146/annurev-cellbio-100616-060758; Liu, et al, 2013, doi: 10.1261/rna.041178.113).

Moreover, wouldn't we expect a decrease in the FL transcript expression here?

As described in the longitudinal RNA-sequencing analyses of mHtt allelic series of Knock-in mice published by Langfelder et al. (doi: 10.1038/nn.4256) FL-Htt transcripts show a slight significant reduction in the striatum of Q111 mice at 6 mo and 10 mo of age but not at 2 mo (Table with the data extrated from Langfelder study is attached below for reviewer consideration). Accordingly, our results show a signification decrease at 8 mo of age but not at 2 mo, which is described in results section (Page 17, lines 425-426)

	Symbol	Name	baseMean.Q	log2FoldChange.Q.111.vs.20	pvalue.Q.111.vs.20	FDR.Q.111.vs.20
Striatum 2mo	Htt	huntingtin	5670	0,0586	0,697	0,927
Striatum 6 mo	Htt	huntingtin	7970	-0,117	0,0000158	0,000375
Striatum 10 mo	Htt	huntingtin	4900	-0,163	0,0158	0,0931

2) in figure 4: I'm also surprised that in YACs128 MEFs at 10uM of inhibitor there is no significant decrease of GGACA methylation while you observe a strong decrease of I1-pA1 transcripts again going against a direct link between methylation and increased htt1a. It may also suggest that other sites are more important than those analyzed. This leads us to criticize the fact that there is no initial mapping of methylated sites along htt transcripts using alternative methods, perhaps calling into question the choice of sites tested.

We acknowledge that the data show a strong decrease in I1-pA1 transcripts without a significant reduction in GGACA methylation at the 10 μM concentration in YACs128 MEFs. As reviewer suggest, this could indicate that other methylated sites that we have not analyzed might also contribute to the observed decrease in I1-pA1 transcripts. However, it is also possible that the concentration of 10 μM STM2457 may not be sufficient to fully inhibit METTL3 activity in these MEFs cells which have a different cellular context than the *STHdh^{Q111/Q111}*. Moreover, these MEFs cells express human *Htt1a* transcripts which might be more resistant to methylation changes due to their sequence and structure compared with the chimeric *Htt1a* expressed in *STHdh^{Q111/Q111}*. We have addressed these observations by elaborating on these potential explanations in Result section (Page 21, lines 550-558)

As explained in Discussion section (page 27) STM2457 also exert effects on the transcriptional machinery or splicing factors that indirectly influence the generation of *Htt1a* transcripts independently of direct methylation changes at GGACA or other methylation sites as a global decrease of m6A has very profound impact on the cellular machinery. Indeed, we observed a significant decrease in global methylation levels after STM2457 treatment which support our hypothesis that this could affect methylation and expression levels of other factors involved in Htt1a biogenesis. To have a better link between methylation and levels of *Htt1a* we designed a targeted demethylation system which demonstrates that site specific demethylation of the GGACA site influence expression of *Htt1a* transcripts.

We agree with the reviewer's suggestion that we may be underestimating the potential role of other sites due to incomplete m6A mapping along the *HTT* transcript. Here, we interrogated selected m6A sites in *Htt* intron 1 from different HD cell lines based on the location of a m6A peak previously detected as significantly enriched in the first 523 bp of *Htt* intron 1 RNA via MeRIP-seq. To add quantitative data on specific methylation sites, we have used the MazF-qPCR which relies on the ability of the bacterial RNase MazF to cleave RNA at unmethylated sites occurring at ACA motifs but not at the methylated counterparts m6A-CA. Therefore, we choose potential m6A sites containing ACA motifs located within this first 523 bp of *Htt* intron 1, a human (hm) GGACA and two mouse (ms) motifs, AGACA and GGACA (**Fig. 2A**). For further clarification we attached below a more detailed scheme of the analysed region.

-This leads us to criticize the fact that there is no initial mapping of methylated sites along *htt* transcripts using alternative methods, perhaps calling into question the choice of sites tested

To accurately map m6A sites in *Htt1a* we performed Nanopore RNA direct sequencing in the polyA RNA obtained from striatum of *Hdh*^{+/Q111} mice. Using this approach, we predicted the m6A sites in the chimeric sequence of *Htt1a*. This analysis confirmed the methylation in other m6A motifs that we haven't analyzed by MazFqPCR, however, it also revealed m⁶A modifications in those sites that were assessed in KI cells (**new Fig. 3A**). Among them the GGACA site for which our results demonstrate change in methylation levels after METTL3 inactivation and knockdown as well as targeted demethylation. These new results are described in Result section (**Page 18, lines 469-477**)

We are very confident in the experiments described in this manuscript, which strongly implicate m6A methylation at the GGACA site as being critically involved in *Htt1a* biogenesis. We believe the novel data presented may be of high interest to others and therefore worthy of timely publication. A more complete and in-depth characterization of methylation at *HTT* transcripts is ongoing and will be the focus of future studies

3) It is also difficult to interpret the relatively low effect on GGACA and the lack of effect of the METTL3/METTL14 inhibitor on the methylation status of other sites. It would be necessary to compare these experiments for example after treatment with siRNA against METTL3/METTL14.

We understand the reviewer's concern on a relatively low effect on GGACA and the lack of effect of the METTL3 inhibitor on the methylation status of other sites. Following the reviewer's suggestion, we have conducted an siMETTL3 experiment in *STHdh^{Q111/Q111}* cells with two different siRNAs and one pool of three different siRNAs. This is shown in a **new Figure EV3**, and described in the results section (**page 22, lines 570-578**). We knocked-down METTL3 RNA expression to around 50%. This reduced expression of METTL3 is sufficient to reduce the methylation on GGACA site along with a reduced expression of *I1pA1* and *I1pA2* transcripts in KI cells. We observed a similar decrease in the methylation ratio at the GGACA site, around 30%, consistent with the effect obtained with METTL3 inhibition. In the stoichiometric measurement of m6A it should be considered that m6A methylation is a dynamic process with constant addition and removal by methyltransferases and demethylases, respectively. Even with METTL3 inhibition some m6A modifications might have already been added by METTL3 before inhibition and wouldn't be affected.

Regarding the methylation status of other sites, previous findings have proposed that it is likely that every DRACH site could be methylated to some degree, and DRACH sites should not be considered methylated or unmethylated, but instead should be considered along a wide degree of methylation. Our data suggest that initial methylation levels in those sites is low and the assay might not be sensitive enough to detect small reductions in m6A methylation. Typically, fewer than 20% of transcript copies in the cell will have m6A at a specific site, with a small subset of DRACH sites having methylation as high as 20% or possibly even higher (*Murakami and Jaffrey, 2022; DOI: <https://doi.org/10.1016/j.molcel.2022.05.029>*)

Considering the relevance of this concern we have including these explanations for better interpretation of our results in Discussion section (**Page 28, lines 725-744**)

We would also like to highlight that MazF-qPCR results for the GGACA human site are consistent between the different experiments conducted in this study: METTL3 inhibitor, siMETTL3 and targeted demethylation indicating that this site is more methylated than others in *mHtt* RNA.

A more global validation of the inhibitory effect of STM2457 on m6A levels in selected cell lines would also be required.

For a global validation of the inhibitory effect of STM2457 and siMETTL3 on m6A levels we have used the EpiQuik™ m6A RNA Methylation Quantification Kit (Colorimetric) which can directly detect the m6A RNA methylation status using total RNA. Results are shown in **Appendix S6A and Figure EV3D** and described in the results section (**Page 21, lines 539-540 and Page 22, lines 573-574**).

4) Relative to Figure5: Regarding DNA damage experiments with γ -H2AX, I'd also like to see controls other than NT-gRNA to avoid the possibility that this NT-gRNA context might somehow increase DNA damage, affecting the overall conclusion.

We thank the reviewer suggestion. We have included in the analysis % of γ -H2AX positive nuclei and mean of γ -H2AX foci per cell of the control dCas13b-gRNA2 which contains the targeting gRNA2 and lacks the ALKBH5. This control (named gRNA2) has also been used in the RNA decay experiment. These results have been included in **new Fig. 5G**, and shortly described in the results section (**Page 25, lines 639-640**)

5) Figure 6: An obvious missing experiment that could add mechanistic value would be to test whether blocking CAG repeats affects METTL3 recruitment or function. In other words, can the length of CAG repeats promote METTL3 recruitment?.

We thank the reviewer for raising this point. We agree that analyzing the effect of blocking CAG repeats on METTL3 recruitment can add mechanistic value. To address this, we have blocked CAG repeats with LNA-CTGs and evaluated the direct binding interaction between METTL3 and *Htt* transcripts by RIP-qPCR assay. See **new Fig. 6E-6H**.

As the reviewer may notice in the representative Western Blot (WB) (Fig. 6E), an additional band above the one representative of METTL3 bound to the beads is also observed in the IP fraction of the ribonucleoprotein lysates from *STHdh^{Q111/Q111}*, with this band being increased in LNA-CTG treated cells, compared with LNA-SCB. This suggests that METTL3 might be interacting with other proteins in a complex in *STHdh^{Q111/Q111}*, particularly, when CAGs are blocked. If these interactions are strong enough, the complex may not fully dissociate under the denaturing conditions of SDS-PAGE, causing it to migrate more slowly during electrophoresis and appearing as a band above the expected size.

The new results described in Results section (**Page 26, lines 629-637**) showed that METTL3 interact with *Htt1a* transcripts remarkably in *STHdh^{Q111/Q111}* cells but not in *STHdh^{Q7/Q7}* which would suggest that METTL3 recruitment in intron 1 is promoted by long CAG repeats. Previous findings have shown that CAG length slow PolIII transcription rates favoring the production of the incompletely spliced HTT exon 1 transcript (*Neueder et al 2018*). Notably, recent evidences have revealed that during transcription, the presence of a paused Pol-II could inhibit the folding of surrounding RNA sequences, thus increasing the likelihood of m⁶A methylation near pausing sites (*Wang et al 2024 <https://doi.org/10.1093/nar/gkac169>*). In agreement with these evidences, our results revealed an increased methylation in *Htt* intron 1 and an increased interaction of METTL3 in *I1-pA1* and *I1-pA2* transcripts in the presence of long CAG repeats (New Fig. 6f and 6g). We also observed that this interaction was significantly reduced in *STHdh^{Q111/Q111}* cells treated with LNA-CTGs. These results are discussed in Discussion section (**Page 32 and 33, lines 845-854**)

Minor comments:

- **Problem with links to Fig5 in the text.** As suggested by reviewer 2 and 3 I have added missing links to Fig 5 in the text. Please, note that the order of panels in **Fig. 5** have changed to better referenced each panel in the text (e.g: panel 5e, 5f and 5g are now Fig. 5e)

- **It would be interesting to describe the rationale of the choice of the position of gRNAs.**

As it was explained in Result section (**Page 18**) and shown in **new Figure EV2A and EV2B**, m⁶A enrichment (m⁶A peak) was detected in the first 523 bp of intron 1. In this region there are several potential DRACH (D=A, G or U; R= G or A; and H=A, C or U) consensus motifs that our analysis with DRS predicted to be methylated. Our gRNAs were designed based on this m⁶A peak detected in the intron 1 sequence of *Htt* RNA, upstream of the first cryptic polyA site. gRNAs were chosen to target positions upstream and downstream of the different m⁶A sites analyzed by MazF-qPCRs. The three gRNAs at distinct positions were designed to be 30 nt long and 100-300 nt away from those methylated sites to enhance demethylation efficiency of the system

according to the characterization of a CRISPR–Cas13b-based tool for targeted demethylation of specific mRNA described by *Li et al 2020* (doi: 10.1093/nar/gkaa269). As suggested by the reviewer we had added this rational in Results section (**Page 23, lines 589-596**)

- Only males have been analyzed. Are there any sex differences in pathogenicity?

Although we recognize the importance of including both sexes in research studies we have used only male mice for the following reasons:

- Our previous studies analyzing m⁶A in *Hdh*^{Q7/Q111} mice have been performed in male mice, which provides us a robust baseline and comparative data. This consistency allows for better comparison and validation of new findings in these HD mice.

- Focusing on male mice ensures that the observed effects are not influenced by sex-specific differences in m⁶A regulation, allowing for clearer interpretation of how these modifications impact HD pathology, in particular, the generation of *Htt1a*. Female mice experience hormonal cycles (estrous cycles) that can introduce variability in gene expression and molecular pathways, potentially confounding the results. Male mice do not have such cycles, leading to more consistent baseline measurements and reducing variability. Indeed, previous studies have shown that estrogen induces m⁶A demethylase FTO nuclear aggregation through the mammalian target of Rapamycin signaling pathway (*Zhu et al 2016* <https://doi.org/10.3892/or.2016.4613>) and overexpression of FTO gene by activating PI3K/AKT and MAPK signaling pathways (*Zhang et al 2012 DOI: 10.1016/j.canlet.2011.12.033*). Moreover, METTL1414 levels can be strongly influenced by chromosomal sex composition and there are sex-specific differences in RNA methylation (*Salisbury et al 2024 DOI: 10.1038/s42255-021-00427-2*) and m⁶A modification modifications deposited in a sex-specific manner (*Wang et al 2021doi: 10.1038/s41467-021-22424-5*) facilitating the alternative splicing of specific genes (Sxl) in female flies.

By focusing on male mice, we aimed to minimize these confounding factors and provide more robust and interpretable results. Anyways, it would be interesting to explore in future studies m⁶A sex differences in the HD transcriptome and particularly in relevant genes altered in the disease, besides *Htt1a*.

For reviewer consideration, here we include a graph showing methylation ratio in HD human fibroblasts samples where no differences can be observed between genders (7 controls and 12 HD adult fibroblast cell lines):

-sentence 388-389 repetition of m6A not needed. Repetition was removed

-Paragraph 396-401 should be rephrased. As suggested by the reviewer we have rephrased this paragraph (**Page 21, lines 544-549**) as follows: “When analyzing by qPCR the levels of FL-Htt and *Htt1a* intronic sequences upstream of the first cryptic polyA site (I1-pA1), a significant reduction in I1-pA1 was observed in *STHdh*Q111/Q111 cells at both concentrations of STM2457 compared

to the vehicle (Figure 4b). No changes were detected in the relative levels of intron 3 (Supplementary Fig. 7a) or in the levels of FL-Htt and I1-pA1 sequences in STHdhQ7/Q7 cells treated with STM2457 (Supplementary Fig. 7b)". Please, note that previous **Supplementary Fig. 3 is now Appendix Fig. S6B and S6C)**

-Ref24 not cited at the right place in introduction. We agree with the reviewer that the sentence should be rephrased to correctly cite the reference, as this study does not describe a role of m6A in PolII pausing in introns. We have corrected this sentence to first present the evidence related to m6A-mediated control of RNAP II pausing and then discuss its role in introns.

Reviewer 3:

Concerns:

- Line 320-321 / Figure 1A: I3 and FL probes were described but not labeled properly in text as in the figure, we recommend that I3/FL be added for consistency.

As suggested by the reviewer we have corrected nomenclature in the text (**Page 16, lines 418 and 419**) and in the Fig. 1A.

- Line 325 / Figure 1B: "The levels of intronic sequences I1-3' were comparable to intron 3;" the graph in figure 1B is convincing, however were there any statistical tests carried out for this?

Student's two-tailed t test has been performed to compare levels of both intronic sequences showing no statistical differences ($p=0,2914$). To better clarification we have added the statistical tests in Fig. 1 caption (**Page 45, 1289 and 1290**)

- Line 356: We suggest the following text changes: Supplementary Figure 2 \rightarrow Supplementary Figure 2a & 2b.

As suggested by the reviewer we have introduced these changes in the text. Please, note that **Supplementary Fig. 2 is now the Figure EV2**

-Line 356-359: Can the authors clarify how the two additional m6A motifs were selected, Supplementary figure 2 only shows enrichment for the first m6A site.

Please, note that **supplementary Fig. 2 is Figure EV2.**

As it was explained in Result section (**now Page 17, lines 468-478**), our previous study revealed by MeRIP-seq, an m⁶A enrichment (m6A peak) in the first 523 bp of intron 1, not any particular m⁶A site (EV Fig. S4A and S4B). In this region there are several potential DRACH (D=A, G or U; R=

G or A; and H=A, C or U) consensus motifs that we demonstrate with a new analysis (nanopore Direct RNA sequencing) that are methylated. m⁶A sites containing ACA motifs were chosen to be analyzed by MazF-qPCR based on the location of this m⁶A peak previously detected as enriched in *Htt* RNA as well as by the prediction of m⁶A sites (New Fig. 2A). Motifs for analysis using this approach ideally reside near an ACA site. However, it's also crucial to select control regions lacking ACA sites. These control regions serve as negative control sites, where the enzyme cannot digest the target amplicons, allowing for normalization of their levels. MazF recognizes and cleaves RNA specifically at ACA sequences, which limits its applicability to other RNA methylation sites that do not contain ACA motifs in this intronic region. This limitation has also been highlighted in Discussion section, **(Page 28, lines 738-744)**

Line 399-401 / Supplementary Figure 3: Text should be modified to contain 3A and 3B where appropriate.

As suggested by the reviewer we have introduced these changes in the text **(Page 21, line 545 and 548)**. Please, note that this **Supplementary Fig. 3 is now Appendix Fig. S6**

Line 438 / Figure 5D and 5E?: Missing reference to Figure 5D in text. Description of Figure 5E is missing from text.

As suggested by the reviewer we have introduced the missing reference and description regarding Fig. 5. Please, note that the order of panels in Fig. 5 have changed to better referenced each panel in the text (e.g: panel 5e, 5f and 5g are now **Fig. 5H**)

Figure 5G: the lighter and darker grey color is hard to see, may not be that important since there was no difference in RNA decay between conditions.

As suggested by the reviewer we changed the colors in the new Figure 5 to facilitate visualization of the different conditions.

Figure 5H, 5I and 5J: 5H & 5I Graphs are missing data points. 5J can benefit from inlets showing differences of gamma H2AX foci between the two conditions.

As suggested by the reviewer we have included in **new Fig. 5H (former 5h and 5i)** graphs with data points representing the means of the different experiments analyzed.

Supplementary Figure 6: A & B matches Figure 6 A & B and is not referenced in text. 6A can benefit from larger texts. 6B can benefit from mentioning of top and bottom.

We have removed Fig. 6a and 6b from supplementary material. As suggested by the reviewer we have referenced and explained Fig. 6a and 6b in results section **Page 24**

Line 483: From figure 6A schematic, HTT-e1f and HTT-e1r are primers that are outside the CAG repeats and HTT-e1*f and HTT-e1*r are primers spanning the CAG, the text mentions "PCR with primers amplifying Exon 1 outside the CAG repeat (HTT-e1*)..." can this be clarified.

We have corrected the mistake in the text. As Fig. 6a schematic shows, *HTT-e1f* and *HTT-e1r* are primers that span outside the CAG repeats while the *HTT-e1*f* and *HTT-e1*r* are primers spanning the CAG repeat

Line 485: "However,..." can be removed.

We have rewritten this sentence

Line 363-366 / Figure 2B: It would be useful to the field to understand if the methylation preference pattern between the 3 m6A sites are altered between HD and Controls, since the authors earlier showed (Figure 1) that m6A is increased in HD at I1-pA1.

The mouse model used in this study is the *Hdh*^{+Q111}, which contains a chimeric mouse/human genetic construct in murine *Htt* not present in control *Hdh*^{Q7/Q7} mice (which only have the mouse sequence). One of the three analyzed m⁶A motifs (GGACA) is exclusively present in the human insert, so no comparison could be performed with controls. Nevertheless, we have included in **Figure EV2D** a graph showing methylation ratio of these m⁶A sites in the striatum of control and HD mice with the corresponding description in Result section (**Page 16**). Analysis of Direct RNA sequencing in PolyA(+) selected RNA WT samples didn't show coverage for *Htt1a*. This is accordance with previous studies (*Sathasivam et al., 2013 DOI: 10.1073/pnas.1221891110* *Neueder et al. 2017, doi: 10.1038/s41598-017-01510-z*) and the RACE-PCR shown in this manuscript (**Fig. 1H and 1I**)

Line 396-399 / 403-405: Can the authors hypothesize why decreasing I1-pA1 levels does not rescue FL-*Htt* levels? Or relate to previous findings around levels of the exon 1 protein versus FL.

The production of *HTT1a* is through the alternative processing of the *HTT* pre-mRNA, and therefore, you would expect that a reduction in *HTT1a* would result in an increase in full-length *HTT*. However, the presence of *HTT1a* does not result in a reduction in full-length *HTT in vivo*. This has been recently shown in Q111 mice (*Landles et al (2024; https://doi.org/10.1101/2024.06.11.598410)*). There may be additional regulatory mechanisms influencing FL-*Htt* mRNA levels. Indeed, recent studies have shown evidence of global transcriptome polyadenylation in HD brains and mouse models, with the *HTT* gene itself exhibiting lengthened poly(A) tails (*Pico et al., 2021; DOI: 10.1126/scitranslmed.abe7104*). This altered polyadenylation can increase mRNA stability, potentially masking changes in expression levels resulting from a shift from incomplete splicing to constitutive splicing. We have further discussed this hypothesis in **Page 30-31, lines 797-801**

Figure 6: from Figure 6C, transfected cells with LNA-CTG show decreased transcript levels of *HTT1a* which would be expected when blocking the CAG expansion. Can the authors clarify or

rule out the possibility that the significant decrease in the m6A methylation ratio seen in Figure 6D is not simply due to lower HTT1a levels caused by LNA-CTG through an m6A-independent manner. From the citation [34] of LNA-CTG used, it seems like LNA-CTG will decrease mHTT, and if m6A is present more so on the mHTT transcript, then decreasing it will likely decrease m6A levels measured by the assays presented in this study. There could be multiple interpretations and as the section stands, it is overinterpreted.

We thank reviewer suggestion for further clarification regarding MazF-qPCR analysis. In order to detect the m6A methylation ruling out the effects of global *Htt1a* reduction due to LNA-CTG treatment, the MazF-qPCR approach uses a formula where methylated *Htt1a* intronic sequence is normalized against total levels of this intronic sequence. This ensures that the observed decrease in m6A levels observed in different treatments or conditions is not due to a reduction in RNA levels but rather a true reduction in methylation levels. The formula used is as follows:

$$\text{Methylation ratio} = \frac{\text{Test motif/ no ACA motif (MazF digested)}}{\text{Test motif/ no ACA motif (MazF non digested)}}$$

To avoid multiple interpretations, we added this explanation in the results section (**Page 22, lines 566-569**) and in **new EV Fig. S2C**.

On the other hand, as already mentioned in results section of **New Fig. 6 (page 24)**, Rue et al have previously demonstrated that locked nucleic acid-modified antisense oligonucleotide complementary to the CAG repeat (LNA-CTG) preferentially binds to mutant *Htt* without affecting *Htt* mRNA or protein levels (*doi*: 10.1172/JCI83185). Indeed, it is proposed that the beneficial activity of LNA-CTG observed *in vivo* may be due to its blocking activity of the CAG repeat expansion in *HTT-e1* RNA. Here, we confirmed that LNA-CTG does not affect the expression of FL-*Htt* but reduces the expression of *Htt1a*. Since we also observed decreased interaction of METTL3 with I1-pA1 and I1-pA2 sequences (**New Fig. 6e-h**), this result indicates that the beneficial effect previously observed by Rue et al might be also due to a mechanism involving an interplay between CAG, METTL3, and potential m6A readers which might be interacting with the spliceosome pathway to contribute on *Htt* incomplete splicing. We extended discussion of these results in Discussion section (**Page 32 and 33, lines 845-854**).

Dear Veronica,

Thank you for the submission of your revised manuscript. We have now received the enclosed reports from the referees that were asked to assess it, and I am happy to say that all support its publication now. All referees still have a few more minor suggestions that I would like you to incorporate before we can proceed with the official acceptance of your manuscript.

A few editorial requests will also need to be addressed:

- Please reduce the number of keywords to 5.
- Please add a "Data Availability Section" to the end of the Methods that lists accession numbers and direct links to data deposited in public databases.
- Please correct the conflict of interest subheading to "Disclosure Statement and Competing Interests"
- Please correct one of the 2 names: Ulf Andersson Vang Orom in the ms file versus Ulf A Orom in our online submission system.
- Please remove the author credits from the ms file. All credits need to be entered during online ms submission.
- Please correct the reference format to our alphabetical EMBO reports (Harvard) style.
- Please send us a new, fully completed author checklist. Especially cells D87, D88, D96, D112 need to be filled in.
- The FUNDING INFO needs to be part of the Acknowledgments section. Also, the following info does not match:
 - i) Ministerio de Economía y Competitividad is in the ms file while Ministerio de Ciencia, Innovación y Universidades (MCIU) is in eJP (our online submission system)
 - ii) PID2020-116474RB-100, PID2020-113953RB-I00 and RTI2018-094374-B100 are missing in eJP
 - iii) PhD grant program by La Generalitat de Catalunya (2018FI_B_00487) is missing in eJP
 - iv) PhD grant from PID2020-116474RB-100 (RE2021-097199) is missing in eJPPlease correct.
- Please add missing callouts for Fig. 1C, Fig. 2E. The Appendix Figure S8 is called out but the figure is missing, please correct.
- Tables EV1-EV7 that are provided in the Appendix file need to be removed from that file and uploaded as individual Table EV1-7 files. Table EV1 is labeled twice while a Table EV2 label is missing. Some of the EV tables could be moved to the Reagent and Tools table that is currently missing, or links to the EV tables can be included in the R&T table.
- Since 1st of July 2024, a Reagents and Tools Table (listing key reagents, experimental models, software and relevant equipment and including their sources and relevant identifiers) needs to be published with every ms. Please download and fill our Reagents and Tools Table template (.docx), which you can find in our author guidelines:
<https://www.embopress.org/page/journal/14693178/authorguide#structuredmethods>.
Please do not include the Reagents and Tools Table in the Methods section of the manuscript but upload it as a separate file choosing the file type "Reagent Table".
- "Background" needs to be correct to "Introduction".
- The manuscript sections should be in the following order: Title page - Abstract & Keywords - Introduction - Results - Discussion - Methods - Data Availability - Acknowledgments - Disclosure Statement & Competing Interests - References - Figure Legends - (Main Tables with legends) - Expanded View Figure Legends.
- The Consent for publication section should be removed.
- The nomenclature for EV figure legends needs to be updated to "Figure EV1", etc. instead of "Expanded View Figure 1"
- Please note that the legends for figures 4b-e are not provided in a sequential manner (legend for figures 4c, e; are provided before legends of figures 4b, d). This needs to be rectified.
- Please note that the exact p values are not provided in the legends of figures 1b, d-g; 2c-e; 3b-c; 4a-d, f; 5c-e, g; 6c-d, f-g; EV 1a; EV 2e; EV 3a-b, d-f.
- Please note that in figure 5h; there is a mismatch between the annotated p values in the figure legend and the annotated p

values in the figure file that should be corrected. Also, please provide the exact p-values for the same if applicable.

- Please note that the box plots need to be defined in terms of minima, maxima, centre, bounds of box and whiskers, and percentile in the legends of figures 5d-e.

- Please note that information related to n is missing in the legends of figures 5h; EV 3f.

I would like to suggest a few minor changes to the abstract that needs to be written in present tense. Do you agree with this:

In Huntington's disease (HD), aberrant processing of huntingtin (HTT) mRNA produces HTT1a transcripts that encode the pathogenic HTT exon 1 protein. The mechanisms behind HTT1a production are not fully understood. Considering the role of m6A in RNA processing and splicing, we investigated its involvement in HTT1a generation. We show that m6A is enriched in intronic sequences before the cryptic poly (A) sites and we identify m6A methylated sites in intron 1 in the mouse and human sequence of Htt1a in the striatum of Hdh^{+/Q111} mice. This intronic m6A methylation is also seen in human HD samples. We further assess m6A's role in mutant Htt RNA processing by pharmacological inhibition and knockdown of METTL3, as well as targeted demethylation of Htt intron 1 using a dCas13-ALKBH5 system in HD cells. Our data show that Htt1a transcript levels in HD cells are regulated by both METTL3 and the methylation status of the Htt intron. They also suggest that m6A methylation in intron 1 depends on expanded CAG repeats, highlighting a role for m6A in the aberrant splicing of Htt RNA.

It is not clear to me what the difference between HTT RNA and Htt RNA is. Is this human versus mouse? And are HD cells (in the abstract) human cells? In this case, Htt versus HTT needs to be correct, I think. Please check carefully.

It would also be better if the word "regulate" in the title and abstract could be replaced by something more specific. For example, would "promote" be correct?

EMBO press papers are accompanied online by A) a short (1-2 sentences) summary of the findings and their significance, B) 2-3 bullet points highlighting key results and C) a synopsis image that is exactly 550 pixels wide and 200-600 pixels high (the height is variable). The synopsis image should provide a sketch of the major findings, like a graphical abstract. Please note that text needs to be readable at the final size. Please send us this information along with the final manuscript.

Referee #1:

I have now completed the review of the paper entitled by 'm6A modification of mutant huntingtin RNA regulates the biogenesis of pathogenic huntingtin transcripts' by Anika Pupak et al.

Most of my concerns were addressed in a satisfying manner, the study is definitely more complete and sounder.

Two points remain puzzling:

1. Related to the possible correlation between HD stage of pathology, CAG length and CAG somatic expansion in the HD post mortem brains, the data doesn't support solid conclusions since i. the authors detected an increase in methylation ratio at the GGACA site in the putamen samples from Vonsattel grades (2-3, and 3), but the number of human post mortem samples is still quite limited to draw final conclusions; ii. the data shows quite some variability (VG 1-2, 2 and VG 2-3, 3 in Figure 3 A) and iii. the link between m6A methylation levels does not correlate with the levels of HTT1a in the putamen of human post mortem samples - as reported in Appendix Fig. S4A and S4B - high levels of methylation are detected with no significant increase HTT1a. The link between M6A methylation and HTT1a production in human HD brains with overt HD pathology is still lacking, thus conclusions about the relevance of m6A methylation for HD pathology and for possible therapeutic development need to be tuned down - starting with this sentence in the abstract 'highlighting m6 A's role in aberrant splicing and its therapeutic potential.', but also in the discussion of the paper.

2. Relevance of the findings for human HD patients: alleles with lower CAGs. Most of the work is conducted in mouse animal models with very extended CAG tracts.

I appreciated the digression of the authors concerning somatic expansion of the CAG tract in the brain regions and how this process seems to be correlated to catastrophic cellular events leading to MSNs degeneration. However, the mouse models and the human post mortem observations do not seem to be completely in line - at least concerning the levels of HTT1a: in the mouse brain high methylation in intron 1 correlates with high levels of Htt1a, which, in turn, might be (at least partially) responsible for neuronal pathology. However, in human, this correlation doesn't replicate: this result on one hand doesn't fully support the relevance of HTT1a for the degeneration in the human pathology. Of note, in HD post mortem brains - as also pointed out by the authors - the CAG repeat expansion is expected to exceed the threshold of CAG>50, thus, HTT1a should be produced.

On the other hand, the different result between mouse and human, doesn't fully sustain the specific role of m6A methylation for HTT1a biogenesis. Thus, what might be the relevance of M6A methylation at the pre-mRNA level if HTT1a is not increased in the most relevant tissue for HD human pathology?

A caution statement highlighting these discrepancies needs to be inserted in discussion.

3. Text writing, labels, nomenclature

The writing sometimes seems very complex and not fully logic, this confounds the general readability of the manuscript. See this sentence in the abstract: 'We show that m6A is enriched in intronic sequences before the cryptic poly (A) sites and we identified m6A methylated sites in Htt1a intron 1 in the mouse and human sequence of Htt1a in the striatum of Hdh+/Q111 .mice.' As for this specific example, I would simplify, first presenting data related to the mouse, and, as such, also the knock-in HD models, then I would move to the human and, also, the HD post mortem brains. Punctuation needs to be checked.

Please check italics for genes and transcripts names (also in figure legends and labels in the figures) and line-up with international nomenclature standards.

I am puzzled by the inset in Figure 5B for the m6A methylation peaks - are the colors of the tracks correct? I would have expected higher methylation in mutant rather than WT cells.

Referee #2:

The experiments carried out in the revised manuscript by Pupak et al. and the answers to the reviewers' questions have brought substantial improvement to the manuscript. I have acknowledged the addition of important clarifications and experiments to the mechanistic view of the proposed model. For these reasons, I believe the revised manuscript is suitable for publication in EMBO reports and will bring interesting insights into the understanding of pathogenic mechanisms leading to Huntington disease.

I have noticed some minor mistakes that need to be corrected:

Line39: its involvement "in"

Line41: Hdh+/Q111 ". "

Fig2A: I think it would be easier for understanding if the m6A sites were numbered

The links to figures should also be checked as there are mistakes:

Line 506: Fig 2D not 2C

Line 508: Fig2E not 2D and 2C not 2B

Line 517: Fig2C and 2E not 2D

line 573: DRS not RDS

Referee #3:

The authors have addressed concerns. A few very minor concerns are as follow:

Minor:

Only figure 1 image had "Figure 1" labeled, the rest of the figures do not have figure labels. This will likely get fixed when article goes to press.

Line 444-450: the statement speculating the correlation between m6A and somatic repeat instability would be better suited for the discussion section.

Line 504-508: there may have been a mistake referencing 2C,2D,2E. When referencing zQ175 in the text, it pointed to 2C, then for YAC128 in the text it pointed to 2D. Please make sure correct.

Line 546-556: Appendix Fig. S6A was referred in the text, however it seems that Appendix S5A had the correct panel.

Dear Esther,

Thank you for supporting the publication of our manuscript. We have incorporated all the changes suggested by the journal and reviewers. We hope the manuscript now meets all the requirements for final acceptance.

Please, find attached our rebuttal for reviewer 1

1. Related to the possible correlation between HD stage of pathology, CAG length and CAG somatic expansion in the HD post mortem brains, the data doesn't support solid conclusions since

- i. the authors detected an increase in methylation ratio at the GGACA site in the putamen samples from Vonsattel grades (2-3, and 3), but the number of human post mortem samples is still quite limited to draw final conclusions;**
- ii. the data shows quite some variability (VG 1-2, 2 and VG 2-3, 3 in Figure 3 A) and**
- iii. the link between m6A methylation levels does not correlate with the levels of HTT1a in the putamen of human post-mortem samples - as reported in Appendix Fig. S4A and S4B - high levels of methylation are detected with no significant increase HTT1a. The link between M6A methylation and HTT1a production in human HD brains with overt HD pathology is still lacking, thus conclusions about the relevance of m6A methylation for HD pathology and for possible therapeutic development need to be tuned down - starting with this sentence in the abstract 'highlighting m6 A's role in aberrant splicing and its therapeutic potential.!', but also in the discussion of the paper.**

While we acknowledge that the sample size available may not allow for overly assertive and final conclusions, the results obtained are suggestive of *"the potential relevance and possible pathological significance of m6A methylation in mHtt RNA for HD patients"* (pages 529-531). Hence, we consistently state that the results "suggest" our interpretations. Each Vonsattel grade has 3-8 samples, and there are 7 control samples. While the sample size is moderate, it is not small either, especially considering the significant differences observed in methylation levels. Variability is normal, as these are samples from human putamen tissue. Each individual's piece of brain obtained for the analysis is unique, and the integrity of RNA can differ between samples due to variations in postmortem processing.

As we clarified to the reviewer in our previous rebuttal letter, in postmortem putamen samples, one cannot expect highly elevated levels of *HTT1a* based solely on the analysis of bulk RNA levels. At any given timepoint, extensive somatic CAG expansion (100-500+ CAGs) in HD brains, which can promote *HTT1a* generation, appears to be present only in a minority of striatal medium spiny neurons (Handsacker et al., 2024, <https://doi.org/10.1101/2024.05.17.592722>)E. Based on this novel understanding, it is worth noting that cells with hyper-expanded CAG repeats are relatively short-lived (i.e., undergo neuronal loss within months upon expanding beyond 150-180 CAGs) and therefore never become predominant. This means that when *HTT1a* is produced, it encodes the aggregation-prone and highly pathogenic HTT1a protein. As toxicity starts and neurons die, *HTT1a* itself becomes difficult to detect.

Reviewer 1 is suggesting that we significantly change the message of the paper where we demonstrate the role of m6A in the aberrant/incomplete splicing of *Htt1a/HTT1a*. While we can “tone down” the emphasis on therapeutic potential, we feel that we should not avoid suggesting that further exploration in this direction is worthwhile, especially considering that the other two reviewers have not expressed any concerns regarding this aspect.

Consequently, to address this reviewer’s concern, we propose to rephrase the following sentence in the abstract:

Original 1: "highlighting m6A's role in aberrant splicing and its therapeutic potential". (abstract)

Revised 1: “Collectively, our findings highlight the potential role for m6A in aberrant splicing of *Htt*.”

Original 2: “In addition, our data suggest that the methylation pattern may be linked to disease progression, highlighting its potential pathological relevance”. (in results section)

Revised: "Overall, these findings suggest the potential pathological relevance of m6A methylation in *mHtt* RNA for HD patients"

We have also changed the final section of discussion toning down the relevance for therapeutic development:

Original: Overall, our findings may lay the ground for the development of novel therapeutic strategies, since disease-modifying therapies targeting the pathological processing of *Htt* mRNA could be the most promising [98]. While strategies involving the use of ASOs targeting the *HTT* exon 1-intron 1 junction to lower *Htt1a* or modulate aberrant splicing in HD mutation carriers have been suggested [99], no evidence has been reported thus far. When designing ASOs to target *Htt1a*, it might be relevant to avoid the m6A-modified sites in *Htt* intron 1 RNA since they could hinder ASO binding or lead to ASO destabilization due to improper base pairing, thereby reducing ASO efficiency. Moreover, our evidence demonstrates that the CRISPR/dCas13b-ALKBH5 approach could allow a moderate reduction in the toxic mutant *HTT1a* fragment without affecting the WT allele. Intriguingly, a reduction of 43% in *mHtt* mRNA has been reported to be enough to prevent further brain loss in symptomatic R6/2 mice and significantly increase lifespan using nonallele-specific ASOs [100]. In contrast, our strategy selectively targets the biogenesis of a highly pathogenic transcript derived from *mHtt*, obtaining a reduction of approximately 20% by merely modifying the methylation status of *mHtt* RNA. Given that slight reductions in *mHtt* mRNA levels are enough to ameliorate HD symptomatology in mouse models, future experiments validating the effects of *Htt1a* demethylation in vivo could provide encouraging results.

CONCLUSIONS

This work provides evidence of the role of dysregulation of m6A modifications as a potential pathogenic mechanism influencing *mHTT* RNA metabolism. Although additional work is needed to deepen our understanding of the function of m6A in *mHTT* intron 1, this study establishes the basis for new gene therapy strategies based on this RNA modification profile to target the mutant RNA allele or to modify the outcome of the splicing reaction that generates *HTT1a*.

Revised: “Overall, our study provides new insights by demonstrating the presence of m⁶A modifications in mutant huntingtin intron 1 and its potential contribution to the pathogenic

mechanism influencing mHtt RNA metabolism. Importantly, our evidence may support the development of therapeutic strategies already proposed (Tabrizi et al, 2019) such as targeting the pathological processing of Htt mRNA or the HTT exon 1-intron 1 junction to lower Htt1a in HD mutation carriers. For instance, these modifications might be relevant to consider when designing ASOs targeting HTT1a, as the m⁶A-modified sites could hinder ASO binding or destabilize ASO due to improper base pairing, thereby reducing its effectiveness. Moreover, we demonstrate that the CRISPR/dCas13b-ALKBH5 approach could achieve a moderate reduction in the toxic mutant Htt1a fragment without affecting the WT allele. Intriguingly, using nonallele-specific ASOs, reduction of 43% in mHtt mRNA has been reported to be enough to prevent further brain loss in symptomatic R6/2 mice and significantly increase lifespan (Kordasiewicz et al, 2012). In contrast, our strategy selectively targets the biogenesis of a highly pathogenic transcript derived from mHtt, obtaining a reduction of approximately 20% by merely modifying the methylation status of mHtt RNA. Given that slight reductions in mHtt mRNA levels are enough to ameliorate HD symptomatology in mouse models, future experiments validating the effects of Htt1a demethylation in vivo could provide encouraging results.

Our study highlights the need for a deeper understanding of the pathogenic mechanisms influencing mHTT RNA metabolism, with a particular emphasis on RNA modifications. This understanding could open new avenues for novel gene therapy strategies aimed at targeting the mutant RNA allele or modifying the splicing process that generates HTT1a, both of which warrant further exploration”.

2. Relevance of the findings for human HD patients: alleles with lower CAGs. Most of the work is conducted in mouse animal models with very extended CAG tracts.

I appreciated the digression of the authors concerning somatic expansion of the CAG tract in the brain regions and how this process seems to be correlated to catastrophic cellular events leading to MSNs degeneration. However, the mouse models and the human post mortem observations do not seem to be completely in line - at least concerning the levels of HTT1a: in the mouse brain high methylation in intron 1 correlates with high levels of Htt1a, which, in turn, might be (at least partially) responsible for neuronal pathology. However, in human, this correlation doesn't replicate: this result on one hand doesn't fully support the relevance of HTT1a for the degeneration in the human pathology. Of note, in HD post mortem brains - as also pointed out by the authors - the CAG repeat expansion is expected to exceed the threshold of CAG>50, thus, HTT1a should be produced.

On the other hand, the different result between mouse and human, doesn't fully sustain the specific role of m6A methylation for HTT1a biogenesis. Thus, what might be the relevance of M6A methylation at the pre-mRNA level if HTT1a is not increased in the most relevant tissue for HD human pathology?

A caution statement highlighting these discrepancies needs to be inserted in discussion.

This is somewhat related to the previous comment. m6A methylation might be deposited co-transcriptionally and play a role only when the repeat expands enough to compromise the mechanisms previously described (in the paper as well as in the rebuttal) to be involved in incomplete splicing. The results showing methylation in putamen samples at pre-RNA are suggesting that this methylation is probably detected in neurons that haven't yet expanded enough to generate *HTT1a*. Again, as mentioned before, the fact that we couldn't

detect it doesn't mean it is not produced. Previous studies have tried to detect HTT1a transcripts in postmortem brain samples and were only able to detect this in samples from patients with Juvenile HD, where the inherited *HTT* CAG is much longer than in traditional HD (Hoschek et al., 2024; <https://doi.org/10.1186/s10020-024-00801-2>). As explained in that same study, due to phase transitioning <https://www.ncbi.nlm.nih.gov/pmc/articles/PMC5555642/> caused by extensive intra- and inter-molecular interactions of the expanded CAG tracts, *HTT1a* RNA clusters might not have been fully solubilized during RNA extraction (Ly et al., 2022; DOI: 10.1093/braincomms/fcac248). Therefore, it is possible that the amount of *HTT1a* generated is underestimated, because only the cytoplasmic, soluble *HTT1a* RNA fraction can be analyzed

Previous data (Neueder et al. DOI: 10.1038/s41598-017-01510-z) has also demonstrated that some levels of HTT1a may be present in both control and HD patients. However, the exon1 HTT protein with a polyglutamine tract in the normal range is not pathogenic. Most of the evidence in our study suggests that m6A contributes to the aberrant splicing of *HTT1a*. Since m6A can be detected in pre-RNA, we cannot discard its involvement in the production of *HTT1a* in HD pathology. *HTT1a* mRNA clustering has already been shown by Ly et al. in post-mortem brain using fluorescence in situ hybridization (FISH) technology.

Taking these points into consideration, we propose to include a caution statement in the discussion, noting the observed discrepancy, while still emphasizing the potential significance of m6A methylation in HTT1a/Htt1a biogenesis.

Caution statement included in the discussion:

“While our results support the role of m6A in the generation of Htt1a in HD mouse models, we cannot establish a correlation between the m6A methylation in mHTT intron 1 and levels of HTT1a in human HD brain cells. This discrepancy could be due to several factors. First, previous studies have reported challenges in detecting increased HTT1a expression in adult post-mortem HD brain samples (Neueder 2018, Hoschek et al., 2024). HTT1a can form RNA clusters in HD human brains (Ly et al., 2022; DOI: 10.1093/braincomms/fcac248) and might not have been fully solubilized during RNA extraction. This suggests that the amount of HTT1a generated might be underestimated, since only the cytoplasmic, soluble HTT1a RNA fraction can be analyzed. Second, bulk RNA analysis from postmortem putamen samples might not be a sensitive enough approach to detect significant HTT1a changes. Recent single-cell studies in HD postmortem brains suggest extensive somatic CAG expansions (>100 CAG repeats), which can promote HTT1a generation, appear to be present in only a minority of striatal medium spiny neurons at any given timepoint (Handsacker et al., 2024). Furthermore, it is generally understood that neurons with extreme CAG repeat expansions (150-180+ CAGs) are short-lived and undergo rapid cell death. This means that even when HTT1a is produced from such expansions, it encodes a highly toxic protein and may be difficult to detect due to the consequential neuronal loss. Our current findings in human HD samples suggest that m6A methylation is primarily detected in pre-mRNA. We hypothesize that this modification might play a role in the generation of HTT1a when longer CAG repeats are present in the mutant HTT gene, as observed in our HD mouse models and cell lines”.

Following reviewer suggestions, we have also edited some sentences different sentences that were complex/confusing.

We hope that you find the changes introduced to be satisfactory in addressing these two specific concerns from reviewer 1 and look forward to hearing from you.

All other referee concerns and all editorial requests have also been addressed by the authors.

Dr. Veronica Brito
Universitat de Barcelona.Institut de Neurosciències
Departament de Biomedicina, Facultat de Medicina, Institut de Neurosciències
Institut d'Investigacions Biomèdiques August Pi i Sunyer (IDIBAPS)
Centro de Investigación Biomédica en Red sobre Enfermedades Neurodegenerativas (CIBERNED), Madrid
Spain

Dear Veronica,

I am very pleased to accept your manuscript for publication in the next available issue of EMBO reports. Thank you for your contribution to our journal.
